

# Arctic hydroclimate variability during the last 2000 years – current understanding and research challenges

Hans W Linderholm[1]*, Marie Nicolle[2], Pierre Francus[3], Konrad Gajewski[4], Samuli Helama[5], Atte Korhola[6], Olga Solomina[7], Zicheng Yu[8], Peng Zhang[1], William J. D'Andrea[9], Maxime Debret[2], Dmitry Divine[10], Björn E. Gunnarson[11], Neil J. Loader[12], Nicolas Massei[2], Kristina Seftigen[1, 13], Elizabeth K. Thomas[14], Johannes Werner[15]

[1]Regional Climate Group, Department of Earth Sciences, University of Gothenburg, 40530 Gothenburg, Sweden
[2]UFR Sciences et Techniques, Université de Rouen Normandie, 76000 Rouen, France
[3]Institut National de la Recherche Scientifique, Centre Eau Terre Environnement, G1K 9A9, Québec, QC, Canada and GEOTOP Research Center, Montréal, QC Canada
[4]Département de géographie, Université d'Ottawa, Ottawa, Ontario K1N 6N5, Canada
[5]Natural Resources Institute Finland, Rovaniemi, Finland
[6]Environmental Change Research Unit (ECRU), Department of Environmental Sciences, University of Helsinki, 00014 Helsinki, Finland
[7]Institute of Geography, Russian Academy of Sciences, 119017 Moscow, Russia
[8]Department of Earth and Environmental Sciences, Lehigh University, Bethlehem PA 18015-3001, U.S.A.
[9]Lamont-Doherty Earth Observatory, Columbia University, Palisades NY 10964, U.S.A.
[10]Norwegian Polar Institute, Fram Centre, 9296 Tromsø, Norway
[11]Department of Physical Geography, Stockholm University, 10691 Stockholm, Sweden
[12]Department of Geography, Swansea University, Swansea SA2 8PP, Wales, UK.
[13]Earth and Life Institute, Université catholique de Louvain, 1348 Louvain-la-Neuve, Belgium
[14]Department of Geology, University at Buffalo, Buffalo NY 14260, U.S.A.
[15]Department of Earth Science, University of Bergen, 5020 Bergen, Norway

*Correspondence to*: Hans W Linderholm (hans.linderholm@gvc.gu.se)

**Abstract.** Along with Arctic amplification, changes in Arctic hydroclimate have become increasingly apparent. Reanalysis data show increasing trends in Arctic temperature and precipitation over the 20th century, but changes are not homogenous across seasons or space. The observed hydroclimate changes are expected to continue, and possibly accelerate, in the coming century, not only affecting pan-Arctic natural ecosystems and human activities, but also lower latitudes through changes in atmospheric and oceanic circulation. However, a lack of spatiotemporal observational data makes reliable quantification of Arctic hydroclimate change difficult, especially in a long-term context. To understand hydroclimate variability and the mechanisms driving observed changes, beyond the instrumental record, climate proxies are needed. Here we bring together the current understanding of Arctic hydroclimate during the past 2000 years, as inferred from natural archives and proxies and palaeoclimate model simulations. Inadequate proxy data coverage is apparent, with distinct data gaps in most of Eurasia and parts of North America, which makes robust assessments for the whole Arctic currently impossible. Hydroclimate proxies and climate models indicate that the Medieval Climate Anomaly (MCA) was anomalously wet, while conditions



were in general drier during the Little Ice Age (LIA), relative to the last 2000 years. However, it is clear that there are large regional differences, which are especially evident during the LIA. Due to the spatiotemporal differences in Arctic hydroclimate, we recommend detailed regional studies, e.g. including field reconstructions, to disentangle spatial patterns and potential forcing factors. At present, it is only possible to carry out regional syntheses for a few areas of the Arctic, e.g.

Fennoscandia, Greenland and western North America. To fully assess pan-Arctic hydroclimate variability for the last two millennia additional proxy records are required.

## 1. Introduction

Global climate is changing rapidly, largely due to increased human activities (IPCC, 2013). However, climate change is not uniform in space, and distinct regional differences in the magnitude of observed warming in recent decades are apparent. The

Arctic has warmed at more than twice the rate of the global average (Cohen et al., 2014). This *Arctic amplification* (Serreze et al., 2009) is due to complex feedback processes within the atmosphere-cryosphere-ocean system, including surface albedo and heat exchange with the ocean (Johannessen et al., 2003), and has led to substantial losses of sea-ice volume and late-spring snow cover (Overland, 2014).

Increasing temperatures due to global warming lead to an intensified hydrological cycle (Huntington, 2006). Increasing precipitation in the Arctic has been linked to higher local evaporation and reduced Arctic sea ice coverage (Bintanja and Selten, 2014; Kopec et al., 2016), but also enhanced transport of subtropical moisture into the Arctic (Zhang et al., 2013). According to most climate models (see section 2), precipitation will increase in the coming century, with the largest changes occurring over the Arctic Ocean (Bintanja and Selten, 2014). However, there are still large uncertainties regarding

hydroclimate variability and changes in the hydrological cycle in the Arctic due to the incompleteness of data (Serreze et al., 2000; Screen and Simmonds, 2012). This makes it difficult to detect changes of, as well as understand the mechanisms behind, hydroclimate variability on different timescales. Such information is needed to predict accurately the potential impacts of changing hydrologic regimes.

There are large spatial differences in the meteorological station density across the Arctic, where few observational records outside Fennoscandia and westernmost Russia reach more than 75 years back in time (Bekryaev et al., 2010). This makes it difficult to provide spatiotemporal understanding of hydroclimate variability for the whole region. Going beyond the observational records, climate proxies are needed. Most reconstructions of past climate for the whole Arctic within the Common Era (CE) have focused on temperature (Overpeck et al., 1997; Kaufman et al., 2009; Hanhijärvi et al., 2013;

McKay and Kaufman, 2014). However, there are a number of proxies recorded in a few archives, such as ice cores, lake and peat sediments and tree rings, which can provide information on hydroclimate related parameters in the Arctic. They provide information with different temporal and seasonal resolution. In a recent study of hydroclimate variability across the Northern



Hemisphere during the last 1200 years, (Ljungqvist et al., 2016) found a tendency for generally wetter conditions during the 9th–11th centuries, corresponding to the Medieval Climate Anomaly (MCA, *ca.* 900-1150 CE), whereas the 12th–19th centuries, including the Little Ice Age (*ca.* 1400-1850 CE) showed more widespread dry condtions. However, for the Arctic region, only 18 records with heterogeneous spatial distribution were included. It impossible to meaningfully infer past hydroclimate for the whole Arctic from such a limited proxy data set. Nevertheless, ongoing efforts in collecting new data, as well as developing new methods, result in a growing network that will increase our understanding of Arctic hydroclimate variability.

The aim of this paper is to summarise the current understanding of Arctic hydroclimate in a palaeoclimate context, focusing on the last two millennia. The definition of the Arctic used here is the region north of 60°N. The paper is organized as follows: in section 2, we briefly present the current state and a future outlook of Arctic hydroclimate and impacts, from observations and climate model simulations from the Coupled Model Intercomparison Project Phase 5 (CMIP5, Taylor et al., 2012). In section 3, we present the various archives and proxies widely used to derive information on past hydroclimate variability for the region. Two multiproxy comparisons of hydroclimate variability in Arctic Canada and Fennoscandia, two relatively well-replicated regions, are presented in section 4. In section 5, we present a compliation of hydroclimate data from the region, which is also compared to a previous reconstruction and model simulations from the Paleoclimate Modelling Intercomparison Project Phase III (PMIP3, Braconnot et al., 2012). We summarise the current understanding of Arctic hydroclimate during the last 2k in section 6, and provide some recommendations for future work in section 7.

## 2. Current and future Arctic hydroclimate and its impacts

### 2.1. Observations and models

Precipitation data derived from the gridded ERA-20C dataset (Poli et al., 2013), averaged over the whole Arctic ($\geq$ 60°N), show a positive trend over the last century, with a notable increase in the last few decades (Fig. 1). This is in line with previous findings (Serreze et al., 2000; Min et al., 2008). A similar trend is seen in an ensemble of 12 historical CMIP5 simulations (1900-2005, see Table S1 for information on the included models). The spatial patterns of precipitation change are heterogeneous across the region (Fig. 2). The largest increases in annual precipitation occurred over the North Atlantic Region and Barents Sea and, to a lesser degree, over eastern Asia and western North America and the North Pacific (Fig. 2a). Annual precipitation decreased in western Eurasia and locally over eastern Greenland and Svalbard. The CMIP5 models show a similar pattern, although the increase is much lower and more homogenous in space (Fig. 2b). A slightly more prominent increase is seen over parts of the North Atlantic region and the Barents Sea, and decreases in areas southwest and southeast of Greenland. However, the model ensemble does not provide distinct regional differences.



For future scenarios, 36 Representative Concentration Pathways (RCP) CMIP5 simulations (12 for each of RCP2.6, RCP4.5 and RCP8.5 scenarios) for the period 2006-2100 were used. Multi-model ensemble means represent robust projections of the temporal variation, spatial patterns and seasonal cycle of the historical and future annual precipitation variability over the Arctic region. All RCPs indicate an increase in precipitation in the coming century, ranging from 4 mm/month (RCP2.6

ensemble mean) to 14 mm/month (RCP8.5 ensemble mean) (Fig. 3). To obtain the spatial pattern of annual precipitation changes, first the spatial pattern changes were calculated based on individual model simulations, and then re-gridded to the same spatial resolution as the GFDL-CM3 model (i.e. 144 longitudinal grid cells × 90 latitudinal grid cells). The re-gridded spatial pattern changes based on the individual model simulations were then averaged to create a multi-model ensemble mean.

The model ensembles indicate that the most prominent increases in annual precipitation will occur over the Barents Sea, Western Scandinavia, eastern Eurasia and western North America, with decreased precipitation over the central parts of the North Atlantic (Fig. 4). Moreover, the model simulations suggest an intensified precipitation cycle, with increases in all months, but more prominently outside late spring/early summer (Fig. 5).

### 2.2 Impacts of Arctic hydroclimate change

**2.2.1. Impacts on Arctic environments**

Changes in hydroclimate will have impacts on Arctic terrestrial and marine environments, including the cryosphere and the Arctic Ocean (ACIA, 2005). Observational studies show evidence of increased precipitation and river discharge in the Arctic, and hence a freshening of the Arctic Ocean, over the last decades (e.g. Peterson et al., 2006; Min et al., 2008). The freshening will have impacts on ocean convection in the subarctic seas, influencing the thermohaline circulation (THC, see

below) (Min et al., 2008). Increased ocean freshening will also have pronounced implications for marine flora and fauna distribution, due to altered light and nutrient conditions (Carmack et al., 2016). Especially primary producing plankton communities are likely to be affected, some positively and some negatively, which may cascade further up in the food web and alter the whole marine ecosystem structure (Li et al., 2009), affecting marine biodiversity. Overall, changes in landscape and biophysical properties, biogeochemical cycling and chemical transport associated with warmer and wetter

conditions will influence ecosystem productivity (e.g. Wrona et al., 2016). Impacts on ecosystems will also affect Arctic's Indigenous populations, e.g. by increased risks to infrastructure and water resource planning (Bring et al., 2016), health (Geer et al., 2008) as well as their subsistence based livelihoods (Ford et al., 2014). As an example of the latter, increased occurrences of rain events during the cold season, causing the formation of ground ice preventing winter grazing, will have negative impacts on herbivores, such as reindeer (Stien et al., 2012).

**2.2.2. Remote impacts**





In general, snow cover in the pan-Arctic region (including snow cover on sea ice in the Arctic Ocean) has decreased over the last several decades (Screen and Simmonds, 2012; Shi et al., 2013). This has been attributed to elevated temperatures and increasing rain fraction of precipitation (relative to snow). In addition to local effects as described above, changes in snow cover, especially during autumn-winter, influences the atmospheric circulation, which can yield remote impacts on

hydroclimate on lower latitudes. For instance, Cohen et al. (2012) suggested that a warmer and wetter Arctic atmosphere during autumn, caused by decreasing sea-ice coverage, favours increasing snow cover in the same season, which dynamically forces negative Arctic Oscillation (AO) conditions in the subsequent winter. The negative phase of the AO is associated with a wavier Jetstream, which allows cold Arctic air to penetrate into lower latitudes, occasionally yielding extreme weather events (Overland et al., 2016).

A distinct decline in sea ice extent and thickness has been observed in the past decades (Stroeve et al., 2012). The melting of Arctic sea ice has local influences, but recent research suggest that it may also have remote impacts on the midlatitudes by perturbing local energy fluxes at the surface and modifying the atmospheric and oceanic circulation (e.g. Budikova, 2009; Francis et al., 2009). It has been suggested that variations in Arctic sea ice extent influences the North Atlantic Oscillation

(NAO) (Pedersen et al., 2016), which has a strong influence on precipitation in the North Atlantic region (Hurrell, 1995; Folland et al., 2009). Wu et al. (2013) suggested that winter Arctic sea ice concentration may be a precursor for summer rainfall anomalies over northern Eurasia and Guo et al. (2014) noted a link between spring Arctic sea ice conditions and the summer monsoon circulation over East Asia. On the other hand, there is also likely lower latitude phenomena also influence the Arctic sea ice conditions. For instance, wintertime sea-ice loss in the Arctic seems to be linked to the different phases of

the Pacific Decadal Oscillation (PDO) (Screen and Francis, 2016).

Enhanced precipitation and melting of the cryosphere increases the runoff from the pan-Arctic land areas and lowers the salinity in the Arctic Ocean, and this will likely have significant impacts on a local, and potentially global, scale (Serreze and Barry, 2011; Rhein et al., 2013; Carmack et al., 2016). Since the density of the water in the Arctic Ocean determines the

location of the thermo- and halo-clines, changes in salinity may greatly influence distribution patterns of organisms and biogeochemical properties (Aagaard and Carmack, 1989; Carmack et al., 2016). Moreover, salinity regulates the density of the water in the Arctic Ocean, and through outflow of Arctic water into the North Atlantic it may impact regions beyond the Arctic, e.g. by affecting deep water formation in the Greenland-Norwegian and Labrador Seas and thus the strength of the THC (Aagard and Carmack, 1989; Rahmstorf, 1995; Slater et al., 2007). A disruption of the THC may have global impacts

(Vellinga and Wood, 2002). Density also determines the location of the thermo- and haloclines so that salinity shifts greatly influences distribution patterns of organisms and biogeochemical properties (Aagaard and Carmack, 1989; Carmack et al., 2016).



## 3. Hydroclimate archives and proxies in the Arctic

While most archives and proxies that are widely used elsewhere to infer past climate variability can be found in the Arctic, their use require specific treatment and interpretation. The following section describes and discusses the particularities and the limitations of these in the Arctic.

### 3.1. Lake sediments, varves and biomarkers

### 3.1.1 Arctic lakes specificities

Most lakes in the Arctic appeared just after loal retreats of glaciers and ice caps. Hence, their ages, and the potential lengths of the records they contain, are diverse across the Arctic, from the entire Holocene in Beringia and Scandinavia, to a few hundreds years in Greenland or Iceland. Nevertheless, except for the recent decade, the cryosphere does not seem to have
dramatically changed in surface area over the last 2000 years, making lakes excellent recorders of hydroclimate variability for this period. What makes the lakes different in the Arctic is the very strong seasonality that is reflected in a long, to very long, ice cover period. Ice cover substantially reduces the flux of particles and matters to the bottom of the lake to amounts that are frequently not measurable, with the exception of some organic components. Therefore, what is recorded in Arctic lacustrine sediments is strongly biased towards the ice-free periods, i.e. spring snowmelt, short summer, and early fall.
Another charateristic of Arctic lakes is physical weathering related to gelifraction and the lack of vegetation making large quantities of minerogenic matter available that is easily eroded and transported into lakes (Zolitschka et al., 2015).

Lake systems in the Arctic can also be very different depending on the presence or absence of a glacier or an ice cap in their watershed. Glaciarised watersheds are not water-limited, i.e. the water supply to the lake tributary can last the entire summer
and autumn until temperatures drop below zero. Discharge (and sediment transport) is usually driven by temperature at the elevation of the glacier and usually at a maximum during summer. In some cases, the lake systems are strongly influenced by variations of the proximity and volume of the glaciers that regulate sediment fluxes to the lake (see section 3.4.1), or in some rare instances, catastrophic floods (called Jökulhlaups) which are due to collapsing ice dams retaining vast amounts of water in intra- or supra-glacial lakes. On the other end, snowfed watersheds experience maximum discharge during spring snow
melt. They become depleted in water once the snow cover has melted away, making sediment transport to lakes less efficient in the later part of the ice-free season.

Many glaciarised and snowfed watersheds in the Artic are also affected by the presence of permafrost. During summer, the permafrost melts in its upper part (i.e., the active layer), leaving sediment unprotected and easily mobilised by the tiniest
amount of liquid precipitation. This increases the risk of slope detachments, and can result in debris flows or very high sediment yields in lake tributaries (Lewis et al., 2005). The presence of permafrost also makes the dating of lacustrine



sediments difficult because organic matter can be stored in the watersheds soils for a long period prior to be included in the sedimentary processes (Abbott and Stafford, 1996).

In the High Arctic, sources of organic matter in lake sediments are both allochthonous and autochthonous, i.e. produced in the watershed or the lake. The relative contribution of these sources may, in part, be controlled by climate (Outridge et al., 2017), although the allochtonous organic mattter remains dominant, and the total amount preserved remains low (Abbott and Stafford, 1996; Gälman et al., 2008). On the other end, lakes located in the southernmost part of the Arctic, such as in the Boreal forest of Scandinavia or North America, experience a season with higher primary productivity. Their total organic carbon content can be relatively high (Gälman et al., 2008) when anoxic conditions at the bottom of the water column prevail, slowing down its degradation.

### 3.1.2 Extracting hydroclimatic information from Arctic lakes

Most of the proxies used elsewhere in the world in the purpose of reconstructing past hydroclimate can be also analysed from Arctic lakes. Extensive experience has enabled their use in the Arctic in spite of the nature of the environment.

*Pollen* can successfully be used to reconstruct precipitation because the response of plants to moisture changes is direct and well-studied. Although a proportion of the pollen in the High Arctic, significant in some cases, arrives from forested regions to the south, pollen assemblages can still be used to reconstruct the local conditions (e.g. Gajewski, 2002; 2006; 2015b). The pollen in the sediments may be contaminated by older pollen stored in the soils or, in some cases, from Tertiary deposits in the watershed (Gajewski et al., 1995). Nevertheless, annual precipitation has, along with temperature, been reconstructed using pollen assemblages (Gajewski, 2015a), and are presented in section 4.2.

*Chironomids:* one of the primary factors affecting chironomids in the Arctic is lake depth, as is temperature and water chemistry (Gajewski et al., 2005). To the extent that changes in precipitation regime affect the depth of a lake, and the pH and nutrient supply, these can be used (Medeiros et al., 2015). However, most work has emphasized the reconstruction of temperature, and probably there would need to be very large changes in depth to have a noticeable effect on the chironomid community (e.g., Barley et al., 2006; Fortin et al., 2015).

*Diatoms:* Diatoms can presumably be used to reconstruct past moisture through various indirect methods. The primary control on diatoms is alkalinity or pH (Finkelstein et al., 2014), and to the extent this is affected by lake level variability, it could be used as an indirect proxy. Lake level changes affecting the relative area of deep and shallow water (littoral zone reconstructions) can be registered by diatoms; these have been used in the South but not in the Arctic. However, new techniques are arising and pave the way new palaeohydroclimatic reconstruction. The first study stable oxygen isotope signals preserved in diatom frustules allowed for a palaeohydrological reconstruction of Nettilling Lake, Baffin Island,



Canada (Chapligin et al., 2016). High-resolution isotope analysis combined with the study of the specific composition of diatom assemblages were an important addition to palaeoenvironmental.

*Hydrogen isotopes and biomarkers:* the source of environmental water in terrestrial systems is precipitation. Precipitation hydrogen isotope (δD) values are influenced by the location, temperature, and relative humidity of the primary evaporation source of the moisture, the air mass trajectory, and the condensation temperature of precipitation (Dansgaard, 1964; Boyle, 1997; Pierrehumbert, 1999; Masson-Delmotte et al., 2008; Frankenberg et al., 2009; Theaakstone, 2011; Sjolte et al., 2014). Evaporative enrichment can cause environmental water, including lake water, soil moisture, and leaf water, to become D-enriched relative to the original precipitation. Thus, sediment-based lipid δD records can provide important insights into variability of both precipitation δD values and evaporative enrichment, and ultimately to local hydrological changes. To date, there are only a handful of published studies using δD values of leaf waxes and algal lipids to reconstruct past hydrological changes in the Arctic (Thomas et al., 2012, 2016; Balascio et al., 2013, 2017; Moosen et al., 2015; Keisling et al., 2017). It is notable that the paleohydrological interpretations based upon these δD records differ among the studies. This reflects the fact that different lake catchments respond differently to hydrologic changes (and over different time scales), but also highlights our incomplete understanding of the biological and environmental factors that influence hydrogen isotope variability in lipids. Paleohydrologic interpretations are better constrained when lipids with δD values representing lake water (e.g., those derived from algae and macrophytes) are considered together with those representing leaf water (e.g., long-chain *n*-alkanes and long-chain *n*-alkanoic acids) (Balascio et al., 2013, 2017; Rach et al., 2014; Muschitiello et al., 2015; Thomas et al., 2016). Together, δD values of these compounds can be used to quantify isotopic differences between lake water and leaf water, which can reveal changes in the duration of summer ice cover (Balascio et al., 2013), seasonality of precipitation (Thomas et al., 2016), or the vegetation type contributing lipids to the lake sediments (Balascio et al., 2017). Rach et al. (2017) propose an approach using paired terrestrial and aquatic lipid δD values and plant physiological models to quantitatively reconstruct relative humidity changes through time. This approach may prove effective in some Arctic settings.

Several physical and geochemical proxies have been used to infer past hydrology: *Mass accumulation rate* is a measure of the amount of sediment accumulated at the bottom of the lake (e.g. Weltje, 2012). It is usually directly linked to the lake tributary discharge in lakes with low primary productivity. Obtaining MAR requires an accurate age model and measurements of density (Petterson et al., 1999). *Density*, *magnetic susceptibility* and *elemental composition* are all indicators of the detrital input, which is again linked to the lake tributary discharge (Petterson et al., 1999; Dearing et al., 2001, Cuven et al., 2010). *Grain-size* of the terrigenous fraction is an indicator of the competence of the flow (maximum discharge), its duration, and physical processes occuring in the lake water column (Lapointe et al., 2012). Altogether, these physical and geochemical proxies are rarely used in Arctic sedimentary sequences with massive structure because of the complexity of their interpretation, however, they proved to be powerful tools in annually laminated sediments.



### 3.1.3 Varved sediments

Varved sediments are difficult to find and probably rarely deposited (Zolitschka et al., 2015). However, several lakes with varved sediments have been found in the Arctic, most probably because the very strong seasonal contrast in sediment supply favors the formation of varves. Lakes containing varves tend to be deep enough to prevent bioturbation and are usually found
in watersheds with high sediment yield. As such, many of the varved records cannot be directly compared to lakes used in diatom and pollen studies because the latter are usually studied in smaller systems. The advantages of varved sediments are that they contain their own internal chronology, that annual fluxes can be measured through the measurement of density, and that their properties can be calibrated against instrumental records (Hardy et al., 1996). In the Arctic, two types of varves exists: clastic varves and mixed clastic-biogenic varves, discussed in Zolitshcka et al., (2015).

### 3.1.3.1. Clastic varves

Clastic varves result from the complex interactions between sediment availability (geomorphological control), seasonal run-off variations carying suspended sediment (hydroclimatic control), the thermal density structure of the lake water column and the bathymetry of the lake (limnological control). These varves are typically composed of a coarse-grained lower lamina that grades into a fine-grained upper lamina (e.g., Lake DV09; Courtney-Mustaphi and Gajewski, 2013; Lake C2;
Zolitschka, 1996). Additional coarse grained laminae can be deposited and can be related to multiple pulses of snow melt or rain events (Ringberg and Erlstöm, 1999; Cockburn and Lamoureux, 2008). The finest clay fraction remains in the water column and is only deposited under quit condition during the following winter (Francus et al., 2008). Therefore, the presence of a distinct clay cap is the main criteria for identifying a year of sedimentation (Zolitschka et al., 2015).

Several individual parameters can be measured from each varve sequence: total thickness, sublamina thickness, density, mass accumulation rate, total and sublamina grain-size, elemental composition, and magnetic susceptibility. Linking these properties with hydroclimate conditions requires a minimal understanding (i.e. monitoring) of the processes occuring in the watershed and the lake, each system being different. Disantlangling the respective effect of the temperature from moisture is a challenge because due in part to the difficulty in obtaining data for calibration in the Arctic. When comparing varves
properties to observational climate data, they often contain signals of both temperature and precipitation (e.g. Table 2 of Cuven et al., 2011; Lamoureux and Gilbert, 2004), although the temperature signal has been more often reported in the literature. However, this may be due to the fact that more robust measurements of instrumental temperature are available compared to precipitation (especially snow), and that precipitation patterns tends to be more variable over a region, making correlation with sediment properties more problematic.

Despite these difficulties, several authors reported correlations with hydroclimate. In general, the hydroclimate is revealed in the measurement of a specific part of the sedimentary cycle, and not by a parameter that integrates the whole year of



sedimentation such as the total varve thickness. For instance, Lapointe et al. (2012) showed a correlation (r = 0.85, p = 0.0001) between the largest rainfall events and the coarsest grain-size fraction of each varves. Lamoureux et al. (2006) found a correlation between varve thickness of Sanagak Lake, Boothia Peninsula and snowwater equivalent in the watershed but they were unable to calibrate the series due to lack of calibration data. Francus et al. (2002) found a correlation (r = 0.53, p < 0.05) between snowmelt intensity and the median grain-size. Lamoureux (2000) found an association of sediment yield estimates of Nicolay Lake, Cornwallis Island, and rainfall events.

### 3.1.3.2 Mixed (clastic-biogenic) varves

In less harsh environments, such as southern Scandinavia, the vegetation in the catchment area and soils are more developed, allowing for decaying organic matter to be incorporated into the lacustrine system. In the same time, the primary productivity in the water column during the warmer seasons is important enough to be recorded in the sedimentary archive. Yet, it results in the accumulation of a mixed varves type, known widely as clastic-biogenic (or clastic-organic), that typically contains a characteristic minerogenic lamina, usually showing graded bedding and that is directly related to the duration and strength of the spring flood (e.g. Ojala et al., 2000; Snowball et al., 2002, Tiljander et al., 2003), and a biogenic lamina that can be composed of autochthonous organic matter (e.g. diatoms frustules) and/or allochtonous organic debris.

Proxies measured with annual resolution on these mixed varved are 1) total varve thickness, 2) growing season lamina (GSL) thickness, 3) winter lamina (WL) thickness (Saarni et al., 2015), and 4) relative X-Ray densitometry (Ojala and Francus, 2002). Correlations with climate parameters vary from sites to sites and sometimes throught times in a single site (Saarni et al., 2015). Only a handfull number of lacustrine sequences, all of them from Scandinavia, have been sucessfully correlated to precipitation or moisture. At Lake Nautajärvi annual and winter precipitation was reconstructed using relative X-Ray densitometry (Ojala and Alenius, 2005), while at Lake Kallio-Kourujärvi, the growing season lamina was linked to annual precipitation (Saarni et al., 2015). Rydberg and Martinez-Cortizas (2014) showed that high accumulation of snow resulted in high mineral matter content, and Wohlfarth et al. (1998) found a significant correlation between early spring/summer precipitation with total varve thickness in north-central Sweden.

As with clastic varves, it is quite difficult to disantlangle the temperature from the moisture signal. Ojala and Alenius (2005) showed that the direct annual and seasonal comparisons between raw varve data and instrumental measurements are complicated. Itkonen and Salonen (1994) showed that total varve thickness of three Finnish lakes were corrrelated with both temperature and precipitation, correlation being weaker for precipitation. Nevertheless, sediment trap studies clearly but qualitatively showed the sensitivity of such systems to varying hydroclimate conditions (Ojala et al. 2013; Rydberg and Martinez-Cortizas 2014).



## 3.2 Peat deposits

### 3.2.1. Peatland processes and peat archive

Peatlands are wetland ecosystems that preserve their developmental history over millennia. Peat deposits are products of the balance between plant production and organic-matter decomposition (Clymo, 1984)–both processes are affected by climate

change. As a result, peat accumulation and processes are inherently influenced by autogenic/ecological and allogenic/climatic factors, and their interactions (Belyea and Baird, 2006). Many peat-based proxies (see below) have been used to reconstruct peatland hydrology and water-table dynamics, likely connected with regional hydroclimate. This inherent, dynamic change ability of wetland communities results largely from their occurrence in environments where a single extremely variable habitat factor, i.e. water supply, is predominant (Tallis, 1983). However, increasing empirical and

modelling studies show the importance of autogenic process and ecohydrological feedbacks (e.g., Swindles et al., 2012; Loisel and Yu, 2013). We are mindful that the consideration of biological processes and ecological feedbacks are needed when using these living systems for climate reconstructions.

Peatland plants shape their own habitat since they form their own growth substrate: peat. Hence, peatlands are capable of recording in their deposits the effects of past vegetational and ecological changes. Within the peat lies a repository of

botanical, zoological, environmental and biogeochemical information, which is extremely important for understanding past climatic conditions. These palaeo records are used to estimate the rates of peat formation or degradation, past vegetation, climatic conditions and depositional environments (Moore and Shearer, 1997; Blackford, 2000). Analysis of peat deposits has undergone major developments during the last several decades regarding coring techniques, peat sampling and analysis, geochronology, identification of plant remains and other microfossils, and quantitative multivariate techniques. Refined

methods of peat humification and quantitative macrofossil and microfossil analysis have been successfully used to reconstruct past climatic and environmental conditions (e.g. Barber et al., 1994; 1998; Charman et al., 2009; Chambers et al., 2011).

Stratigraphic studies in peatlands have shown a hydroseral succession, where wet swamp and fen communities gradually develop into dry bog communities (Tallis, 1983; Korhola, 1992). These changes are largely autogenic, connected to growth

of wetland communities, and caused by past climatic variability or artificial drainage. Hilbert et al., (2000) developed a model of peatland growth that explicitly incorporates hydrology and feedbacks between moisture storage and peatland production and decomposition. They suggest that drier ombrotrophic peatlands (most bogs) will adjust relatively quickly to perturbations in moisture storage, while wetter ombrotrophic peatlands (mineral rich fens) are relatively unstable and can withstand only a very small range change in water tables. Climate change will affect the hydrology of individual peatland

ecosystems mainly through changes in precipitation and temperature. As the hydrology of the surface layer of a bog is dependent on atmospheric inputs (Ingram, 1983), changes in the ratio of precipitation to evapotranspiration may be expected to be the main factor in driving ecosystem change. In particular, ombrotrophic peatlands are regarded as directly coupled to

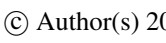



the atmosphere through precipitation and hydrology change (Barber et al., 1994), such that their water levels and dominant plants will reflect the prevailing climate. More specifically, their water-table variability has been shown to be highly correlated with the total summer seasonal moisture deficit (that is, precipitation – evapotranspiration; Charman, 2007).

Modern investigations of past climate are performed with an emphasis on obtaining the highest possible time resolution for a
given archive. Radiocarbon dating is one of the main methods used to establish peat chronologies. The best material for ensuring accurate dates are aboveground remains of plants that assimilated atmospheric $CO_2$, e.g. short-lived plant macrofossils and pollen, whose $^{14}C$ age is consequently not affected by a old carbon effect. Suitable materials for sample selection are *Sphagnum* mosses (branches, stems and leaves) or, if not present, aboveground leaves and stems of dwarf shrubs (e.g. Nilsson et al., 2001). Age-depth models, that are considered ecologically plausible and that take into account
likely modes of peat accumulation, include: (1) linear accumulation; (2) concave curves (through continuing decomposition of fossil matter in the peat deposit; Yu et al., 2001); (3) convex curves (with deposits slowing down their accumulation when close to a height limit; Belyea and Baird, 2006); and (4) Bayesian models that can include prior information on stratigraphy, accumulation rate and variability, and/or detect outlying dates (reviewed by Parnell et al., 2011). The robustness of age models can be significantly improved, and the uncertainties reduced, by using multiple dating methods on a single core.
Most commonly, the uppermost layer can be dated using atmospheric fall-out radionuclides (e.g. $^{210}Pb$; see Le Roux and Marshall, 2011) and spheroidal carbonaceous particles (SCPs) profiles (Yang et al., 2001), while tephrostratigraphy can potentially be applied throughout the core (Swindles et al., 2010). With suitable statistical treatment, all results can be combined into one reliable chronology which provides the backbone for interpretations of palaeoclimatic and palaeoenvironmental change data.

**3.2.2. Peat-based hydroclimate proxies**

Peatland formation can initiate via three processes: primary peatland formation, terrestrialization or paludification (Rydin and Jeglum, 2006). In primary peatland formation, peat is formed directly on wet mineral soil when the land is newly exposed due to crustal uplift or deglaciation, whereas in terrestrialization and paludification the area colonized by peatland vegetation has experienced previous sediment deposition or soil development (e.g. Tuittila et al., 2013). Potentially, much
information on hydroclimatic conditions can be derived from these processes, especially when the different peat formation types show systematic and isochronic patterns over wide geographic areas. For example, in paludification, the prerequisite is that the local hydrological conditions become wetter, for instance, induced by climatic change, fire or beaver damming, resulting in waterlogged soil conditions that promote peat accumulation (Charman, 2002; Gorham et al., 2007; Rydin and Jeglum, 2006). Furthermore, a new conceptual model of episodic, drought-triggered terrestrialization presents the infilling as
an allogenic process driven by decadal-to-multi-decadal hydroclimatic variability (Ireland et al., 2012).





Recently, Ruppel et al. (2013) presented a comprehensive account of postglacial peatland formation histories in North America and northern Europe using a data set of 1400 basal peat ages accompanied by below-peat sediment-type interpretations. Their data, mainly focusing on Boreal-Arctic regions, indicated that that the peat formation processes exhibited some clear spatiotemporal patterns. Unfortunately, the overwhelming majority of the basal peat accounts originate

from the deepest, and often the oldest parts of peatlands, for which reason the last two millennia are clearly underrepresented in the present data. However, the existing studies clearly emphasises the potential of using peat initiation and expansion data for accounting for changes in regional moisture regimes also for the more recent times. It should further be noted that the formation of new peatland areas does not necessarily decrease when the initiation rates decrease but that new peatland areas are continuously formed via lateral expansion.

As one of the peat physical properties, bulk density (or organic matter density) down-core profiles have been used as reflecting overall peat decomposition – which in many peatland regions is controlled by surface moisture and hydroclimate conditions (e.g. Yu et al., 2003). The rationale is that well-preserved peat is loose and has low organic matter density, most likely deposited under wet conditions promoting protection of organic matter in an anaerobic environment. Peat bulk density values are typically 0.05 to 0.2 g cm$^{-3}$ in high-latitude regions (Chamber et al., 2011). Peat humification is another proxy for

the degree of peat decomposition. It can be estimated or measured in the field or laboratory using a range of methods (Chamber et al., 2011). Humification can be used as a proxy of peatland surface wetness – as moisture is a key determinant of decomposition – and regional hydroclimate in the Arctic (e.g. Borgmark, 2005; Vorren et al., 2012).

Net carbon accumulation is the balance between production and decomposition, both which are influenced directly or indirectly by climate (Yu et al., 2009). However, recent syntheses indicate that temperature-driven production might be more

important than moisture-controlled decomposition in determining net peat peat accumulation (e.g. Beilman et al., 2009; Charman et al., 2013). Therefore, without constraints from other proxies, it is difficult to infer hydroclimate from peat accumulation records. As mosses are dominated plants in peatlands, carbon isotopes from these mosses and peat are useful for inferring peatland moisture conditions. In wet conditions, water films around moss leaves will reduce conductance of pores on leaf surface to $CO_2$ uptake, reducing discrimination against $^{13}C$ and resulting in high carbon isotope values (Rice,

2000). Carbon isotopes have shown to reflect surface moisture in peatlands (e.g. Loisel et al., 2009). In addition, Nichols et al. (2009) used compound-specific carbon and hydrogen isotopes from peatlands in the Arctic to evaluate summer surface wetness and precipitation seasonality.

Because plant macrofossils reflect changing abundances of climatically sensitive peatland vegetation, they have been used not only for reconstructing the local vegetation history of peatlands but also for inferring past peatland hydrological changes

and, by extension, regional climate variability (e.g. Barber et al., 1998; Hughes et al., 2000; Swindles et al., 2007; Mauquoy et al., 2008). Traditionally, plant-based peatland surface wetness reconstructions have been qualitative or semi-quantitative from the identification of phases of relatively low local water tables (showing increased representation of hummock species)



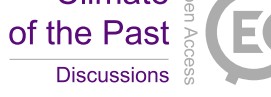

and phases of higher local water-table depths (lawn and hollow species) (Mauquoy et al., 2002; Pancost et al., 2002; Sillasoo et al., 2007). More recently, ordination techniques (e.g. PCA and DCA) have been used to create a single index of peatland surface wetness based on the total sub-fossil dataset for a peat profile (Barber et al., 1994; Mauquoy et al., 2004; Sillasoo et al., 2007), in which it is assumed that the principal axis of variability in the dataset is linked to hydrology. The most recent

progress in identification and quantification techniques of plant macrofossils (e.g. the Quadrat Leaf Count method, see Mauquoy et al., 2010 for detailed description), together with careful calibration with modern plant community data, allow quantification of past peatland water table fluctuations with great accuracy. Väliranta et al. (2007) developed a transfer function by calibrating plant macrofossil records against the modern vegetation/water-table relationship, in order to quantitatively reconstruct peatland surface wetness trends for the late Holocene. The inferred water tables showed strong

fluctuations, with an overall amplitude of ca. 40 cm. During the last two millennia, they found a significant increasing trend in summer dryness from ca. 3.5 to 1.5 cal kyr BP, with the last 1500 years becoming wetter in summers. Superimposed on this long-term trend, there are two periods showing summer dryness from 1.5 to 1.0 (ca. 500 - 1000 CE), and from 0.7 to 0.1 cal kyr BP (ca. 1300 to 1900 CE), and periods with wet summer from 2.0 to 1.5, 0.9 to 0.3 cal kyr BP (ca. 1100 to 1700 CE), and from 1900 to the present. Furthermore, the comparison of water-table reconstructions based on macrofossils and testate

amoebae at two different bogs in Estonia and Finland yielded further confidence for using bog plants in quantitative hydrological reconstructions (Väliranta et al., 2011).

Testate amoebae (Protozoa: Rhizopoda) are unicellular animals with distinct environmental preferences, which live in abundance on the surface of most peat bogs. These amoeboid protozoans produce morphologically distinct shells, which are commonly used as surface moisture proxies in peat-based paleoclimate studies (Mitchell et al., 2008). Although the moisture

sensitivity of these organisms has been known for a long time, work over the past several decades has demonstrated the utility of testate amoebae as quantitative peatland surface-moisture indicators. Their indicator value in documenting surface-moisture variation has been demonstrated by coherence in reconstructions of wet and dry fluctuations within and between peatland sites (Hendon et al., 2001; Booth et al., 2006). A protocol of their use in paleohydrological studies is provided by Charman et al. (2000) and by Booth et al. (2010). Testate amoebae have been used for tracing hydrological changes in

temperate peatlands in several regions of the world, as well as in boreal and subarctic peatlands of Canada and the US (Payne et al., 2006; Loisel and Garneau, 2010; van Bellen et al., 2011; Bunbury et al., 2012; Lamarre et al., 2012; 2013). In addition to bogs, their applicability also applies to fens (Payne, 2011). Recently, Swindles et al. (2015) tested the potential of testate amoebae for peatland paleohydrological reconstruction in permafrost peatlands, based on sites in Arctic Sweden (~200 km north of the Arctic Circle). Their evaluation confirmed that water-table depth and moisture content are the

dominant controls on the distribution of testate amoebae also in Arctic peatlands, corroborating the results from studies in mid-latitude regions. They created a new testate amoeba-based water table transfer function with a good predictive power ($R^2$ = 0.87, RMSEP = 5.25 cm). The new transfer function was applied to a short core from permafrost peatland, and revealed a major shift in peatland hydrology coincident with the onset of the Little Ice Age (ca. 1400 CE). The new transfer



function will enable paleohydrological reconstruction from permafrost peatlands in Northern Europe, thereby permitting greatly improved understanding of the long-term hydrological dynamics of these ecosystems as well as the general variability in hydroclimatic conditions.

### 3.3. Tree-ring data

5 Distinct, precisely dateable tree-rings are generally formed in areas with pronounced seasonality, which results in a single period of cambial activity (growth) and dormancy per calendar year. The width, density and isotopic compositions of a tree-ring are partly determined by local weather and climate, and, in this case, the closer to the ecological limit of distribution a tree grows, the more sensitive to climate it will be. Due to the large spatial distribution of dendrochronologically dateable trees across extratropical regions, their capacity to live for many years and their potential for developing precise, annually 10 resolved chronologies, tree-ring data have been widely used to infer late Holocene variations in a range of climate parameters on local to hemispheric scales.

### 3.3.1. Tree-ring width and density

Measurements of annual tree-ring widths (TRW) are perhaps the most important data source for quantitative estimates of high- to low-frequency climate variability during the past centuries to millennia. The advantage of TRW comes from its 15 annual resolution and a comprehensible understanding of the climatic controls on the tree-ring growth dynamics (e.g. Vaganov et al., 2006). Tree rings have the advantage of numerical calibration, verification and potential to capture seasonal extreme events not possible using lower-resolution, less temporally well-constrained archives. The tree-ring community has generated an expansive network of TRW chronologies, covering a wide range of species and ecosystems, across the globe, not least in the Boreal-Arctic ecotones. In general, trees growing close to their latitudinal or altitudinal limit of distribution 20 will be sensitive to warm-season temperature, while trees growing in semi-arid to arid regions are limited by precipitation/moisture (St George, 2014; St George and Ault, 2014; Hellman et al., 2016). Consequently, most high-latitude TRW chronologies will exhibit strong positive associations with summer temperature and no or weak correlations with summer or winter rainfall. Indeed, tree-ring data from the high northern latitudes have been used in several reconstructions of Northern Hemisphere temperature (e.g. D'Arrigo and Jacoby, 1993, Jones et al., 1998; Briffa et al., 2001; D'Arrigo et al., 25 2006; Wilson et al., 2016), as well as reconstructions targeting Arctic temperatures (Overpeck et al., 1997; Kaufman et al., 2009; Hanhijärvi et al., 2013; McKay and Kaufman, 2014). The few chronologies in the cool boreal and Arctic regions developed from precipitation sensitive trees are mainly located in continental climate zones, such as western Canada, Alaska and eastern Fennoscandia (Fig. 6). Importantly, many TRW records are negatively correlated with summer rainfall, and most of these are found in the colder high-latitude regions. Compared to the association between TRW and rainfall during the 30 current growing season, positive correlations with prior-summer precipitation are also common across the Arctic TRW network. This carry-over effect may be caused by increased photosynthetic reserve accumulation in years with sufficient




moisture supplying resources that can be used for secondary tissue growth in subsequent years. It is also not unlikely that a proportion of this association also reflects the inverse relationship between summer temperature and precipitation observed in these regions.

Although TRW is the most commonly used tree-ring proxy, at high latitudes, wood densitometric measurements, specifically
maximum latewood density (MXD) and its surrogate Blue Intensity (BI), are commonly being viewed as superior temperature proxies compared to TRW. It would seem that the strong correlation between MXD/BI and temperature would prevent their use in hydroclimate reconstructions. However, recent studies (Cook et al., 2015; Seftigen et al., 2015a; 2015b) have indirectly used high-latitude temperature sensitive tree-ring data to reconstruct soil moisture availability, by considering the inverse relationship between available soil moisture and clear skies, higher temperatures, increased evaporation, and
reduced rainfall. Thus, the negative correlation between the high-latitude tree-ring data and drought metrics, such as the self-calibrating Palmer Drought Severity Index (scPDSI, van der Schrier et al., 2006a, b) and the Standardised Precipitation-Evapotranspiration Index (SPEI, Vicente-Serrano et al., 2010), can be used to generate reconstructions that are comparable to those from arid and semi-arid regions where tree growth is strongly limited by rainfall. Many high-latitude MXD data that are mainly influenced by temperature also exhibit a negative, albeit weak, statistical association to summer precipitation
(Briffa et al., 2002). This mixed response explains why such data have successfully been used in reconstructions of drought indices which integrate both temperature and precipitation. However, where temperature sensitive MXD (or TRW) data are used to develop reconstructions of hydroclimate, they should be interpreted with extreme care and never to derive a paired temperature-hydroclimate climate history where the reconstructions of temperature used for comparison with hydroclimate are developed using the same (or very similar) temperature-sensitive proxies.

**3.3.2. Stable isotopes in tree rings**

The isotopic ratios of wood, lignin and tree-ring cellulose are influenced by a different and more limited range of environmental and physiological controls than TRW and MXD. For this reason, stable isotopes in tree-rings provide additional palaeoclimate information to support and enhance the information attainable from the physical proxies (McCarroll and Loader, 2004, Gessler et al., 2014). Similar to the physical proxies of TRW and MXD, the strength and relative
expression of these climatic controls will vary geographically and to a degree with local edaphic conditions and tree species. In simple terms, carbon isotopic variability reflect changes in the balance between conductance of carbon dioxide ($CO_2$) from the atmosphere to the site of photosynthesis, and assimilation rates, which are influenced by moisture stress and photosynthetically active radiation (PAR), respectively. Temperature and nutrients availability may also contribute to this signal through an influence upon the rate of chemical reactions and production of photosynthetic enzymes (Farquhar et al.,
1982, Scheidegger et al., 2000, Hari and Nöjd, 2009). Oxygen and hydrogen isotopes are more closely related to the isotopic composition of the water used by the tree during photosynthesis, which may reflect a combination of moisture sources,



subsequently modified by evaporative enrichment of leaf water (vapour pressure deficit and relative humidity) and plant physiological processes (Barbour et al., 2001; Danis et al., 2006; Treydte et al., 2014; Roden et al., 2000).

Since the earliest isotopic dendroclimatology studies conducted in the Arctic (Sonninen and Jungner, 1995; McCarroll and
Pawellek, 1998; Waterhouse et al., 2000), several isotope studies have made significant contributions to palaeohydrology (e.g. Waterhouse et al., 2000; Holzkämper et al., 2008; 2012; Sidorova et al., 2008; 2009; Porter et al., 2009). The combination of long-lived trees, robust dendrochronologies and excellent sample preservation both on land and in lakes have facilitated the development of several multi-centennial to millennial length isotopic records (Boettger et al., 2003; Kremenetski et al., 2004; Sidorova et al., 2008; Young et al., 2010; Gagen et al., 2011; Loader et al., 2013; Porter et al.,
2014). However, because moisture is rarely the dominant tree-growth limiting factor across much of the Arctic region, there is a limitation of the hydroclimate information that can be reconstructed using the isotopic approach. Using a multi-parameter approach several studies (Loader et al., 2013; Young et al., 2010; 2012; Gagen et al., 2011) provided sunshine/cloud estimates and were able to demonstrate large-scale shifts in the dominance of Arctic and Maritime air masses over the Northern Fennoscandian region during the Little Ice Age and Mediaeval period. Such multi-parameter studies are
potentially very powerful as they help to develop testable hypotheses relating to the future response of the Arctic atmosphere and provide a foundation for developing a circum-polar isotope network to track changes in atmospheric circulation and its relationship to climate throughout the Common Era.

Reconstructions based upon oxygen and hydrogen isotopes are yet to reveal the same clear and stable correlations against
instrumental data observed for carbon, but are likely to relate most closely to local and regional hydroclimate through their close link with stable isotopes in precipitation (Roden et al., 2000), although in the Arctic the relative contributions of the isotopic signal from snowmelt and growing season precipitation used to form the tree rings is an area requiring investigation. Links between $\delta^{18}$O and both moisture and temperature have been identified (Sidorova et al., 2009; Knorre et al., 2010). Further south Hilasvuori and Berninger (2010) linked oxygen (and carbon isotopes) most strongly to cloud cover, with
precipitation, relative humidity and temperature exhibiting lesser correlations. Hydrogen isotopes did not correlate as strongly as oxygen or carbon, the strongest statistically significant relationships being reported with precipitation. Seftigen et al. (2011) linked $\delta^{18}$O to precipitation, but noted that this relationship was unstable through time, suggesting that this may reflect changes in the atmospheric circulation. If the same close relationship observed between water isotope composition and tree ring cellulose in other mid-latitude regions (Danis et al., 2006; Labuhn et al., 2014; Treydte et al., 2014; Young et
al., 2015) is confirmed in the Arctic then exciting potentials will exist for developing long records of the isotopic composition of precipitation suitable for large-scale mapping of isotope climate (Hemming et al., 2007; Saurer et al., 2012; Young et al., 2015) and the use of these data to evaluate the performance of isotope-enabled Earth system models. Reconstructing the stable isotopic composition of precipitation will likely provide a more useful, more direct link to the global hydrological cycle "isotope climate" (Birks and Edwards, 2009; Bowen, 2010), than a statistical calibration of water





isotopes developed against a measured (indirect) meteorological variable which may vary in the degree of its control across space or over time.

### 3.3.3. Tree-ring based hydroclimate reconstructions

Tree-ring data have been used to estimate a variety of local (site specific) hydroclimate variables, such as precipitation,
drought, streamflow, cloud cover and snowpack (e.g. Stahle and Cleaveland, 1988; Waterhouse et al., 2000; Pederson et al., 2001; Meko et al., 2001; Woodhouse, 2003; Gray et al., 2003; Young et al., 2010, Gagen et al., 2011). Moreover, networks of tree-ring chronologies have been used to make spatial (or field) hydroclimate reconstructions (Nicault et al., 2007; Cook et al., 1999; 2004; 2010; Fang et al., 2011; Touchan et al., 2011; Hua et al., 2013). However, these studies have almost exclusively utilized tree-ring data from lower latitudes outside the Arctic region.

Within the Arctic 2k region, trees are naturally constrained to exist below the latitudinal tree line, corresponding to *ca.* 73°N, and as noted above, the majority of the tree-ring data in the region comes from temperature sensitive trees (Fig. 6). Still, with careful site selection, it is possible to find trees that are sensitive to moisture variability, and a few studies have inferred past precipitation variability using ring-width data. Indeed, a handful of reconstructions of local hydroclimate from the Arctic
have been published. These have mainly focused on late spring/early summer precipitation. The longest record, and presently most widely used high-resolution hydroclimate proxy for high-latitude Fennoscandian, comes from south-eastern Finland, where Scots pine TRW was used to reconstruct annual May–June precipitation over the last millennium (Helama and Lindholm, 2003). Later an updated chronology from the same region was used to highlight the distinct and persistent "mega drought" from the early 9th century to the 13th century CE (Helama et al., 2009). The same parameter was also
reconstructed from Scots pine for east-central Sweden back to 1560 by (Jönsson and Nilsson, 2009). In North America, Pisaric et al. (2009) reconstructed June precipitation of the Northwest Territories, Canada using a TRW network. The only reconstruction of hydroclimate outside the growing season was presented by Linderholm and Chen (2005), who developed a 400-year long winter (September-April) precipitation reconstruction with 5-year resolution, based on Scots pine TRW data from west-central Scandinavia.

Focusing on hydroclimate field reconstructions, one of the earliest works to use tree-rings to reconstruct past moisture variability in a high-latitude region was the *North American Drought Atlas* (NADA, Fig. 7a). The atlas was first released in 2004 (Cook et al., 2004), then covering continental U.S., but later updated (Cook et al., 2007) with an expanded tree-ring network to include parts of the Canadian Arctic. Although significant portions of the latter region are at present under-
represented in NADA, the tree-ring coverage still provides informative hydroclimate reconstructions for a number of regions there. The summer PDSI reconstruction data for the Arctic part of NADA extends back to the 1000 CE, indicating slightly dry conditions during most of the MCA, except for a wet period in the 12th century, and a highly variable LIA albeit with a tendency for progressively wetter summers before the early 19th century (Fig. 7c). Turning to the European sector, two



efforts have used extensive tree-ring data networks to infer past drought/pluvial variability for Fennoscandia (Seftigen et al., 2015a, 2015b) and Europe (The *Old World Drought Atlas*, OWDA, Cook et al., 2015). These atlases, where tree-ring data was used to create gridded (field) reconstructions of the SPEI (Seftigen et al., 2015b) and the scPDSI (Cook et al., 2015), included regions above 60°N. The reconstructions cover the last millennium and provide an important basis for studying pre-

industrial hydroclimate variability and its causes and consequences. This allowed for detailed investigations of the MCA, and the LIA. The OWDA (Fig. 7a) provides strong evidence that the MCA in continental Europe and southern Scandinavia was significantly drier that either the LIA or the post-industrial period (1850-present). It is likely that also the Arctic regions in Europe experienced a severe drought during this period (Fig. 7b), which is in agreement with the findings of Helama et al. (2009). Interestingly, the timing of the MCA dryness seems to temporally coincide with multicentennial droughts previously

reported for large areas of North America (Cook et al., 2007), specifically in California and Nevada. This suggests a common forcing across the North Atlantic, likely related to the North Atlantic Oscillation (NAO) and/or Atlantic Ocean sea surface temperatures. However, the restricted temporal coverage of the high-latitude part of NADA does not provide an opportunity to compare hydroclimatic variability across the Arctic region during the MCA. Figure 7 further reveals the existence of large-amplitude, decadal to centennial, European Arctic hydroclimatic variability over the past millennium,

which is not only restricted to the MCA. Pronounced periods dryness are for example recorded in the first half of the 15[th] century CE and in the 1750s-1850s, and may not have been restricted to the Arctic region (Cole and Marsh, 2006).

Another option to derive hydroclimate information from north of the treeline in the Arctic, is the utilisation of annual growth rings from shrubs. For example, Zalatan and Gajewski (2006) presented a short *Salix alexensis* growth-ring series from

northwestern Victoria Island in the Canadian Arctic. The widths of the shrub rings were found to be correlated with winter precipitation. Although the reported record was too short to be useful for palaeoclimate studies, it may be possible to obtain longer series by using larger specimens (some are tree-sized in this area, Edlund and Egginton, 1984) or cross-dating dead and buried wood.

### 4.3.4. Pine regeneration patterns as indicators of hydrological shifts

In the high northern latitudes, tree remains can be preserved for several millennia buried in lakes or peat, so called subfossil wood. Especially trees from lakes have been used to reconstruct temperatures for large parts of the Holocene in Fennoscandia (see Linderholm et al., 2010 for a review). Also, more or less well preserved trees can also be found in dark layers of well humified peat, an indicator of dry conditions conducive to tree colonization and growth in these regions (Gunnarson, 2008).

In west-central Sweden, more than 1000 subfossil and peatland Scots pine (*Pinus sylvestris*) samples have been collected since the late 1990s. Most samples come from different lakes at varying altitudes, and the temporal distributions of the dated samples show wavelike patterns of regeneration with clearly distinguishable mortality and germination phases. Such



generation pulses have been related periods of climatic conditions favourable for seed production and successful germination, i.e. warm and dry periods (Zachrisson et al., 1995). However, Gunnarson (2008) suggested that temporal variation of pine samples from both bog and lakes, shown in figure 8, reflect fluctuations in peatland ground water tables and lake levels caused by regional changes in hydroclimate.  It is likely that these variations were governed by changes in precipitation rather than changes in temperature. In southeastern Finland evidences of depositional histories of subfossil pines from lakes, where most trees have grown adjacent to or on the lake shores (so called riparian trees), and peatland pines were combined by Helama et al. (2017a). Divergent courses of depositional histories (i.e. replication curves) were obtained for the two environments during the Common Era. High accumulation of peatland pines during the MCA indicated dry surface conditions beneficial for pine colonization (Torbenson et al., 2015; Edvardsson et al., 2016). This phase overlapped with a phase of low accumulation (low preservation) of riparian pine trees. In contrast, the accumulation of riparian pines increased towards the LIA, culminating around 1300 CE, suggesting rising lake water level contributing to tree mortality and increased preservation potential of trees in lakes. Again, this phase overlapped with a phase of strongly declining accumulation of peatland pine trees. These results were supported by taphonomical interpretation (Gastaldo, 1988) of the depositional histories, especially their dissimilarities, and by comparisons with palaeolimnological reconstructions of water level fluctuations during the MCA and LIA (Luoto, 2009; Nevalainen et al., 2011; Nevalainen and Luoto, 2012). Similar to the study conducted in west-central Sweden (Gunnarson, 2008), the depositional histories in southeastern Finland were found to reflect past hydroclimatic variations. Likely, the replication curves of pine chronologies from near the northern edge of the species range reflect summer temperature conditions, especially in subarctic sites (Helama et al., 2005; 2010). Further south, the tree accumulation in different sediments seem to be more strongly influenced by recruitment and preservation potentials which, in turn, are driven by local hydroclimatic conditions.

## 3.4. Glaciers

### 3.4.1. Glaciers as direct and indirect climate indicators

Glaciers provide valuable information of past climate variability though variations in length, area and volume (Oerlemans, 1994, 2001). In the Arctic and subarctic, observations and indirect evidences of glacier fluctuations, have been widely used as sources of information about past climates (Solomina et al., 2016 and references therein). Changes in glacier length through advances or retreats are indirect, lagged responses to climate change, while glacier mass balance variations, as indicated by changes in ice thickness and volume, are direct responses to the annual weather conditions (Haeberli and Hoelzle, 1995). Direct measurements of glacier variability across the world, derived from annual mass balance measurements using glaciological or geodetic methods, are generally limited to the last half century (Zemp et al., 2009). In addition, annual mass balance records have been extended for several centuries using meteorological and proxy data such as historical records and tree-ring data (e.g. Lewis and Smith, 2004; Watson and Luckman, 2004; Nordli et al., 2005;



Linderholm and Jansson, 2007). However, to yield information about glacier variability beyond the direct observation, indirect indicators are mainly used.

There are two types of glacier records: classical discontinuous series usually based on moraines delimiting the former glacier positions and continuous records from e.g. lakes (Solomina et al., 2016). Geomorphological evidence of glacier advances, such as terminal moraines or proglacial lacustrine sediments, give relative dates of glacier fluctuations, usually with some uncertainty. Lichenomertry, a method where lichens sizes are used to infer the timing of colonisation, can provide rough estimates of moraine formation (Bickerton and Matthews, 1992; Armstrong, 2004). If the moraines contain organic material they can be dated by the $^{14}$C method (Karlén and Denton, 1976) or dendrochronological methods (Luckman, 1993; Carter et al., 1999). Cosmogenic isotopes (e.g. $^{10}$Be) can be used to directly identify the age of moraine deposition (Gosse and Phillips, 2001; Granger et al, 2013). Continuous records derived from lake sediment properties represent both the advance and retreat phases of glacier variations (Dahl and Nesje, 1994; Matthews et al., 2005; Bakke et al., 2008). As soon as the meltwater signal in proglacial lake sediments co-varies with the distance between the glacier and the lake, it can be used as an indicator of glacier extent and the corresponding Equilibrium Line Altitude (ELA, which is the altitude of equal values of accumulation and ablation, Dahl and Nesje, 1994). Reconstructions of the ELA are based on multi-proxy sediment analysis (e.g. loss-on ignition, bulk density, magnetic susceptibility grain-size distribution, and AMS dating control).

Glacier mass balance measurements demonstrate that for most regions summer temperature is the dominant control on annual mass balance (Koerner 2005; Björnsson et al., 2013). Some exceptions have been noted; glacier advances in coastal areas of Scandinavia, SE Alaska, Kamchatka and New Zealand in the late twentieth century were forced primarily by high winter precipitation (e.g. Lemke et al., 2007). This means that in order to derive precipitation information from records of glacier variations, the data should be complimented by independent temperature reconstructions. Thus, if an advance of a glacier corresponds to inferred warm summers (which would lead to increased ablation), it is likely that the advance was due to increased solid precipitation during winter (and vice versa). As has been noted previously, temperature proxies are readily available in the northern high latitudes. Various summer temperature proxies have been used to interpret past glacier fluctuations: macrofossils at the upper tree line (Dahl and Nesje, 1996), pollen (Bakke et al., 2008), chironomids (Axford et al., 2009), tree-ring data (Anchukaitis et al., 2013), sedimentary chlorophyll content (Boldt et al., 2015), melt features (Henderson, 2002), borehole temperatures (Wagner and Melles, 2002) and oxygen isotopes from ice cores (Kirkbride and Dugmore, 2006). Several sources of uncertainties should be taken into account when this approach is applied, such as the lag between glacier advances and corresponding climatic forcing, which may last for decades even for moderate size glaciers (Oerlemans, 2001) and the dating uncertainties for both geomorphic and stratigraphic data (for details see Nesje, 2009; Solomina et al., 2015; 2016).

**3.4.2. Hydroclimate signals inferred from glacier fluctuations**




Numerous detailed reconstructions of winter precipitation during the Holocene are available from Norway, where the mass balance of many maritime glaciers largely depends on accumulation rather than temperature changes (Nesje, 2009). Dahl and Nesje (1996) calculated winter precipitation at Hardangerjøkulen in southcentral Norway using proglacial sediments and tree line altitude variations over the Holocene. They found out that winter precipitation during the period from 1250 cal. BP to

600 cal. BP (ca. 750 to 1400 CE) were similar to today's values (reference period 1961-1990), then it increased to more than 120% compared to the modern values until the LIA maximum (1750 CE) before being reduced again with up to 90%. These results conflict with those obtained from Bjørnbreen in central Norway, where a comparison of the ELA with reconstructed July temperature showed that the highest values of winter precipitation during the past 2 ka occurred in the MCA at around 1000 CE (Matthews et al., 2005). The explanation for this disagreement could, to some extent, be related to the high spatial

variability of winter precipitation in Norway. To explore the spatial precipitation patterns in Norway during the Holocene, Bakke et al. (2008) used data from two proglacial sites at Folgefonna (southern Norway) and Lenangsbreene (northern Norway) together with a pollen-based July temperature reconstruction. They found that the differences in the distribution of precipitation were related to the changes in the position of the westerlies. The southernmost position of the westerlies, leading to a smaller S-N precipitation distribution gradient and large positive precipitation anomalies during the last 2ka in

western Norway, occurred around 800 and 1600 CE. The suggested link between the atmospheric circulation (NAO) and precipitation/ glacier fluctuations (Nesje, 2009), is evidenced by the advance of Nigardsbreen in Southern Norway between 1710 and 1735 CE, which was attributed mainly to increased winter precipitation linked with a period of the positive mode of the NAO (Nesje and Dahl, 2003).

Early studies of glacier advances during the LIA on Svalbard them to be responses to low temperatures (Svendsen and Mangerud, 1997; Humlum et al., 2005). However, some recent studies attribute a number of them, at least those which occurred in the 19[th] and early 20[th] centuries, to increased precipitation associated with a positive phase of the NAO (Reusche et al., 2014 D'Andrea et al., 2012). In western Svalbard, Røthe et al. (2015) suggested that open water associated with a loss of sea ice was the source of increased solid precipitation leading to the advance of Karlbreen glacier from ca. 1700 to 1500

cal. yr. BP (ca. 300 to 500 CE). Provided that Icelandic glaciers are in phase with Greenland climate, the large LIA advances in coastal areas could also reflect increased precipitation (Kirkbride and Dugmore, 2006). In Franz Josef Land, Lubinski et al. (1999), based on geomorphological evidence and [14]C dating, identified glacier advances during the ~10th, and 12th centuries, 1400 and 1600 CE, and in the early 20[th] century. The last advance occurred despite warm summers, as recorded from melt features in the Windy Dome ice core, due to anomalously high snow accumulation (Henderson, 2002). A glacier

advance at ca. 1400 CE was also noted in Novaya Zemlia (Polyak et al., 2004). It is worth noting that this was a time of increased winter precipitation as interpreted from the GISP-2 ice-core record (Zeeberg and Forman, 2001). Wagner and Melles (2002) suggested that the Holocene fluctuations of the Ymer O ice cap in east Greenland depended mainly on precipitation since the inferred fluctuations disagreed with the Greenland borehole temperature. The advance of glaciers in



the Miki and IC Jacobsen Fjords in eastern Greenland, which have been dated with lichenometry to around 900-950 CE corresponds to the MCA, and could also reflect the response of glaciers to increased precipitation (Geirsdottir et al., 2000).

In Alaska, most glacier advances have been related to cool summers (Anchukaitis et al., 2013; Wiles et al., 2014). However, the MCA advance of the Sheridan Glacier (Zander et al., 2013), also observed for glaciers in Alaska and western Canada (Menounos et al., 2009; Koch and Clague, 2011), can be probably attributed to increased precipitation due to extended La Niña like conditions (Koch and Clague, 2011). The advance of the Sheridan Glacier in 1600s CE coincides with warming summers as recorded by tree-rings (Anchukaitis et al., 2013) and a peak in sedimentary chlorophyll (Boldt et al., 2015), and is thus probably also a sign of increased winter precipitation. Using lichenomety dated moraines and the density of the sediment in Kurupa Lake (the Brooks Range in Alaska), Boldt (2013) produced continuous reconstruction of ELA variations for several glaciers in the region. By regressing ΔELA against average Arctic-wide summer temperatures from Kaufman et al. (2009), and using the residuals as a proxy for winter accumulation, he identified periods of increased (150-550, 650-1000 and 1500-1650 CE) and reduced (600, 1050-1450 and 1750 CE) accumulation. In the Chugach Mountains, south-central Alaska, McKay and Kaufman (2009) used the differences between inferred summer temperature and evidences for glacier advances and retreats to suggest a period of increased winter precipitation from 1300 to 1500 CE, and reduced winter precipitation from 1800 to 1900 CE, changes which were likely associated with variability in the strength of the Aleutian Low.

### 3.4.3. Hydroclimate from ice cores

Ice cores provide information of past climates, which is embedded in the annual layers deposited on glaciers. Several ice core parameters are used as palaeoclimate proxies, such as isotopic composition (mainly temperature), dust (e.g. storminess, aridity), air bubbles (atmospheric composition) and acidity (volcanic eruptions) (Rozanski et al., 1997). Ice core data have been used to infer past hydroclimate variability, mainly at lower latitudes like Tibet (e.g. Thompson et al., 2000; Yao et al., 2008) and the Andes in South America (e.g. Thompson et al., 1985). If annual layers can be identified and dated in ice cores, the annual accumulation may be interpreted as records of past precipitation rates (Paterson and Waddington, 1984), however the accuracy of the data is affected by processes such as re-distribution by wind, melting and dating/measuring errors (e.g. Mosley-Thompson et al., 2001). Also the cosmogenic isotope $^{10}$Be can provide estimates of palaeoaccumulation rates (Yiou, et al. 1997). Paterson and Waddington (1984) analysed ice core accumulation rates from Camp Century (Greenland) and Devon Island ice cap (Arctic Canada) and concluded that precipitation rates had only shown minor fluctuations during the last 2k, being similar to present conditions. Direct studies of ice core accumulation rates on Greenland have aimed at providing better understanding of spatiotemporal mass balance variability (e.g. Mosley-Thompson et al., 2001; Box et al., 2013), estimating precipitation trends (Mernild et al., 2015), or assessing links between precipitation/accumulation and large-scale modes such as the NAO (e.g. Appenzeller et al., 1998). Such long-term accumulation records may be useful for hydroclimate estimations. Box et al. (2013) provided a reconstruction of annual Greenland ice sheet snow accumulation



showing an increasing trend in accumulation over the last 410 year. Ljungqvist et al. (2016) used "lamina thickness" from 5 sites across the Greenland ice sheet as proxies for annual precipitation in their hemispheric hydroclimate reconstruction (see supplementary table 1 in their paper).

## 4. Regional comparisons

Here we present comparisons of hydroclimate records from two of the most replicated regions in the Arctic.

### 4.1. Canadian Arctic

In the Canadian Arctic Archipelago, available quantitative paleoclimate reconstructions fall into three classes of data: (a) relatively low temporal resolution (100-year) regional precipitation reconstructions from pollen assemblages for the boreal zone, (b) individual site-based reconstructions of lake level or precipitation, sampled at variable but relatively low temporal resolution, based on various proxies and (c) annually-resolved reconstructions typically based on varves or tree-rings. The most extensive used paleoclimate proxies in this region are pollen records from lake sediment cores. Typically annual precipitation, along with temperature, or lake levels have been the targets (Gajewski, 2015b). An extensive modern database of pollen data (Whitmore et al., 2005) enables quantitative reconstructions, and a number of reconstructions are available from the Canadian Arctic. A recent review of Holocene climate variations in the Canadian Arctic also indicated a number of other proxies in use (Briner et al., 2016), based on isotope or other physical or chemical measures. Although potentially available at sufficient temporal resolution, networks of these are not yet available. Most of the records are, however, considered as temperature records, even if some have been related to moisture and sometimes to storms.

Viau et al. (2008) and Viau and Gajewski (2009) presented regional reconstructions of annual precipitation using all available pollen records from the boreal zone of Canada and Alaska. At the scale of this study, spatial patterns in the precipitation reconstructions are not clear. Toward the west, there seems to be an increase in precipitation during the past 2000 years, whereas a long-term decrease is seen towards the east. There was no clear difference between the MCA and LIA (Fig. 9). From the Canadian Arctic, only four low-resolution reconstructions of annual precipitation are available and these are based on pollen records (Peros and Gajewski, 2008; 2009; Peros et al., 2010). All show a comparable signal, with lower precipitation during the MCA and slightly higher moisture during the LIA and during the period from 400 BCE to 600 CE, slightly earlier at site KR02 from Victoria Island (Fig. 10).

### 4.2. Fennoscandia

In Fennoscandia, several palaeolimnological studies have recently produced records indicative of past regional hydroclimatic variability. These records are based on micro-, macro- and megafossil assemblages, in addition to lithological data. Here we use a collection of sixteen palaeolimnological records (Table 1) to illustrate hydroclimatic shifts and variations in



Fennoscandia over the Common Era. In addition to outlining the hydroclimatic evolution during the past 2 ka, these data are expected to reflect the varying conditions through the MCA (here, 1000-1200 CE) and LIA (here, 1550-1750 CE). The records are derived from depositional histories of subfossil trees (Gunnarson et al., 2003; Gunnarson, 2008; Helama et al., 2017a), estimates of peat humification (Gunnarson et al., 2003; Andersson and Schoning, 2010), sediment grain-size (Si/Ti; Berntsson et al., 2015), varve thickness (Saarni et al., 2015), varve minerogenic lamina ('light sum' sensu Saarni et al., 2016), plant macrofossils (Väliranta et al., 2007), as well as chironomids and cladoceran assemblages (Luoto, 2009; Luoto and Helama, 2010; Nevalainen et al., 2011, 2013; Nevalainen and Luoto, 2012; Luoto and Nevalainen, 2015; Berntsson et al., 2015). These records originate from Sweden and Finland and represent inland areas east of the Scandinavian Mountains.

A visual inspection does not indicate any strong agreement among the records (Fig. 11). Overall, the proxy dataset suggest spatially highly variable hydroclimatic conditions throughout the Common Era. The dataset also implies notable variations between the sites and their proxy indications. Correlating the smoothed series (green lines in Fig. 11) yields correlations as low as 0.08. In fact, one might expect to find such disparity considering the peculiarities in local climate, range of proxy types and their expected indications, with an additional issue arising from dating uncertainties. Similarly estimated mean correlation remains virtually at the same level when calculated among the Swedish (0.09) and Finnish (0.13) records respectively. We note that the dating issues may not constitute a critical factor for the observed low correlations among the records; comparing the depositional histories of subfossil trees from lake archives in Sweden (SWE01; Gunnarson et al., 2003; Gunnarson, 2008) and Finland (FIN12; Helama et al., 2017a), dated by means of dendrochronology and thus without dating uncertainties, results in a correlation of -0.20. The highest correlation between any pair of sites of 0.83 is obtained between the two cladorecan-based lake water depth reconstructions from southern Finland (FIN08 and FIN14; Nevalainen et al., 2011; 2013). The highest inter-proxy correlation, on the other hand, of 0.72 was found between the cladoceran-based lake water depth reconstruction (FIN07; Nevalainen and Luoto, 2012) and FIN12, where both multi-proxy records represent southern Finland recently described by Helama et al. (2017a). Among the Swedish data, the highest correlation of 0.42 was obtained between the peat humification index (SWE03; Gunnarson et al., 2003) and a chironomid-based record of catchment erosion (SWE05; Berntsson et al., 2015). The highest inter-country correlation of 0.54 was found between SWE05 and lake water depth (FIN14; Nevalainen et al., 2013).

Considering the general palaeoclimatic interest on detecting possible changes in regional hydroclimatic conditions during the MCA-LIA transition (Trouet et al., 2009), the Fennoscandian records were classified into those indicating this MCA-LIA change based on their indications of climate becoming either wetter or drier over the past millennium, calculated simply as the difference between the proxy value means over the two periods, the LIA and the MCA. Nine out of sixteen records indicate wetter conditions towards the LIA, where eight of these records are located in Finland. Moreover, four out of seven proxies that indicate changes towards drier LIA conditions originate from Sweden. In general, these findings imply a more pronounced change towards wetter conditions in the eastern part of the region. However, it is also possible that part of these



deviations arise from varying sensitivity of the proxies to different seasons. While most of the studied proxy records likely represent hydroclimatic variations during summer, at least four of the records indicating relatively drier LIA (Luoto and Helama, 2010; Berntsson et al., 2015; Saarni et al., 2016) may actually reflect climatic and environmental factors attributable to boreal winter/spring phenomena, such as flooding, erosion, stream flow. In boreal settings, an outstanding peak in runoff

is generally attained during the spring season. The strength of this peak is strongly related to snowmelt and, in fact, the respective proxy data may be largely responding to antecedent snow conditions and thus winter precipitation. This has previously been described for eastern Finland, where a collection of proxy records reflecting either winter/spring or summer variability were found to exhibit contrasting hydroclimatic trends in respective variables through the MCA and LIA (Luoto and Helama, 2010). Therefore, the observed division of proxy records according to their indications of climate becoming

either wetter or drier through the MCA-LIA transition, may reflect, at least partly, their response to precipitation in either winter/spring or summer.

The issue of seasonal responses may be particularly interesting in the context of the long-term development of the NAO. The Fennoscandian study sites are situated in a region where the positive NAO phase is especially well attributable to increases in precipitation, and thus enhanced snowfall, during winter (Hurrell, 1995), but with decreased precipitation during much of

the summer season (Folland et al., 2009). The issue of different seasonal responses may, at least partly, explain the deviating patterns of hydroclimatic trends through the MCA and LIA among the proxies if the same climatic forcing (i.e. NAO) is anticipated to result in contrasting trends in respective records, according to their target season sensitivity. These results are in line with a predominantly positive NAO phase during the MCA, associated with generally wet winters but dry summers (Trouet et al., 2009), while a negative NAO phase during the LIA has been linked with dry winters and wet summers (Luoto

and Helama, 2010; Luoto et al., 2013; Luoto and Nevalainen, 2017). While the view of a prolonged positive phase during the MCA has been challenged by recent proxy observations (Ortega et al., 2015), additional support of generally positive NAO phase overlapping the MCA have also been presented (Wassenburg et al., 2013; Baker et al., 2015). Still, it is notable that not all of the analysed proxies indicated any distinct change from the MCA to the LIA. Moreover, the records are characterised by low resolution, and high autocorrelation makes it difficult to perform any statistical tests for this change so

the results should be regarded cautiously.

Compared to the hydroclimate fluctuations during the MCA and the LIA, a notable feature that characterises several of the Fennoscandian proxies (SWE03, SWE05, FIN07, FIN08 and FIN14) during the first millennium CE is a dry pre-MCA period of multi-centennial duration (Fig. 11). The timing of this phase appears to overlap with that of Dark Ages Cold Period (DACP, ca. 300-800 CE; Ljungqkvist, 2009). Apart from climatic changes related to temperature fluctuations, the DACP

was likely a period of generally disturbed climate conditions with considerable hydroclimatic effects. A review of palaeoclimate during the DACP showed that both wet and dry conditions have been reported from north-west Europe (Helama et al., 2017b). Using peat humification records Blackford and Chambers (1991) showed multi-site indications





towards wet conditions for the British Isles around 550 CE. Likewise, indications of wet rather than dry spring/summer conditions during the DACP have been noted around North-West Europe (Helama et al., 2017b). Thus, despite an indication of dry conditions during the DACP in some of the Fennoscandian records, the findings would imply a general lack of agreement between the available proxy indicators to demonstrate any well-defined hydroclimate DACP anomaly. These

findings illustrate the need to developing proxy records to span over the first millennium CE, in order to delve into the hydroclimatic evolution through the Arctic beyond the MCA-LIA evidence.

Finally, there is no general tendency for any anomalous 20th century conditions among the records. While some of the series exhibit trends towards wetter conditions during the past century, other records indicate relatively drier conditions over the same period (Fig. 11). This finding implies that no unprecedented hydroclimatic changes, as recorded by the Fennoscandian

dataset, can be linked to anthropogenic factors over the most recent past. The value of this finding is limited by the fact that the post-1950s interval is not present in more than half of the records, but is in agreement with the findings of Seftigen et al. (2015b).

## 5. Arctic hydroclimate synthesis from proxies and PMIP3 simulations

### 5.1. A composit of Arctic hydroclimate variability during the last 1200 years

As noted in the introduction, Ljungqvist et al. (2016) presented a reconstruction of northern hemisphere hydroclimate variability focusing on centennial varioability, where the Arctic region was represented by 18 records. Here we perform a new analysis on Arctic hydroclimate variability back to 800 CE, utilizing both high-and low-resolution records. Note that this is not a reconstruction, but rather a composite of records selected according to the method outlined below. The length of the analysis is restricted by the temporal coverage of the available series. All records have been used in previous studies and

are publicly available (Sundqvist et al., 2014, doi:10.5194/cp-10-1605-2014-supplement; Ljungqvist et al., 2016, www.ncdc.noaa.gov/palaeo/study/19725; Weissbach et al., 2016, doi:10.1594/PANGEA.849161). Figure 12 shows the location of all available proxy records and their temporal distribution, and the characteristics (location, archive and proxy type, time coverage and resolution) of each individual record are presented in table 2. The dataset is composed of 40 series and is heterogeneous regarding proxy sources: 17 records are from ice cores, 16 from lake sediment, 6 from peat and one

series is from tree rings. The majority of the records are located in the North Atlantic area (Fennoscandia, Greenland and Canadian Islands) and Alaska; only one record is available from Russia.

Here we briefly assess the potential to derive an Artcic hydroclimate record with more high-frequency variability compared to what is was found for the Arctic region in Ljungqvist et al. (2016). Note that we do not attempt a new reconstruction, but instead utilize the existing data to make a hydroclimate composite. Moreover, since we wanted to make a comparison with

the PMIP3 simulations (see below), we focused on the last 1200 years. However, rather than just merging all existing



records, the selection of the proxy records to include followed several quality criteria previously used used for the Arctic 2k database (McKay and Kaufman, 2014). Specifically, all records should i) be from north of 60°N; ii) extend back to at least 800 CE; iii) at least extend into the 1900s CE in order to include the recent warming period of the 20th century (Pages 2k Consortium, 2013); iv) have an average sample resolution of less than 50 years; and v) have at least two age control points

during the defined study period. Following these criteria, 17 records were selected (Fig. 13, table 3). This drastic selection is necessary to allow for comparison of data at centennial scale and even more important for time serie analyses. It is clear that several records did not pass the dating criteria. The spatial coverage is mainly confined to Alaska, Arctic Canada, Greenland and Fennoscandia, but these well-dated records, including many annually resolved records such as ice cores and varved sediments, offer the possibility to interpret hydroclimate variability in the Arctic area from low (e.g. tendencies) to high

frequencies (e.g. multidecadal scales). The Arctic hydroclimate variability was analysed using statistical (correlation, trend analysis) and signal analysis (wavelet analysis) in order to characterize long-term, secular and multidecadal fluctuations that occurred during the past 1200 years.

In order to extract a common pattern from the records, we created an average signal in order to reduce random noise and enhance a possible Arctic hydroclimatic signal, as classically used in climatology (see for instance Moron et al., 2006;

Hassan and Anwar, 2010). Although such common signal obtained from several climatic proxies can not be considered as a reconstruction, it is suitable for investigating and exctract different modes of variability present in the various hydrological signals. This signal is not a signal of precipitations but more certainly a summary of all processes related to the hydrological cycle (precipitation, evaporation etc.). By calculating a standardized index of the palaeoclimatic series, we reduce the part of "external" variance (Zwiers, 1987 ; Rowell, 1998), i.e. the part of variance that is not spatially coherent. This external part of

the signal is associated with, or can be considered as the part of the signal associated with, the spatially independent stochastic noise of a broad-scale climate signal. It can be white or red noises.

The trend analysis was performed using the Mann-Kendall test (Mann, 1945; Kendall, 1975), a statistical test widely used for trend analysis in climate time series. It is a non-parametric test, which has two advantages: it does not require the data to be normally distributed and it has low sensitivity to abrupt breaks due to inhomogeneous time series. The Mann-Kendall's

tau statistic corresponds to the strength of the relationship between variables and gives values between -1 and +1. Positive values indicate that the ranks of both variables increase together, indicating an increasing trend, while a negative values indicate decreasing trend. The closer to +1 or -1 the value of Kendall's tau, the stronger the trend in the time series. For this study, we choose the 95% confidence level, corresponding to $\alpha = 0.05$ significance level.  All records were standardized (i.e. zero mean and unit standard deviation) to be comparable among each other.

Continuous Wavelet transform (CWT) allows the decomposition of a time series over a time-scale space. It is used for analysis of non-stationary processes that contain periodic or aperiodic components, noise, progressive or abrupt changes (progressive transitions, singularities and breaks). CWT is useful for identifying the dominant mode of variability existing in



a time series (e.g. Debret et al., 2007; Steinhilber et al., 2012; Lapointe et al., 2016). The resulting plot of the wavelet transform, also called a scalogram, is a time versus wavelet scale or frequency contour diagram with time on the x-axis, frequency, wavelet scale or equivalent Fourier period on the y -axis and power on the z -axis. To minimize the wavelet spectrum variance resulting from sampling, the data length was artificially increased by adding zeroes (zero-padding) to the edges of the data time series. The zero-padding accentuate the edge effect and provokes a decrease of power near the edge of the spectrum as the zeroes added enter the calculation. The region of the spectrum for which the zero padding decreases the power of the wavelet transform is known as the cone of influence. In this area, energy bands are likely to be less powerful than they actually are. To determine the significance of the observed signal fluctuations, local wavelet spectra were compared to spectra of random signals that would theoretically correspond to other realizations of the same random process. Also for this analysis, we choose the 95% confidence level (Torrence and Compo, 1998).

We calculated an Arctic mean hydroclimatic record for the period 800-1975 CE (Fig. 14). The number of records decreases rapidly before 800 CE (less than 40% of the 19 records). Calculating a regional mean record allows us to investigate the common spatial climate signal and reduce noise. The period 800-1975 displays a negative trend between 800 and 1075 CE (tau = -0.404, p-value < 0.01). Subsequently the record does not show any significant trend, but a distinct decrease between 1456 and 1485 CE is notable. The Wavelet analysis performed on the Arctic hydroclimate record reveals variability on multidecadal to multicentennial scales (Fig. 15). To minimize the impact of the 1456-1485 CE event on the wavelet analysis, it was extracted by wavelet filtering and reconstructed by inverse Fourier transform. A ~80 year oscillation is present from 1050 to 1500 CE, while a ~ 140 year fluctuation is present from ~ 900 to ~1650 CE.

To determine the influence of the spatial distribution on the variability recorded in our Arctic mean record, we compare it with two regional records derived from data from the North Atlantic region (12 series) and Alaska (5 series). The two regional mean records are presented in figure 16. Visual comparison, but also correlation coefficients, between the Arctic mean record and each regional mean record (Fig. 17) highlight the stronger influence of the North Atlantic records ($r^2$=0.93, p-value < 0.01) compared to those from Alaska ( $r^2$=0.35, p-value < 0.01).

**5.2. Comparing pan-Arctic hydroclimate from proxies with PMIP3 simulations**

In addition to hydroclimate proxies, palaeoclimate modeling provides another mean to investigate temporal and spatial hydroclimate variability in the Arctic during the last millennium. As a part of the third phase of the Palaeoclimate Modeling Intercomparison Project (PMIP3: Braconnot et al., 2012), last-millennium climate simulations were performed using a set of atmosphere-ocean general circulation models according to the same experiment-protocol (Schmidt et al., 2011). These last-millennium simulations cover the period of 850-1850 CE, and can be used to investigate climate responses of changes in external forcings, such as solar irradiance and volcanic eruptions. Some of the included models were also used to simulate climate variability for the period 1850-2005 and these simulations are referred to 'historical simulations' (Taylor et al.,



2012). In this section, 6 simulations (including 3 last-millennium simulations and 3 historical simulations) performed using 3 atmosphere-ocean general circulation models (AOGCMs) including HadCM3 (Schurer et al., 2013), IPSL-CM5A-LR (Dufresne et al., 2013) and MPI-ESM-P (Jungclaus et al., 2014), are used to illustrate Arctic hydroclimate variability over the last millennium (see Table 4 for more information on the models). Modelled Arctic precipitation is then compared with the reconstructed Arctic hydroclimate (extracted from the Northern Hemisphere field reconstruction by Ljungqvist et al., 2016, henceforth referred to as L16) as well as the new synthesis presented above. We transfer both hydroclimate reconstructions and simulated annual total precipitation into z-score series, because the reconstructions represent hydroclimate indices that are not comparable with annual total precipitation in their magnitudes. Moreover, because L16 has centennial resolution, data from the simulations and the new synthesis were filtered using Gaussian filter to preserve centennial-scale variability.

Figure 18a shows differences between the MCA and the LIA over the Arctic region based on the data derived from L16. This multiple proxy reconstruction has a limited spatial coverage in the Arctic, so that hydroclimate variability can only be shown for Fennoscandia and part of Greenland. Compared to the LIA, it was wetter in northern than in southern Fennoscandia during the MCA. In contrast, Greenland shows an opposite pattern, indicates a slightly stronger precipitation increase in the south compared to the north. Turning to the models, a 3-model ensemble mean shows different spatial pattern from that of the reconstruction (Fig. 18b). The model ensemble mean shows increasing precipitation over most of Fennoscandia, except in the westernmost areas precipitation decreased during the MCA compared to the LIA. Over eastern Greenland, a distinct precipitation increase is seem at the coast, while moderate increases or even reduction (e.g. southern Greenland) are shown over other areas. From the spatial pattern derived from the individual models, we see that the discrepancy between the reconstruction and the model ensemble mean is not caused by anomalous outputs by any single model but a combination of all the models (Fig. S1). The individual models all show differences in spatial patterns compared to the reconstruction. This may imply that changes in spatial pattern of hydroclimate between MCA and LIA over Fennoscandia and Greenland are not related to changes of external forcings, suggesting that it may be caused by randomly internal variability of climate system. Another reason for the discrepancy between the reconstruction and the model simulations could be inadequate spatiotemporal availability of proxies across the Arctic region making it unsuitable to capture any robust changes in the spatial precipitation patterns between the MCA and the LIA. Hence, proxy-based hydroclimate reconstructions covering a wider area of the Arctic are needed in order to make a comprehensive model-data comparison, and further to investigate changes in spatial patterns of Arctic hydroclimate variability and their causes.

The new hydroclimate mean record shows quite coherent variability with L16 on centennial-scale (Fig. 19), especially during the early MCA (ca. 900-1200) and early LIA (ca. 1400-1600). This is not surprising since they share much of the proxy data. However, the new record suggest a shorter period of wet anomalies during the MCA compared to L16, and the variance of the new hydroclimate mean record is much larger after ca 1200 CE. Compared to the model simulations, there is





a discrepancy with the multi-proxy records during the later part of the MCA, where the model ensemble mean suggests a prolonged wet period, lasting until 1200 CE, compared to the proxy-based records. All records show drying trends over the last millennium, but both proxy-based records do not show any distinct wetting trend in the 19th century as the simulations suggest. One of the distinct features in the new hydroclimate mean record is the two distinct wetting anomalies between 1400 and 1600 CE, which are more prominent than in the model simulation and where the latter anomaly is not present in L16, which becomes quite flat after ca 1450 CE. Overall, there is a better agreement between the model simulations and the new hydroclimate mean from the 14th century and onwards compared to L16.

## 6. Arctic hydroclimate variability in the past 2000 years

### 6.1. Current understanding

Palaeoclimate research in the Arctic is challenging, due to its remoteness and lack of firm understanding of the current conditions (due to inadequate observations). However, it is evident that a better understanding of past hydroclimate variations, as well as the mechanisms behind them, are vital, as Arctic amplification does not show any signs of abating. As has been shown in this paper, significant efforts have been made to increase our understanding of hydroclimate variability in the Arctic region over the last several decades. However, it is also evident that the available records are insufficient to fully represent such a hydroclimatically inhomogeneous region. Moreover, there are still uncertainties regarding the temporal representation of some proxies and the interpretation of the actual hydroclimate information gained, as well as the season that is recorded by the records.

Over the last 1200 years, a commonly studied period as it includes the MCA and the LIA, the proxies provide no clear evidence of any systematic hydroclimate patterns across the Arctic or even regionally. In general, drier conditions during the MCA are indicated in several records in Fennoscandia (Fig. 11) and Arctic Canada (Fig. 10), but not in the North American boreal zone (Fig. 9). Similarly, the LIA seems to have been a generally wet period, as indicated by the regional comparisons and also evidence of glacier advances (see section 3.4), but again the picture is far from clear cut. The new Arctic hydroclimate mean presented in figure 14 suggested a drying trend during the MCA, but no clear trend afterwards. This is in agreement with L16, albeit the new Arctic mean displays more variability during the LIA whereas L16 is quite flat. However, it should be rememberd that L16 is a calibrated reconstruction, while the new Arctic hydroclimate record presented here is the average of a compilation of selected series. Still, the general evolution of both records suggest a more regionally coherent hydroclimate during the MCA compared to the LIA, where regional differences are more prominent. The negative trend in the new average Arctic hydroclimate in the latter part of the 20th century is surprising since it contradicts both reanalysis data and evidence from the two drought atlases presented in section 3.3 (Fig. 7). Indeed L16 also indicates dry (or no) anomalies in the Arctic region in the 1900s (see Fig. 2 in Ljungqvist et al., 2016). The discrepancies between the reanalysis data and the proxies could reflect the low spatial density of the proxies, or that many proxy records do not cover



the entire 20th century, or even an effect of low-resolution proxies not fully capturing the observed changes in the latter half of the 20th century. In the latter case, an increased number of high-resolution proxies could partly alleviate this problem. In any case, it is clear that both Arctic hydroclimate records derived from L16 and the composite presented here are unsufficient for drawing any firm conclusions for the whole region.

Compared to more recent pluvials and droughts during the MCA and the LIA, hydroclimatic variations during the first millennium CE have received relatively less attention. Detailing the climates of these times, in particular hydroclimatic variability would nevertheless be crucial in placing the strength and duration of MCA and LIA anomalies, as more frequently reconstructed from shorter proxy records, in the context of the Common Era, in addition to viewing the 20th and

21st century changes in a long-term perspective. Here, the comparison of Fennoscandian proxy series highlighted a phase of anomalous pre-MCA hydroclimate conditions, this phase overlapping notably with the timing of the previously defined Dark Ages Cold Period. As recently discussed (Helama et al., 2017b), this period did possibly undergo noticeable fluctuations, not only in temperature but also in other climatic/environmental variables including hydroclimate. In this context, our results highlight the increasing need for extending the proxy records to cover this climatic period and thus to effectively span much

of the first millennium CE. Only such proxy data can be used to reveal the extent and duration of pluvials and droughts over this interval of which hydroclimatic characteristics are still only scarcely understood.

## 6.2. The impact of comparing hydroclimate information for different seasons

Arctic hydroclimate proxies provide information for different target seasons. This is likely to have an impact on any

synthesis or reconstruction. Figure 20a shows the 20th century trends in seasonal Arctic precipitation from the ERA-20C reanalyses data (Poli et al., 2013). The trends are positive in all seasons, but most pronounced in autumn. The spatial patterns (Fig. 20b) show that the strongest precipitation increase occurred over the North Atlantic and Pacific Oceans in all seasons, but also that over the Arctic Ocean precipitation has increased. The changes over land are characterized by regional differences in both North America and Eurasia, which are more pronounced during summer, a season which is the target

season for many proxies. Regional differences are also evident in a millennium model perspective (Fig. S2). From 900-1900 CE, the model ensemble mean does not show any trends in precipitation, except for negative trend during autumn (Fig. S2a). Moreover, regional differences in long-term trends are indicated both within regions and between seasons in the three studied models (Fig. S2b). The implication of this is that in order to provide an average view of hydroclimate variability for the Arctic, there must be an even representation of records. However, given the spatial differences, it would be more

valuable to highlight those. More attention should be paid to the target season of the climate signal to avoid mixing of hydroclimate information across the seasons.

## 6. 3. Towards better understanding of spatiotemporal hydroclimate variability in the Arctic



Spatially explicit hydroclimate reconstructions provide excellent opportunities to study spatiotemporal variations, influences of forcings (e.g. Seager et al., 2007) and for proxy-model comparisons. However, due to the low number of available hydroclimate proxy records from the Arctic region, and the unbalance in spatial coverage (Table 2, Fig. 12), there is presently a long way to go before such a field reconstruction can be achieved. As noted in section 3.3, there exist two tree-ring drought atlases covering parts of the Arctic (see fig 7). However, as noted above, the data representation is limited in the Arctic region and also the usage of temperature sensitive tree-ring proxies as hydroclimate indicators need to be properly addressed. Given the precipitation sensitivity of some high-latitude trees (Fig. 6), as well as more efforts in utilizing isotope records from trees, it may be possible to extend any analyses of hydroclimate variability into Eurasia. Still, most proxies are presently confined to Arctic Canada, Greenland and Fennoscandia. Possibly, targeted regional spatial reconstructions could be achieved for well-replicated regions, such as Fennoscandia, the Nordic Sea region, or western North America. As seen from the review of the proxies in section 3, there is likley useful data that is not yet in the public domain, data that would be highly valuable for achieving a comprehensive study of Arctic hydroclimate in the past 2k. To facilitate the compilation of data, a dedicated hydroclimate proxy database needs to be developed with firm criteria for which records to include.

## 7. Recommendations for future work

- Increase the spatial coverage of hydroclimate proxies. This is particularly important for Eurasia (except Fennoscandia) and parts of North America. There are several hydroclimate records that would add valuable information which are not publicly available, so it is important to encourage palaeoclimate researchers to share their data.

- Assemble a proper Arctic2k hydro database, where the first step would be to develop criteria for which records to include as guided by priority research goals and talso aking into consideration the seasonalities in the proxies as an important next step toward a robust and defensible synthesis.

- Consolidate data, and even attempt to make a field reconstruction, for regions with sufficient number of hydroclimate proxy records in time and space. Presently there seems to be opportunities for a cross-Atlantic study, which may shed light onto observed regional hydroclimate patterns and the mechanisms behind those.

- Closer collaboration with the palaeoclimate modelling community. From the comparison between the existing "observational" data (reanalysis and proxies) and climate model simulations, discrepancies in both rate and spatial distribution were evident, and this needs to be addressed. Moreover, climate models are highly useful tools for investigating potential mechanisms behind past hydroclimate variability in the Arctic.

## Data availability

The the raw CMIP5/PMIP3 climate data used in this paper can be obtained from http://cmip-pcmdi.llnl.gov/cmip5/data_getting_started.html. The specific analyses presented here (Fig.s 1-5) will be made available



through NOAA. The tree-ring data used in Fig. 6 are available from the obtained from the International Tree Ring Data Bank (ITRDB) at https://data.noaa.gov/dataset/international-tree-ring-data-bank-itrdb and the Standardized Precipitation Evapotranspiration Index from http://spei.csic.es/database.html. The drought atlases (NADA and OWDA, Fig. 7) are accessible through https://www.ncdc.noaa.gov/paleo/study/6319 (NADA) and

http://kage.ldeo.columbia.edu:81/expert/SOURCES/.LDEO/.TreeRingLab/.DATASETS/.OWDA/.pdsi/ (OWDA). Part of the data used in section 4 (Canadian Arctic) are available from doi:10.5194/cp-10-1605-2014-supplement. The data presented in figure 11 is currently unavailable, but the aim is to make it available through NOAA in a near future. The data described and partly used for the compilation of an Arctic hydroclimate mean in section 5 are available from the following sources (see text for references): doi:10.5194/cp-10-1605-2014-supplement and www.ncdc.noaa.gov/palaeo/study/19725,

doi:10.1594/PANGEA.849161.

*Acknowledgements* This is a contribution to the PAGES 2k Network [through the Arctic 2k working group]. Past Global Changes (PAGES) is supported by the US National Science Foundations and Swiss Academy of Sciences. We thank the World Climate Research Program's Working Group on Coupled Modeling, which oversees CMIP, and the individual model

groups (listed in tables 4 and S1) for making their data available. Sofia Andersson, Annika Berntsson, François Lapointe, Tomi P. Luoto, Liisa Nevalainen, Saija Saarni and Minna Väliranta kindly provided their published hydroclimate records for this study and are thanked for their contributions. We also thank Darrell Kaufman for valuable discussions and suggestions. All authors contributed to the planning and structuring of the paper, and sections 1 and 6 were jointly written by all authors. Contributions for the other sections were composed by writing teams as follows: Sect 2: PZ and HL; Sect 3.1: PF, WD, ET

and KG; Sect 3.2: AK and ZY; Sect 3.3: HL, KS, BG, NL, KG and SH; Sect 3.4: OS and HL; Sect 4.1: KG; Sect 4.2: SH; Sect 5.1: MN, MD and NM; Sect 5.2: PZ. The following support is acknowledged: HL: the Swedish research Council (VR, grant numbers 2012-05246 and 2015-04031); SH: the Academy of Finland (grant number 288267); PF and KG: Discovery grants from the Natural Sciences and Engineering Research Council of Canada (NSERC) (grant mumber RGPIN- 2014-05810 to PF); MN: French Ministry. NJL: UK NERC (NEB501504, NE/P011527/1) EU 017008 Millennium and the

Leverhlme Trust (RPG-2014-327).

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



## Tables

**Table 1.** Characterization of Fennoscandian proxy records from Sweden (SWE) and Finland (FIN) indicative of hydroclimatic variations over the Common Era by their site name, the code, latitude and longitude, the proxy type and the type of indication, an approximate resolution of the proxy (Res; in years) and the original reference.

| Site | Code | Lat/Long | Proxy/Indication | Res | Reference |
|---|---|---|---|---|---|
| Håckren | SWE 01 | 63.17 13.50 | Tree accumulation/ Lake level | 1 | Gunnarson et al.,(2003); Gunnarson (2008) |
| Backsjömyren | SWE 02 | 62.68 14.53 | Peat humification/ Peatland water table | 40 | Andersson & Schoning (2010) |
| Stömyren | SWE 03 | 60.38 15.27 | Peat humification/ Peatland water table | 40 | Gunnarson et al.,(2003) |
| Vuoksjávrátje | SWE 04 | 66.25 15.72 | Si/Ti (coarse grain size)/ Flooding | 1.5 | Berntsson et al.,(2015) |
| Vuoksjávrátje | SWE 05 | 66.25 15.72 | Chironomids/ Catchment erosion | 80 | Berntsson et al.,(2015) |
| Kontolanrahka | FIN 06 | 60.78 22.78 | Plant macrofossils/ Peatland water table | 10 | Väliranta et al.,(2007) |
| Iso Lehmälampi | FIN 07 | 60.33 24.60 | Cladocera/ Water depth (intralake) | 80 | Nevalainen & Luoto (2012) |
| Iso Lehmälampi | FIN 08 | 60.33 24.60 | Cladocera/ Water depth (multilake) | 80 | Nevalainen et al.,(2011) |
| Iso Lehmälampi | FIN 09 | 60.33 24.60 | Chironomids/ Lake water depth | 80 | Luoto (2009) |
| Kalliojärvi | FIN 10 | 63.22 25.37 | Varve light sum/ Spring floods | 1 | Saarni et al.,(2016) |
| Kallio-Kourujärvi | FIN 11 | 62.57 27.00 | Varve thickness/ Precipitation | 1 | Saarni et al.,(2015) |
| SE Finland | FIN 12 | 61.95 28.97 | Tree accumulation/ Lake water depth | 1 | Helama et al.,(2017) |
| SE Finland | FIN 13 | 61.80 29.75 | Tree accumulation/ Peatland water table | 1 | Helama et al.,(2017) |
| Pieni-Kauro | FIN 14 | 64.28 30.12 | Cladocera/ Lake water depth | 40 | Nevalainen et al.,(2013) |
| Pieni-Kauro | FIN 15 | 64.28 30.12 | Chironomids/ Stream flow | 40 | Luoto & Helama (2010) |
| Kylmänlampi | FIN 16 | 64.30 30.25 | Chironomids/ Lake water depth | 100 | Luoto & Nevalainen (2015) |





**Table 2**. Hydroclimate proxy records available for the Arctic area. Excepted for annually resolved series, the resolutions given correspond to a mean. The asterisks indicate (*) series used by Ljungqvist et al. (2016, data available at https://www.ncdc.noaa.gov/paleo/study/19725); (**) indicates series from Weissbach et al. (2016, data available at https://doi.pangaea.de/10.1594/PANGAEA.849161); and (***) series from Sundqvist et al. (2016, data available at doi:10.5194/cp-10-1605-2014-supplement). For original references to the data, we refer to the aforementioned publications.

| ID | Region | | Site | Lat (°N) | Long (°E) | Archive | Proxy | Oldest | Youngest | Resolution (years) |
|---|---|---|---|---|---|---|---|---|---|---|
| Hydro2k_01 | Greenland | *** | N14 | 59,98 | -44,18 | Lake | BSi | 11 | 1480 | 33 |
| Hydro2k_02 | NW Norway | *** | Fiskebølvatnet | 68,413 | 14,802 | Lake | Mass Acc. Rate | 786 | 1569 | 17 |
| Hydro2k_03 | SE Norway | *** | Nattmålsvatn | 69,1793 | 17,3943 | Lake | MS (SI) | 6 | 849 | NA |
| Hydro2k_04 | N Alaska | ** | Wolverine Lake | 67,098 | -158,914 | Lake | Mass Acc. Rate | 800 | 1926 | 31 |
| Hydro2k_05 | N Norway | *** | Rystad 1 | 68,2389 | 13,7839 | Peat | Humification | 776 | 1646 | 827,5 |
| Hydro2k_06 | W Greenland | *** | SS16 | 66,91 | -50,46 | Lake | Diatom | 820,9 | 1999,6 | 26,8 |
| Hydro2k_07 | C Sweden | ** | Stömyren | 60,2083 | 13,4667 | Peat | Humification | 794 | 1928 | 37 |
| Hydro2k_08 | W Greenland | *** | SS1381 | 67,014 | -51,102 | Lake | Mineral flux | 795 | 1811 | 41 |
| Hydro2k_09 | W Hudson Bay | *** | Unit Lake | 59,404 | -97,493 | Lake | AMR/IRM | 799 | 2010 | 71 |
| Hydro2k_10 | E Finland | *** | Saarikko | 62,25 | 27,67 | Lake | d18O | 788 | 1822 | 47 |
| Hydro2k_11 | N Norway | ** | Over Gunnarsfjorden | 71,0383 | 28,1685 | Lake | Pollen | 814 | 1989 | 49 |
| Hydro2k_12 | N Norway | *** | Sellevollmyra | 69,1083 | 15,9417 | Peat | Humification | 798 | 1495 | 69 |
| Hydro2k_14 | Greenland | ** | Crête | 71,12 | -37,32 | Ice | Lamina | 800 | 1973 | 1 |
| Hydro2k_15 | Grennland | ** | Dye 3 | 65,11 | -43,49 | Ice | Lamina | 800 | 1978 | 1 |
| Hydro2k_16 | Canada | ** | East Lake | 74,88 | -109,53 | Lake | Lamina | 800 | 2005 | 1 |
| Hydro2k_17 | Greenland | ** | GISP2 | 72,6 | -38,5 | Ice | Lamina | 800 | 1987 | 1 |
| Hydro2k_18 | Greenland | ** | GRIP | 72,35 | -37,38 | Ice | Lamina | 800 | 1979 | 1 |
| Hydro2k_19 | Greenland | ** | NGRIP | 75,1 | -42,32 | Ice | Lamina | 800 | 1995 | 1 |
| Hydro2k_20 | Alaska | ** | Dune Lake | 64,42 | -149,9 | Lake | d13C | 795 | 1992 | 16 |
| Hydro2k_21 | Alaska | ** | Ongoke Lake | 59,25 | -159,42 | Lake | Diatom | 498 | 2004 | 15 |
| Hydro2k_22 | Canada | ** | Marcella Lake | 60,07 | -133,81 | Lake | d18O | 798 | 2008 | 10 |
| Hydro2k_23 | N Norway | ** | Nerfloen Lake | 61,93 | 6,87 | Lake | Particle size | 786 | 1969 | 25 |
| Hydro2k_24 | Greenland | *** | Milcent | 70,3 | -44,55 | Ice | Acc. Rate | 1174 | 1966 | 1 |
| Hydro2k_27 | Alaska | *** | Takahula Lake | 67,35 | -153,66 | Lake | d18O.calcite | 753 | 2001 | 50 |
| Hydro2k_31 | E Finland | ** | Pieni-Kauro Lake | 64,28 | 30,12 | Lake | Chironomid | 800 | 1990 | 46 |
| Hydro2k_32 | Finland | ** | Southern Finland | 61,5 | 28,5 | Trees | Ring width | 800 | 1993 | 1 |
| Hydro2k_34 | S Sweden | ** | Fågelmossen 1 | 59,29 | 14,27 | Peat | Humification | 794 | 1914 | 15 |
| Hydro2k_35 | S Sweden | ** | Fågelmossen 2 | 59,29 | 14,27 | Peat | Humification | 793 | 1967 | 12 |
| Hydro2k_37 | Finland | ** | Kontolanrahka Lake | 60,78 | 22,78 | Peat | Humification | 750 | 1913 | 30 |
| Hydro2k_38 | Greenland | ** | NGT B16 | 73,9 | -37,6 | Ice core | Acc. Rate | 1471 | 1992 | 1 |
| Hydro2k_39 | Greenland | ** | NGT B17 | 75,25 | -37,62 | Ice core | Acc. Rate | 1363 | 1992 | 1 |
| Hydro2k_40 | Greenland | ** | NGT B18 | 76,61 | -36,4 | Ice core | Acc. Rate | 874 | 1992 | 1 |





| Hydro2k_41 | Greenland | ** | NGT B19 | 78 | -36,39 | Ice core | Acc. Rate | 753 | 1953 | 1 |
| Hydro2k_42 | Greenland | ** | NGT B20 | 78,83 | -36,5 | Ice core | Acc. Rate | 775 | 1993 | 1 |
| Hydro2k_43 | Greenland | ** | NGT B21 | 80 | -41,1 | Ice core | Acc. Rate | 1372 | 1993 | 1 |
| Hydro2k_44 | Greenland | ** | NGT B22 | 79,34 | -45,91 | Ice core | Acc. Rate | 1372 | 1993 | 1 |
| Hydro2k_45 | Greenland | ** | NGT B23 | 78 | -44 | Ice core | Acc. Rate | 1023 | 1993 | 1 |
| Hydro2k_46 | Greenland | ** | NGT B26 | 77,25 | -49,21 | Ice core | Acc. Rate | 1505 | 1994 | 1 |
| Hydro2k_48 | Greenland | ** | NGT B29 | 76 | -43,49 | Ice core | Acc. Rate | 1471 | 1994 | 1 |
| Hydro2k_49 | Greenland | ** | NGT B30 | 75,01 | -42 | Ice core | Acc. Rate | 1242 | 1988 | 1 |



**Table 3.** List of the 17 proxy records used for the new synthesis. For detailed information about the records, see table 2.

| ID |
| --- |
| Hydro2k_04, Hydro2k_07, Hydro2k_14, Hydro2k_15, Hydro2k_16, Hydro2k_17, Hydro2k_18, Hydro2k_19, Hydro2k_20, Hydro2k_21, Hydro2k_22, Hydro2k_23, Hydro2k_31, Hydro2k_32, Hydro2k_34, Hydro2k_35, Hydro2k_37 |

**Table 4.** Information on the 3 PMIP3 climate models used in this study. The spatial resolution of atmosphere is expressed by the number of longitudinal grid cells × the number of latitudinal grid cells.

| Model | Institute/Country | Spatial resolution of atmosphere |
| --- | --- | --- |
| HadCM3 | MOHC/UK | 96×73 |
| IPSL-CM5A-LR | IPSL/France | 96×95 |
| MPI-ESM-P | MPI-M/Germany | 196×98 |



**Figures**

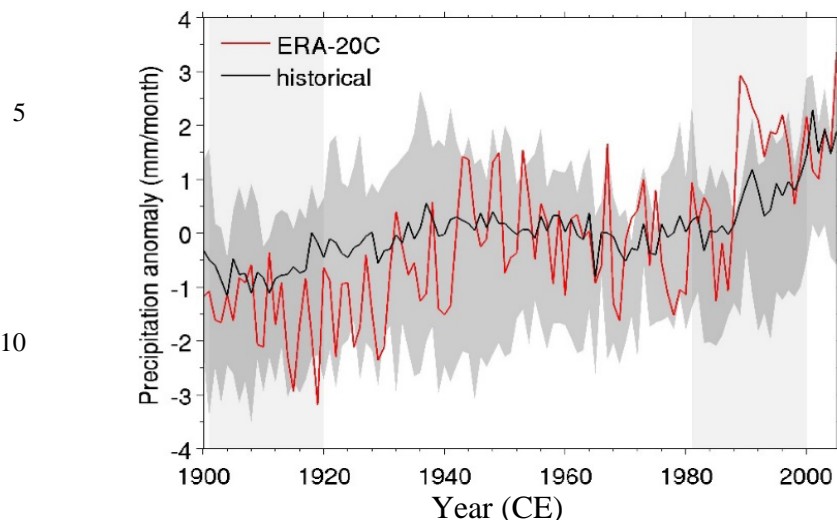

**Figure 1.** Comparison between the modelled annual precipitation anomaly (relative to the period of 1961-1990) variability (black) of the Arctic region (>= 60°N) derived from ensemble mean of historical (1900-2005) simulations performed using 12 CMIP5 climate models (Taylor et al., 2012) and observed variability (red) obtained from the ERA-20C reanalysis dataset (Poli et al., 2013). Solid lines represent multi-model ensemble means, while shadings around the solid lines represent the interquartile ensemble spreads (25th and 75th quartiles). The two light gray shadings mark the time period of 1901-1920 (left) and 1981-2000 (right).





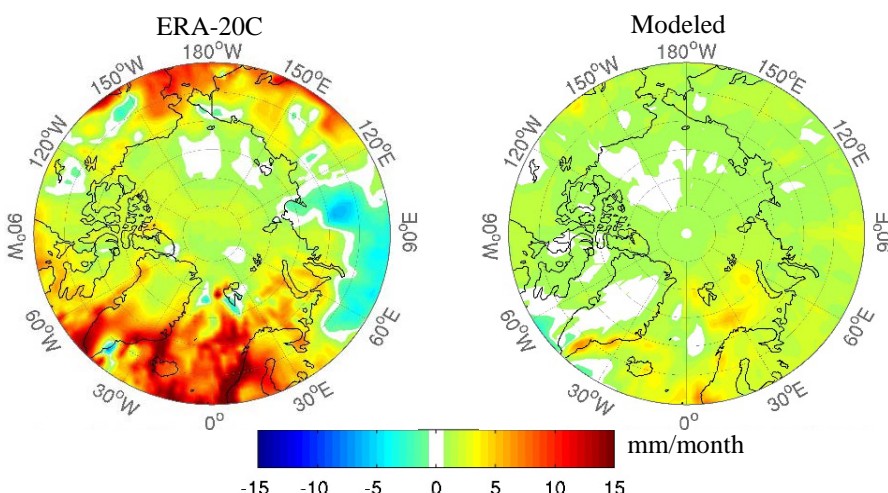

5 **Figure 2.** Observed (left) and modeled (right) changes in annual precipitation over the Arctic region (>= 60°N) relative to the reference period 1901-1920 averaged over the period 1981-2000. The observed pattern is obtained from the ERA-20C reanalysis dataset (Poli et al., 2013). The modeled pattern is derived from ensemble mean of historical (1900-2005) simulations performed by 12 CMIP5 climate models (Taylor et al., 2012).





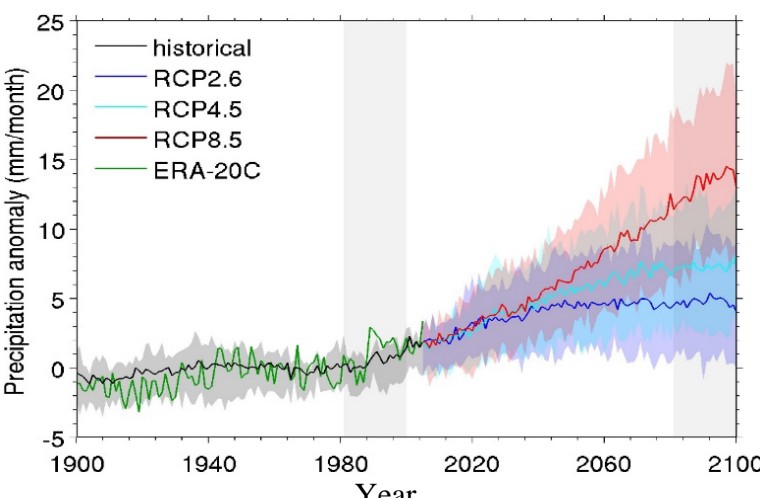

**Figure 3.** Annual precipitation anomaly (relative to the period of 1961-1990) variability of the Arctic region (>= 60°N) derived from ensemble mean of historical (1900-2005) and RCP (2006-2100) simulations conducted using 12 CMIP5 climate models (Taylor et al., 2012). Green line shows the variability of annual precipitation anomaly (relative to the period of 1961-1990) derived from the ERA-20C reanalysis dataset (Poli et al., 2013). Solid lines represent multi-model ensemble means, while shadings around the solid lines represent the interquartile ensemble spreads (25[th] and 75[th] quartiles). The two light gray shadings mark the time period of 1981-2000 (left) and 2081-2100 (right).





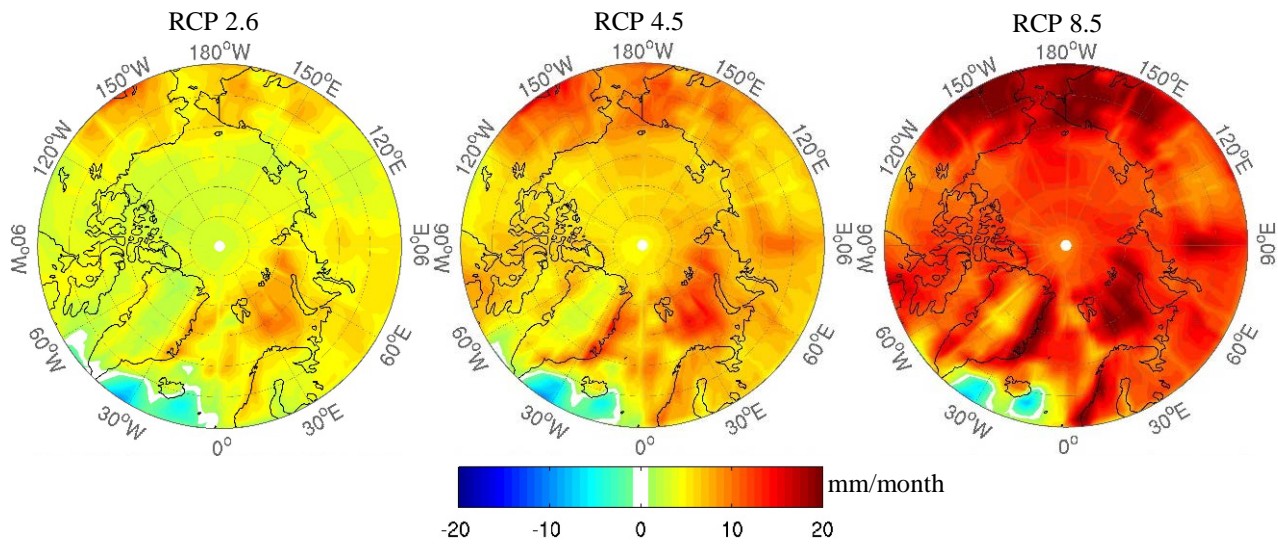

**Figure 4.** Multi-model (12 CMIP5 models, Taylor et al., 2012) average changes in annual precipitation relative to the
reference period 1981-2000 averaged over the period 2081-2100 under RCP2.6 (left panel), RCP4.5 (middle panel) and
RCP8.5 (right panel) forcing scenarios.



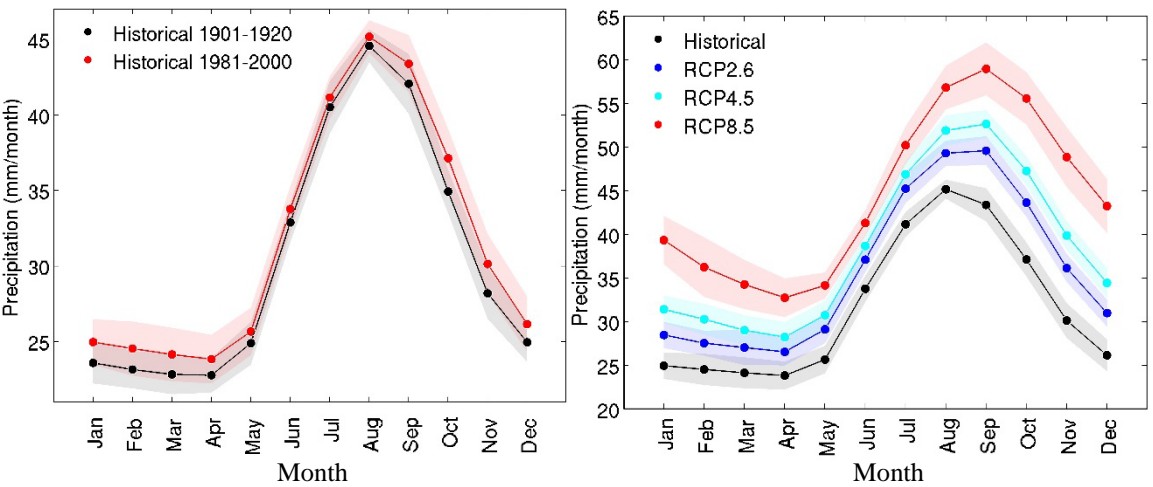

**Figure 5.** Multi-model (12 CMIP5 models, Taylor et al., 2012) average of seasonal cycle of annual precipitation over the Arctic region (>= 60°N) over the periods of 1901-1920 (black) and 1981-2000 (red) (left panel), and over the periods of 1981-2000 (black) and 2081-2100 (other colors) (right panel). Solid lines represent multi-model ensemble means, while shadings around the solid lines represent uncertainties expressed as ±2 times the standard deviation of the mean monthly precipitation over a 20-years period.





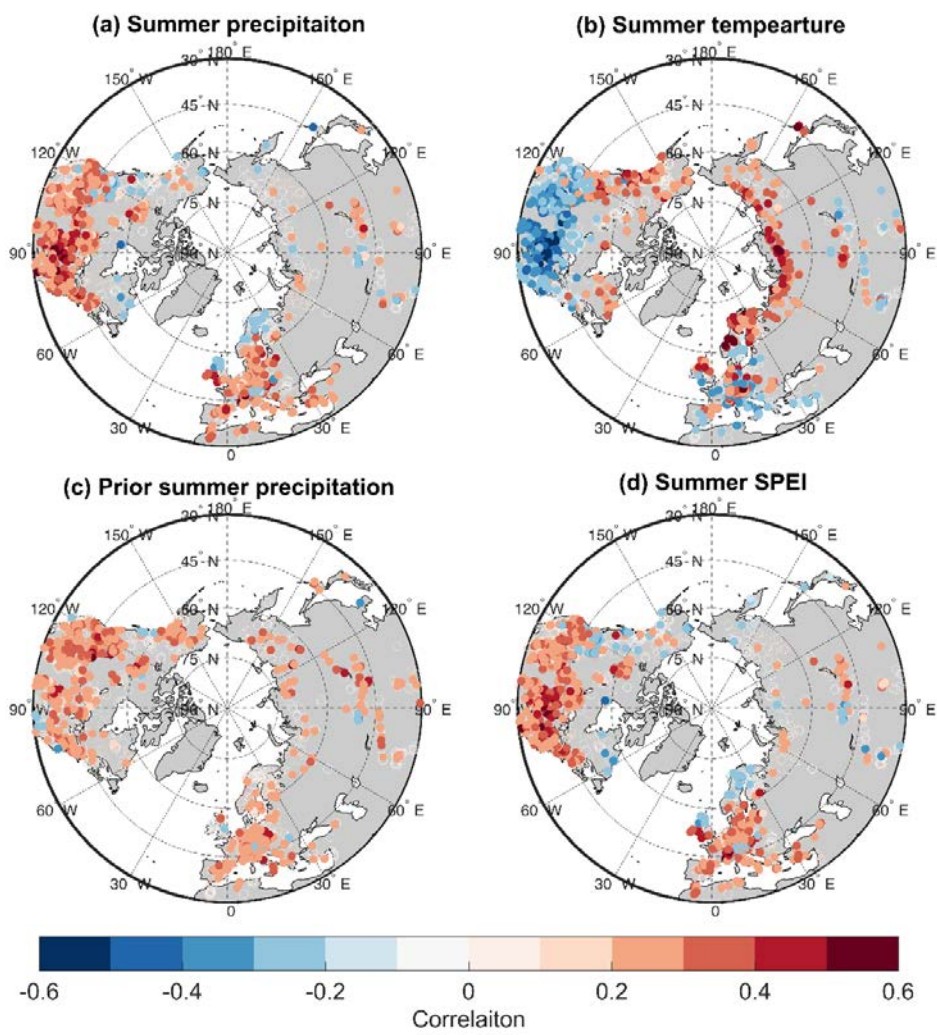

**Figure 6.** Correlation coefficient between tree-ring chronologies (obtained from the International Tree Ring Data Bank (ITRDB), https://data.noaa.gov/dataset/international-tree-ring-data-bank-itrdb) and summer (JJA) precipitation (a), temperature (b) prior summer precipitation (c), and summer Standardized Precipitation Evapotranspiration Index (SPEI, Vicente-Serrano et al., 2010, data obtained from http://spei.csic.es/database.html) computed over a 1-month time-scale. Correlations are computed on detrended data over the overlapping period (minimum 30 years). Filled markers represent coefficients significant at P < 0.05.



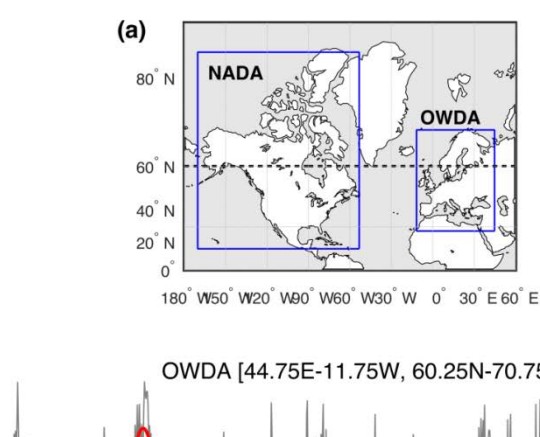

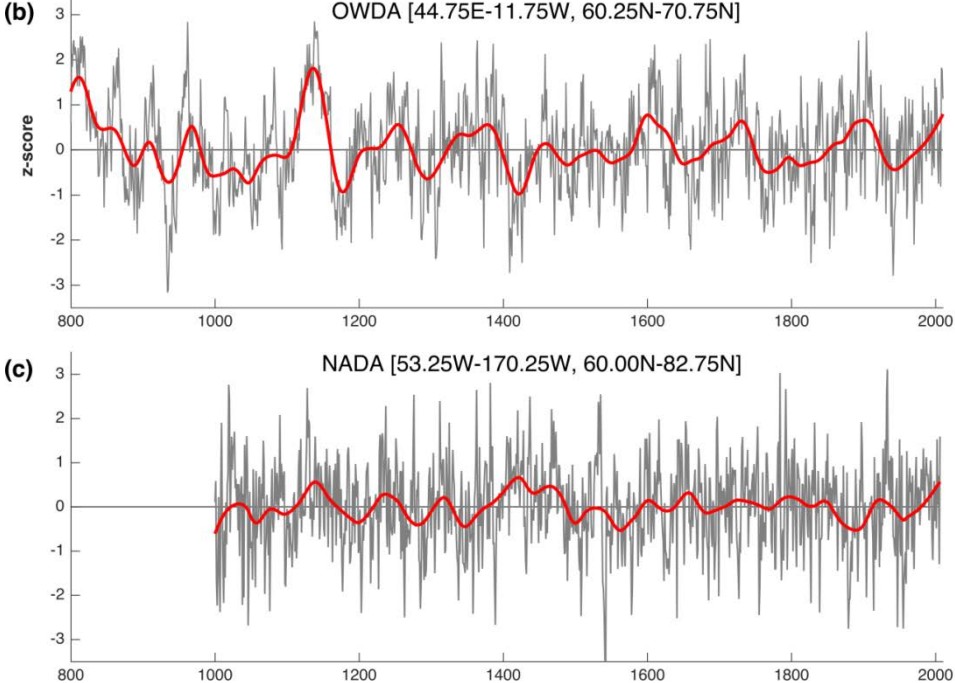

**Figure 7.** The North American (NADA, Cook et al., 2004, https://www.ncdc.noaa.gov/paleo/study/6319) and Old Word (OWDA, Cook et al., 2015, http://kage.ldeo.columbia.edu:81/expert/SOURCES/.LDEO/.TreeRingLab/.DATASETS/.OWDA/.pdsi/) Drought Atlases over the Arctic region. The full spatial domains of the two atlases (a), and regional averages over latitudes > 60°N (b-c) transformed into z-scores and filtered with a filtered with a 100-year loess filter (red lines).



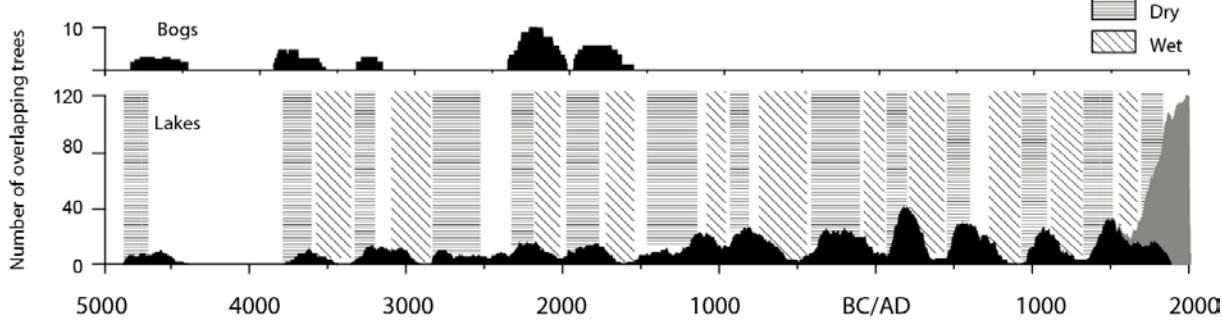

**Figure 8.** Changes in Scots pine sample depth through time for the bog- and lake sites. The grey shaded area in the end represents living trees. Interpreted wet and dry periods are sown in grey breaks (Adapted from Gunnarson 2008).




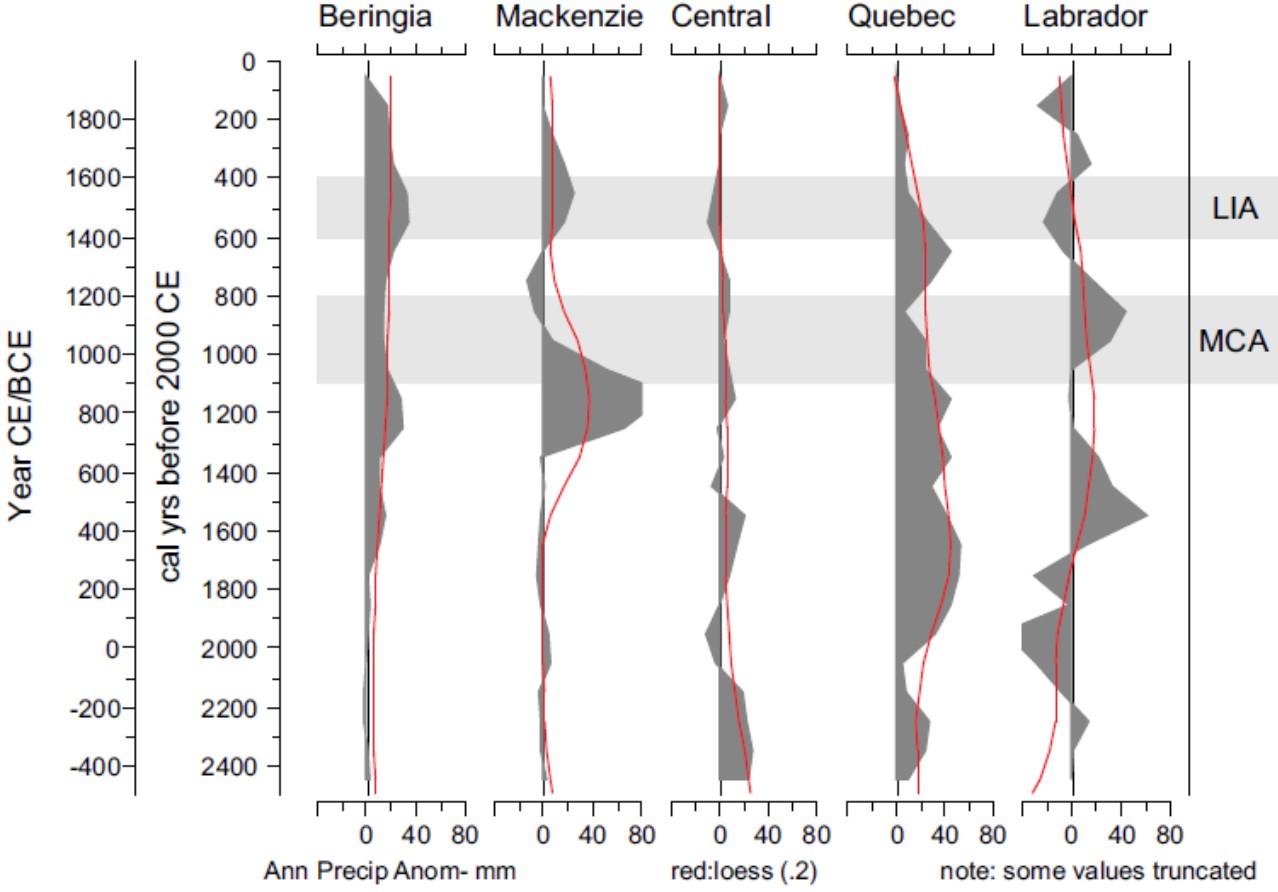

**Figure 9.** Regional reconstructions of annual precipitation from the boreal zone of North America. Each is the average of all
5   pollen records from the region. Beringia is Alaska and western Yukon, and the others are self-explanatory.





**Figure 10.** Annual precipitation reconstructions, based on pollen assemblages, from four sites in the Canadian Arctic. BC01:
Melville Island; MB01 & KR02: western Victoria Island; SL06: Boothia Peninsula. Dotted lines are loess smoothers. For
Lake KR02: red=Modern analogue technique, blue=WAPLS, black=PLS





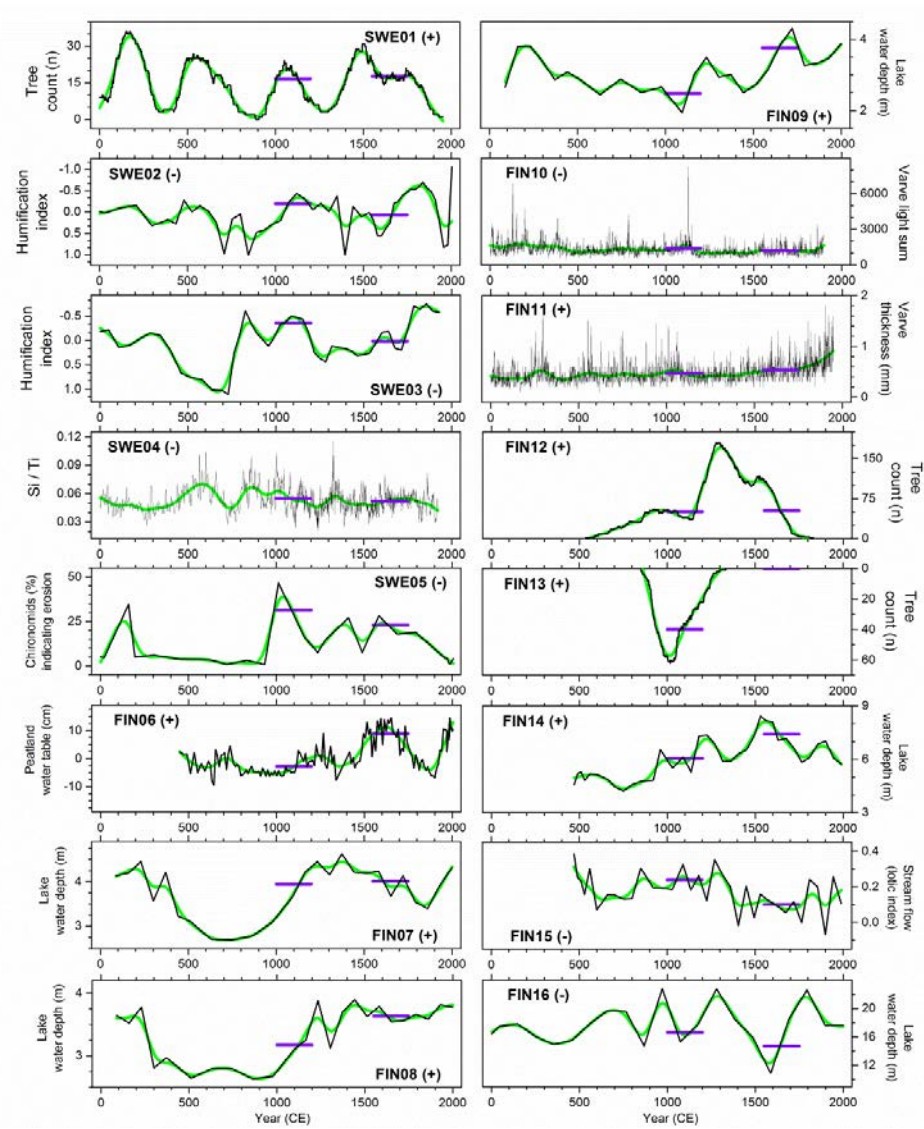

**Figure 11.** Hydroclimatic variations in the Nordic proxy records from Sweden (SWE) and Finland (FIN) over the Common Era (see Table NN). The mean levels (violet line) during the Mediaeval Climate Anomaly (MCA; 1000-1200 CE) and Little Ice Age (LIA; 1550-1750 CE) were calculated from the published records (black line), those being additionally smoothed using 200-year spline function (green line). Proxy data indicating change from MCA towards wetter (drier) LIA conditions are remarked with plus (minus) sign. The graphs have been arranged so that wet conditions are indicated upward and dry conditions downward change in the figure.





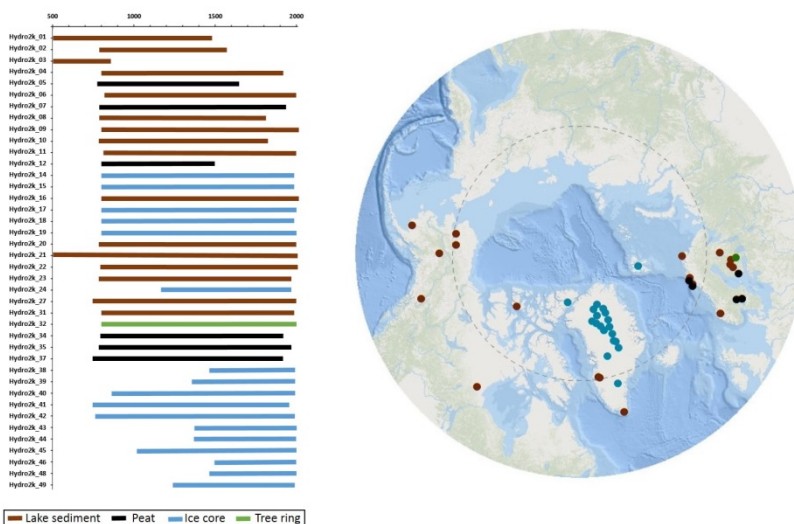

**Figure 12.** Temporal distribution of the hydroclimate proxy records from 500 to 2000 AD and their locations.



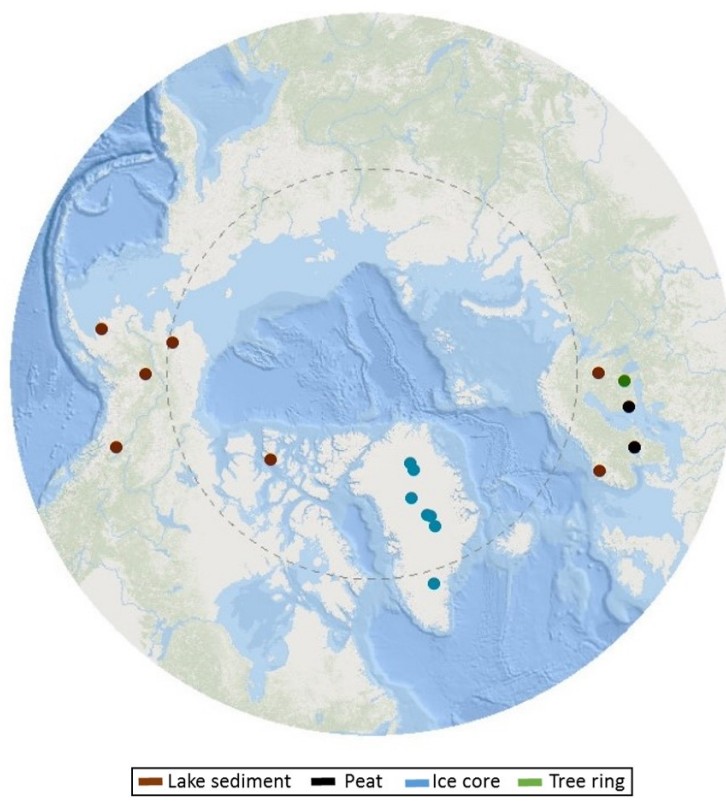

**Figure 13.** Map showing the location of hydroclimate proxy records used for the new synthesis. For information about the records, see tables 1 and 2.





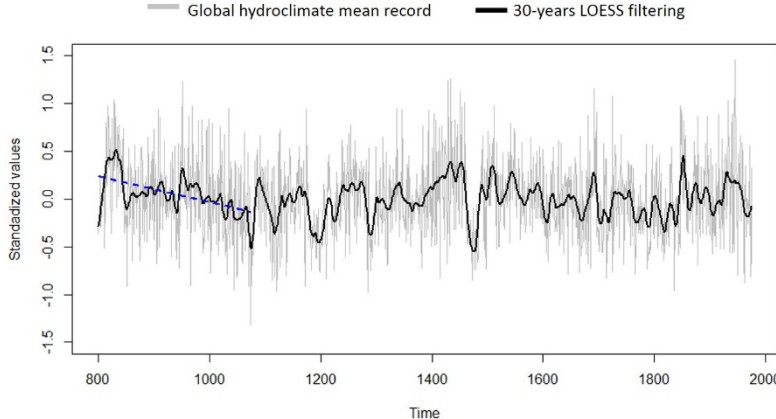

**Figure 14.** Global Arctic hydroclimate mean record and it corresponding 30-years LOESS filtering based on the 19 series

selected and presented in table 2. Blue and red dashed lines correspond to linear trends determine using Mann-Kendall test.



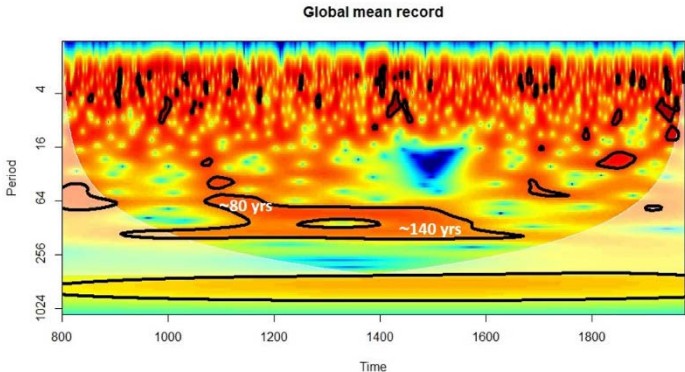

**Figure 15.** Wavelet analysis of the Arctic hydroclimate mean record. Colors represent the amplitude of the signal at given time and spectral period (red equals highest power, blue lowest). White line and dashed black line corresponds to cone of influence on wavelet coherence spectrum and global wavelet spectrum. Confidence level of 95% (α=0.05) is indicated on wavelet spectrum with the black line.



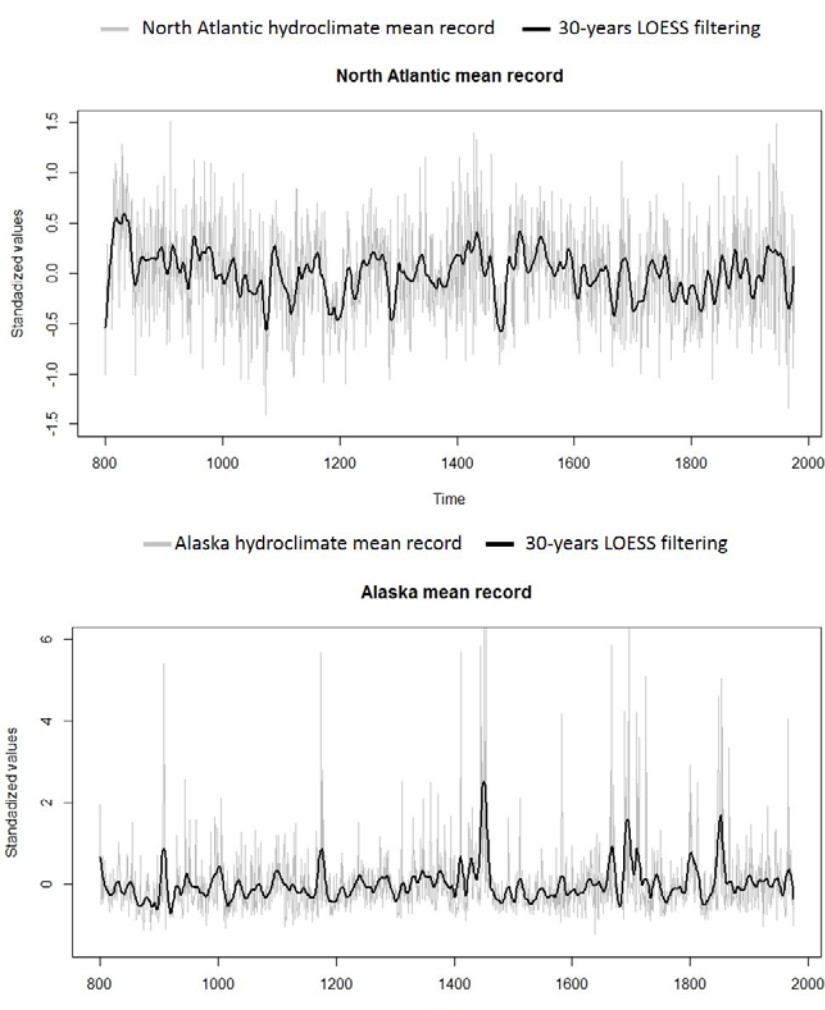

**Figure 16.** Regional hydroclimate mean records and their corresponding multidecadal variability (30-year loess filtering) for

the North Atlantic region (top) and Alaska (bottom).





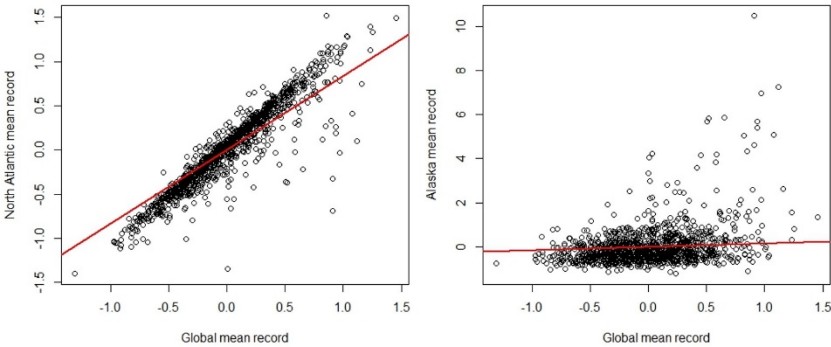

**Figure 17.** Linear regression between the global Arctic mean record and the North Atlantic regional hydroclimate mean record (Left) and the Alaska regional hydroclimate mean record (Right).



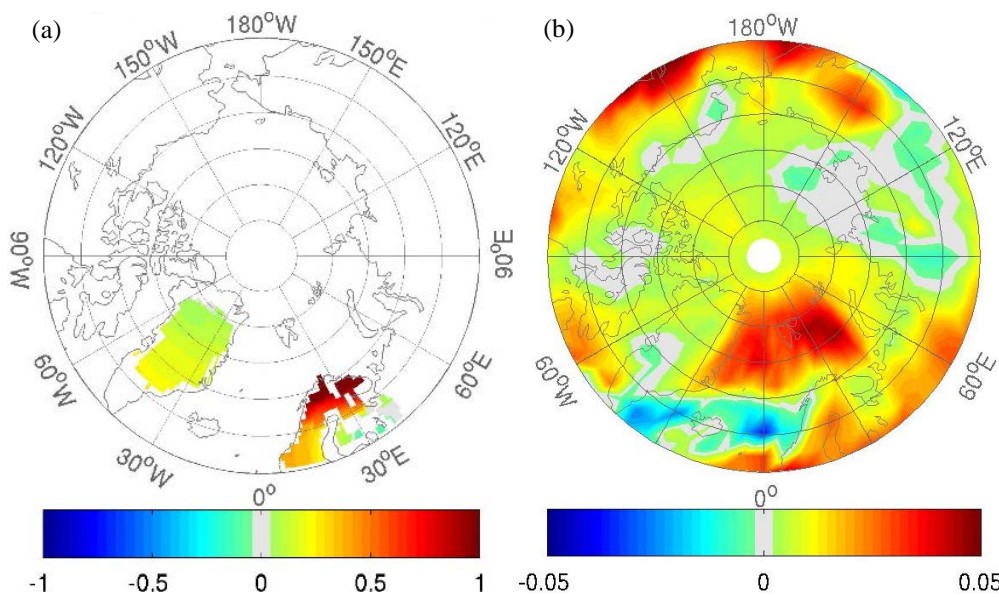

**Figure 18.** Spatial pattern of differences in annual hydroclimate between MCA (950-1250) and LIA (1450-1850) based on (a) hydroclimate reconstruction (Ljungqvist et al., 2016) and (b) ensemble mean of 3 last-millennium simulations. The values in (a) are hydroclimate index, while those in (b) are standardized annual total precipitation.





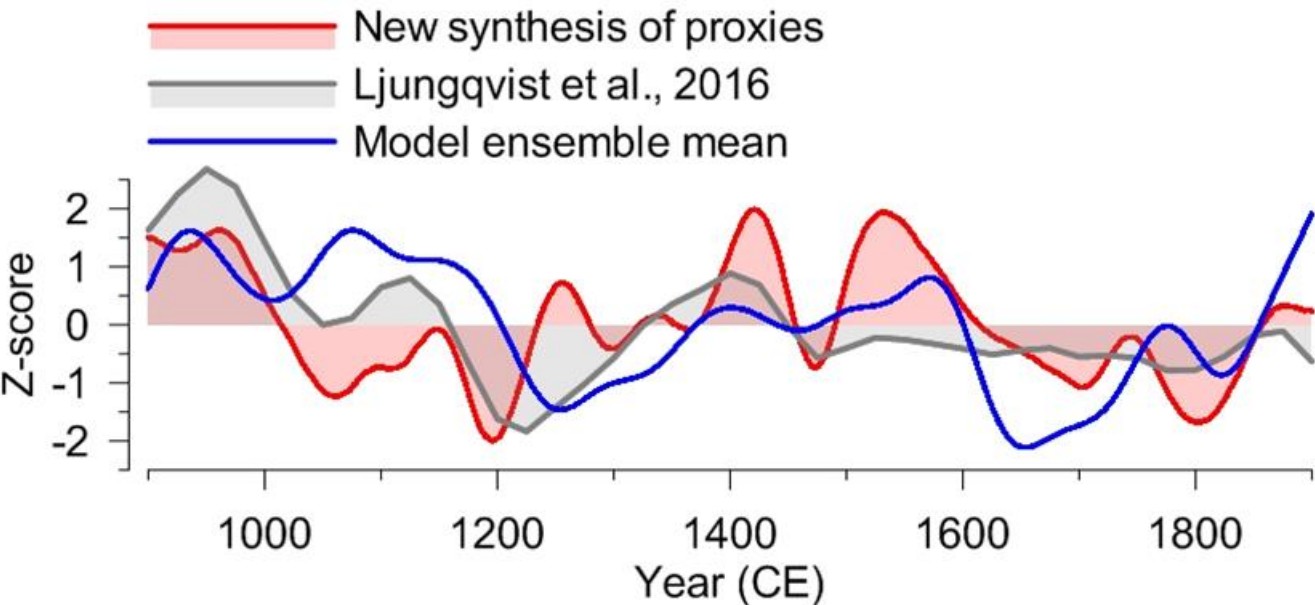

**Figure 19.** Comparison of centennial-scale annual hydroclimate variability (after Gauss filter) based on hydroclimate index reconstruction (red, Ljungqvist et al., 2016), the new hydroclimate proxy synthesis (gray) and ensemble mean of 3 last-millennium precipitation simulations (blue) over the Arctic (≥ 60°N) north Atlantic section.





**Figure 20.** Top: Variability and linear trends of the Arctic spring, summer, autumn and winter total precipitation anomalies over 1900-2010 from the ERA-20C reanalysis dataset (Poli et al. 2013). Bottom: Spatial patterns of linear trends of the Arctic spring, summer, autumn and winter total precipitation anomalies over 1900-2010. Dots mark grids where the trend is significant (p<0.01).