# Peer review of "Arctic hydroclimate variability during the last 2000 years – current understanding and research challenges"

_Climate of the Past, 2017_

## Referee Comment (RC1) · Anonymous Referee #1 · 17 Apr 2017

The overall quality of the discussion paper "Arctic hydroclimate variability during the last 2000 years – current understanding and research challenges" intended for the Special Issue "Climate of the past 2000 years: global and regional syntheses", is good. The manuscript presents a substantial, thorough, and updated contribution on hydroclimate variability during the past 2 ka in the Arctic. The concepts, ideas, methods, and data from different climate archives are clearly presented. The results are discussed in an appropriate and balanced way, including appropriate references. The scientific results and conclusions are presented in a clear, concise, and well structured way. The number and quality of figures and tables are appropriate, as well as the English language.

Specific comment: Please note that Lake Nerfloen listed in Table 2 is located in western

Norway, not Northern (N) Norway.

1. Does the paper address relevant scientific questions within the scope of CP? YES 2. Does the paper present novel concepts, ideas, tools, or data? YES 3. Are substantial conclusions reached? YES 4. Are the scientific methods and assumptions valid and clearly outlined? YES 5. Are the results sufficient to support the interpretations and conclusions? YES 6. Is the description of experiments and calculations sufficiently complete and precise to allow their reproduction by fellow scientists (traceability of results)? YES 7. Do the authors give proper credit to related work and clearly indicate their own new/original contribution? YES 8. Does the title clearly reflect the contents of the paper? YES 9. Does the abstract provide a concise and complete summary? YES 10. Is the overall presentation well structured and clear? YES 11. Is the language fluent and precise? YES 12. Are mathematical formulae, symbols, abbreviations, and units correctly defined and used? YES 13. Should any parts of the paper (text, formulae, figures, tables) be clarified, reduced, combined, or eliminated? NO 14. Are the number and quality of references appropriate? YES 15. Is the amount and quality of supplementary material appropriate? YES

---

## Short Comment (SC1) · 5 May 2017

The PAGES Data Stewardship Integrative Activity seeks to advance best practices for sharing data generated and assembled as part of all PAGES-related activities. As part of this activity, a team of reviewers has been constituted for the "Climate of the Past 2000 years" Special Issue. The data team is reviewing the data handling within each of the CP-Discussion papers in relation to the CP data policy and current best practices. The team has identified essential and recommended additions for each paper, with the goal of achieving a high and consistent level of data stewardship across the 2k Special Issue. We recognize that an additional effort will likely be required to meet the high level of data stewardship envisaged, and we appreciate dedication and contribution of the

authors. This includes the use of Data Citations (see example in supplement). We ask authors to respond to our comments as part of the regular open interactive discussion. If you have any questions about PAGES Data Stewardship principles, please contact any of us directly.

Best wishes for the success of your paper,

2k Special Issue Data Review Team (Darrell Kaufman, Nerilie Abram, Belen Martrat, Raphael Neukom, Scott St. George) and ex-officio team members (Marie-France Loutre, Lucien von Gunten)

Essential additions for this paper:

(1) Table 1: Add Data Citations for all of the proxy datasets listed in this table and shown in Fig 11. For those data not already in a public repository, submit essential metadata and data, and add the corresponding Data Citations.

(2) Table 2: List the proxy climate time series shown in Figs 8, 9, and 10, along with a corresponding Data Citations. Add the original publication citations for each record in Table 2 (like in Table 1).

(3) Submit the time series of the resulting hydroclimate composites (Figs 14, 16 and 19) for archival and include the data citation.

Possibly essential, depending on source of the data:

(4) If the data shown in Fig 6 are based on chronologies already archived and easily accessible in the ITRDB, then all is well. If instead the chronologies from the ITRDB were detrended or otherwise modified by the authors, then those new chronologies must be submitted for archival as part of this study. Either way, please clarify the data methods used to create Fig 6.

Recommended:

(5) Contrary to what is shown in Table 2, many of the records appear to have been

taken from Ljungqvist et al.'s global compilation. Essential metadata that are needed for intelligent reuse of the data in new synthesis are missing from the Ljungqvist et al. compilation, which undermines a primary goal of the PAGES data stewardship activity. We strongly encourage the authors to use the opportunity of this synthesis paper to start with the original datasets and to submit a more complete set of metadata for archival. We note that the Ljungqvist et al. dataset is truncated at 850 AD (the time frame considered in their study). For the current study, the full time series should be used and archived.

Please also note the supplement to this comment:
http://www.clim-past-discuss.net/cp-2017-34/cp-2017-34-SC1-supplement.pdf

---

## Short Comment (SC2) · 7 May 2017

F. Charpentier Ljungqvist

fredrik.c.l@historia.su.se

This is a very well written – and very timely and important – article that I hope will be published speedily after only minor revision. It serves both as an excellent review article of the state-of-the-art knowledge about the hydroclimate signal in various hydroclimate proxy records from the Arctic/sub-Arctic region and at the same time presents new important findings leading the research forward. Because the article to a considerable extent discusses a recent article of mine (Ljungqvist et al. 2016), I have read it very carefully with great interest and found some minor things that the authors may want to correct or improve prior to final publication.

The article, with new additional proxy records, represents a clear improvement of

the understanding of centennial-scale Arctic hydroclimate variability compared to Ljungqvist et al. (2016). The new reconstruction shows more variability during the Little Ice Age. Although this likely partly is because of new additional proxy records it may also be related to slightly different filtering techniques to extract centennial-scale variations.

I have listed my comments below after page number and line number:

Abstract, line 29: To mention the Arctic amplification phenomenon in the introduction to the Abstract seems a bit out of place here in this article devoted to the study of long-term Arctic hydroclimate variations.

Page 2, line 8: It would be clearer to write "anthropogenic greenhouse gas emissions" here instead of the more vague "human activities".

Page 2, line 11: Add the reference Hind et al. (2016) to the discussion of Arctic amplification.

Page 2, lines 29–30: Add Shi et al. (2012) to the list of references here.

Page 6, lines 8–11: References should be provided to the statement that lake cryosphere has not changed during the past two millennia. I am not so sure that this statement is fully correct, a least not at all locations in the Arctic.

Page 6, lines 29–30: These processes are partly dependent on the depth of the active layer. In regions with a depth active layer (e.g. permafrost regions with warm summers) it is less the case.

Page 10, line 9: Southern Scandinavia is south of 60°N and not in the Arctic. Better to write Central Scandinavia – a region that still is "less harsh" from an Arctic point of view.

Page 13, line 17: Also cite Borgmark and Wastegård (2008) here.

Page 14, lines 12–14: Please, double-check the time periods here.

Page 15, lines 24–25: Add the also very relevant references Esper et al. (2002), Schneider et al. (2015), and Stoffel et al. (2016) here.

Page 15, line 26: Add Shi et al. (2012) to the list of references here.

Page 17, line 14: Medieval Warm Period/Medieval Climate Anomaly – medieval is too vague and a different meaning (as a time period) in history than as a climate period.

Page 18, line 11: Maybe it is worth to mention that a tree-line as high as 73°N only occurs in parts of central Siberia (e.g. the Taimyr Peninsula)?

Page 22, line 31: "GISP-2" should be written "GISP2".

Page 22, line 32: "O" in "Ymer O" should be with upper case "O".

Page 23: line 2: Geirsdóttir with "ó".

Page 25, line 2: Why this two time periods for the MCA and LIA, respectively? Some motivation for the choice of time periods would be good. Most data for Fennoscandia indicates pretty old cold conditions during parts of the 12th century whereas many regions appear to have been rather, or very, warm during the 10th century (which also seems to have been the warmest century of the MCA in the Northern Hemisphere).

Page 26, line 29: Add a reference to the new article by Helama et al. (2017) about the DACP here. The reference to Ljungqvist (2009) is wrong here: Ljungqvist is misspelt ("k" instead of "q") and it should be Ljungqvist (2010) – that discusses the DACP – and NOT Ljungqvist (2009) that does not do so.

Page 26, line 30: The word "disturbed" is ambiguous and vague here. It was cold but in what other ways "disturbed" compared to other periods. Larger variability in the climate?

Page 27, line 16: The word "variability" is misspelt here.

Page 28, line 3: "PAGES" should be written with upper case letters (e.g. PAGES and

not Pages).

Page 29, line 28: Also cite Schmidt et al. (2012) here.

Line 31, 26: There is an error here: Ljungqvist et al. (2016) does NOT present a calibrated reconstruction. It is an uncalibrated index, ranging from –2 to +2, and with exceeding values truncated to –2 and +2, respectively. All values are standard deviations with respect to the mean of 1000–1899 CE. In some aspects, the approach in Ljungqvist et al. (2016) has some similarities with PDSI and other hydroclimate indices. So, in this respect it is no real differences between Ljungqvist et al. (2016) and the new hydroclimate index in this article.

Page 42, line 8. "ans" should be "and".

Page 61, line 20 (and in numerous citations throughout the article): Weissbach should be Weißbach.

References:

Borgmark, A. and Wastegård, S.: Regional and local patterns of peat humification in three raised peat bogs in Värmland, southcentral Sweden, Geologiska Föreningens i Stockholm Förhandlingar (GFF), 130, 161–176, 2008.

Esper, J., Cook, E. R., and Schweingruber, F. H.: Low-frequency signals in long tree-ring chronologies for reconstructing past temperature variability, Science, 295, 2250–2253, 2002.

Helama, S., Jones, P. D. and Briffa, K. R.: Dark Ages Cold Period: A literature review and directions for future research, Holocene, doi:10.1177/0959683617693898, 2017.
Hind, A., Zhang, Q., and Brattström, G.: Problems encountered when defining Arctic amplification as a ratio, Scientific Reports, 6, 2016.

Ljungqvist, F. C.: Temperature proxy records covering the last two millennia: a tabular and visual overview, Geogr. Ann. A., 91A, 11–29, 2009.

Ljungqvist, F. C.: A new reconstruction of temperature variability in the extra-tropical Northern Hemisphere during the last two millennia, Geogr. Ann., 92A, 339–351, 2010.

Ljungqvist, F. C., Krusic, P. J., Sundqvist, H. S., Zorita, E., Brattström, G., and Frank, D.: Northern Hemisphere hydroclimate variability over the past twelve centuries, Nature, 532, 94–98, 2016.

Shi, F., Yang, B., Ljungqvist, F. C., and Yang, F.: Multi-proxy reconstruction of Arctic summer temperatures over the past 1400 years, Climate Research, 54, 113–128, 2012.

Schmidt, G. A., Jungclaus, J. H., Ammann, C. M., Bard, E., Braconnot, P., Crowley, T. J., Delaygue, G., Joos, F., Krivova, N. A., Muscheler, R., Otto-Bliesner, B. L., Pongratz, J., Shindell, D. T., Solanki, S. K., Steinhilber, F., and Vieira, L. E. A.: Climate forcing reconstructions for use in PMIP simulations of the Last Millennium (v1.1), Geosci. Model Dev., 5, 185–191, doi:10.5194/gmd-5-185-2012, 2012.

Schneider, L., Smerdon, J. E., Büntgen, U., Wilson, R. J. S., Myglan, V. S., Kirdyanov, A. V., and Esper, J.: Revising midlatitude summer temperatures back to A.D. 600 based on a wood density network, Geophysical Research Letters, 42, 4556–4562, doi:10.1002/2015gl063956, 2015.

Stoffel, M., Khodri, M., Corona, C., Guillet, S., Poulain, V., Bekki, S., Guiot, J., Luckman, B. H., Oppenheimer, C., and Lebas, N.: Estimates of volcanic-induced cooling in the Northern Hemisphere over the past 1,500 years, Nat. Geosci., 8, 784–788, doi:10.1038/ngeo2526, 2015.

---

## Referee Comment (RC2) · Anonymous Referee #2 · 11 Jul 2017

With apologies for the delay in providing this review.

This paper attempts to provide a synthesis of palaeoclimate records spanning the last 2000 years in the Arctic region. In general, the content and subject matter are important and certainly well suited to Climates of the Past. However, prior to publication, I strongly recommend the authors undergo major revisions and resubmit their manuscript for further review.

Major comments

The content of this paper performs two broad functions. The first 23 pages provide a brief background to Arctic climate research, followed by a long review of the techniques

used to infer past climate variability in the Arctic. The second part of the paper consists of a synthesis of published hydroclimate reconstructions and model hindcasts for the region, spanning the last 2000 years. The paper's title and abstract only refer to the second component (the synthesis), thus the extremely long introductory review comes as some surprise. As a first step, I suggest that the authors consider cutting the paper in half and creating (1) a review of palaeoclimate techniques applied to the Arctic, and (2) a synthesis of the palaeoclimate data. With respect to (1), the authors must carefully consider whether this would represent a valuable addition to the literature beyond the several books and review papers on palaeoclimate techniques. However, in order to meet the objective described in the abstract, this paper needs to be shorter and more focused on the data synthesis.

With respect to the palaeoclimate synthesis, this section warrants a more detailed and systematic approach than is provided in the current manuscript. This systematic approach should include reverting to a more traditional journal article format, with an introduction, methods, results and discussion.

As a minimum, the methods section should provide a clear and detailed description of the process of identifying and screening the published records for the Arctic region, which is not satisfactorily clear. The results section should detail which records were considered, how many were included/excluded and for what reasons. The PAGES 2k network have provided very clear guidelines for this process, and the screening process is described briefly on pages 27-28, however a detailed description and summary is necessary in order for readers to appreciate how comprehensive the search has been. For example, are all records described here included in Ljungqvist et al. (2016)? If not, which additional records were included, and which were excluded?

Furthermore, the approach to deriving the new hydroclimate proxy synthesis, described perfunctorily on page 28, requires a much more detailed description and appraisal as is afforded here. In this respect, I have several questions which are not answered in the manuscript: (1) How was the age uncertainty in these records dealt with when deriving averages for the multiple records?; (2) How were the timesteps aligned in order to derive an average of the multiple records? Was this by linear interpolation or another approach? Were the data smoothed in any way, or binned? (3) The synthesis contains records that have an average sample resolution of <50 years, yet the resulting timeseries suggests variability at much higher frequencies – how is this possible? Is the synthesis weighted more heavily towards the annually resolved records? (4) The spatial coverage of records used is uneven, with certain regions being more heavily sampled than others. Of note, for example, are the several Greenland ice core records are included in the synthesis. How does the regional synthesis deal with the bias towards those heavily replicated regions? (5) Finally – I would argue it is misleading to state that the results generated here are 'not a reconstruction'. True, the hydroclimate timeseries isn't calibrated against a particular climate signal, however it is a qualitative reconstruction of relative hydroclimate variability in the Arctic. Generally speaking, given the proliferation of numerical approaches to deriving regional and global syntheses of time-uncertain palaeoclimate records (see for example Anchukaitis and Tierney, 2012, Climate Dynamics), there is considerable un-realised potential in this research that could (and should) be investigated in more detail. If more involved numerical approaches are deemed unsuitable, then some justification as to why must be given.

Related to the review of regional palaeoclimate records, I found the multiple plots of palaeoclimate timeseries (Figures 7-11) quite unhelpful, not least due to the variety of ways the data are plotted (including the use of various graphical styles and time axes being both vertical and horizontal). It would be much more helpful to view a smaller selection of these records in a single figure (maximum two if necessary) on a common timescale in order to assess the Arctic-wide synchronicity or otherwise. It would also be helpful to view the regional synthesis timeseries in comparison with the records from which it was derived, so the reader can get a feel for how certain records have influenced the synthesis.

Parts of the manuscript read well, however I would advise the authors ask a native

English speaker to proof-read the manuscript before resubmission.

Minor comments

Abstract: The abstract describes 'inadequate proxy data coverage' (Page 1, Line 37), yet then goes on to call for 'detailed regional studies, e.g. including field reconstructions' (P2, L3). How is the latter possible if there's inadequate data?

Section 2.2.1: I'm not entirely sure this section is necessary for this paper.

P4,L25: the Arctic's. Errors related to the articles (misuse or non-use of the and/or a) are frequent throughout the manuscript.

P5,L18: 'there are', not 'there is'; 'phenomenon, which also...'

P6,L7: This sentence could be worded better – e.g.

P24,L11: 'extensive' -> 'extensively'

P24, L11: 'Typically annual precipitation... have been the targets'. This is not a complete sentence.

P24, L15: 'Although potentially...' Also not a complete sentence, and what is meant b the records not being available – not published?

P24, L20: 'Towards the west'. The spatial context is very vague here – do you mean western Canada?

P24, L20: 'there seems to be'. Use of present tense. In next line, past tense is used. Ensure there's a consistent approach to tense (ideally use past when discussing past events) throughout.

P24, L23: 'All show...' What shows? Maybe better link up to previous sentence.

P24, L26: 'Several'. Be more specific here when reviewing records. How many have been published?

[Figure]

P24, L27: 'These..' merge with previous sentence.

P25, L9: 'A visual inspection. . .' As described above, it would be preferable to summarise what records exist before identifying those relevant to this synthesis.

P26, L25: By this point, it would be useful to refer to a figure with some data.

P27, L16: 'variability' typo

P27, L18: 'method outlined below'. As described above, it would be better to outline this in a proper methods section.

P28, first paragraph. As above, put this in the methods.

P28, L6: What is meant by 'even more important'?

P28, L9: What is meant be 'e.g. tendencies'

P28, L17: 'This signal is not a signal of precipitations' This sentence needs some attention.

P28, L22: The value of the Mann-Kendall test is not clear in this context.

P29, L3: Wavelet description. Unless you are using a non-standard wavelet package, I don't think it's necessary to provide such detail. That said, wavelet analyses are notoriously susceptible to errors related to unevenly spaced data – was this considered in your analysis?

P29, L16: 'To minimise the impact of the 1456-1485 CE event'. . . Please provide more justification as to why it was necessary to filter out this event, and on the effects of that decision.

P29, L20: Comparing the North Atlantic and Alaskan records to the 'global' analysis, which constitutes both regions. This (as far as I can tell) is a flawed comparison, since surely the North Atlantic subset will be most similar to the global record, since 12 of the 17 constituent records are from the North Atlantic.

P30, paragraph 2. Comparing models with palaeoclimate data. This is a very brief and one-dimensional comparison given the importance of models for future projection. Much more detail should be provided on the similarities/differences and what that means for either the validity of the models or the palaeo- data.

P30, L30. At some point you need to justify why this new synthesis is an improvement on than L16, or indeed why it is necessary beyond L16.

P31, L25: 'Quite flat' – a more scientific term could be used here.

P31, L26-27: I fail to see why the absence of calibration for the new record would have any effect on the trend or variability within the record. The units and range would change, but the pattern would be identical before and after calibration.

P32, L1: 'not fully capturing the observed changes in the latter half of the 20th century'. Have you considered that other (non-climate) anthropogenic activities, such as recovery from acid rain, nutrient deposition or other atmospheric transport of pollutants may have influenced the recent signal in some proxies?

P32, L12: 'this period did possibly undergo'. Mixed up nouns and verbs in this sentence – need to re-word.

P32, L19: The paragraph on seasonal effects would be better merged into the proceeding text and not afforded a separate subheading.

P33, L3: change 'unbalance' -> 'imbalance'

P33, L9: Here you list future recommendations. Why not include these ideas in the bullet points listed below?

P33, L15: Bullet point 1 is two points. Also, by listing all records identified and screened in the results section, you would clearly make the point about data suitability and availability.

P33, L19: 'Proper Arctic2k hydro database. I thought this was the point of this paper?

P33, L23: 'Field reconstruction' – I got the feeling from reading this paper that a field reconstruction isn't really feasible due to a lack of spatial data coverage.

P33, L25: Better collaboration between modellers and palaeo-data collectors is often called for. Can you be more specific as to what the two disciplines could do to improve collaboration?

Tables 1 and 2: Why do we need two tables here? Why not merge? Also, are these all the published records from the Arctic, or just those you could access?

Table 3: this is unnecessary. Just indicate which records are used in table 2.

Figures 1-5. Five figures here is too many. Boil them down to one or 2 most important.

Figures 7-11. See comment above.

Figures 12-13. Merge these figures to 1.

Figure 17. What are the red lines here? Best fit lines? If so, they don't appear to bisect the data as would be expected. Perhaps there's an issue?

Figure 18. It would be useful to map z scores, as is the case in the final synthesis. Also, I fear you may be over-interpreting the scale of the yellow-green change in Greenland in Fig. 18a – the range is just 0.2 hydroclimate index units (also explain what that unit actually is).

Figure 20. I'm not sure this figure is necessary.

With respect to the specific editorial guidelines:

Does the paper address relevant scientific questions within the scope of CP? YES

Does the paper present novel concepts, ideas, tools, or data? NO

Are substantial conclusions reached? NO

Are the scientific methods and assumptions valid and clearly outlined? NOT CLEARLY

[Figure]

OUTLINED

Are the results sufficient to support the interpretations and conclusions? NO

Is the description of experiments and calculations sufficiently complete and precise to allow their reproduction by fellow scientists (traceability of results)? NO

Do the authors give proper credit to related work and clearly indicate their own new/original contribution? YES

Does the title clearly reflect the contents of the paper? NO

Does the abstract provide a concise and complete summary? NO

Is the overall presentation well structured and clear? NO

Is the language fluent and precise? COULD BE IMPROVED

Are mathematical formulae, symbols, abbreviations, and units correctly defined and used? N/A

Should any parts of the paper (text, formulae, figures, tables) be clarified, reduced, combined, or eliminated? YES - see comments.

Are the number and quality of references appropriate? YES

Is the amount and quality of supplementary material appropriate? YES

---

## Author Comment (AC1) · 8 Aug 2017

We thank the two anonymous reviewers for their time reviewing our manuscript and their insightful comments and suggestions.

Below we provide responses to each of the reviewers' comments and indicate our plans to revise the manuscript. Author responses are shown in bold italics.

Referee #1

The overall quality of the discussion paper "Arctic hydroclimate variability during the last 2000 years – current understanding and research challenges" intended for the Special

[Figure]

Issue "Climate of the past 2000 years: global and regional syntheses", is good. The manuscript presents a substantial, thorough, and updated contribution on hydroclimate variability during the past 2 ka in the Arctic. The concepts, ideas, methods, and data from different climate archives are clearly presented. The results are discussed in an appropriate and balanced way, including appropriate references. The scientific results and conclusions are presented in a clear, concise, and well structured way. The number and quality of figures and tables are appropriate, as well as the English language.

Response: We thank the reviewer for these encouraging comments.

Specific comment: Please note that Lake Nerfloen listed in Table 2 is located in western Norway, not Northern (N) Norway.

Response: this has been corrected in the revised version

──────────────────────────

---

## Author Comment (AC2) · 8 Aug 2017

We would like to thank Dr. Kaufman and the data review team for their valuable comments on our manuscript cp-2017-34: "Arctic hydroclimate variability during the last 2000 years – current understanding and research challenges". Below are our responses to their queries.

Essential additions for this paper:

(1) Table 1: Add Data Citations for all of the proxy datasets listed in this table and shown in Fig 11. For those data not already in a public repository, submit essential

metadata and data, and add the corresponding Data Citations.

Response: In the updated version of the manuscript, "Data citations" will be added with the Data URL for all the series used in this study. All series except one (which we could not get hold of, and which will be removed from the study if it turns out to be impossible) have been uploaded and are now publicly available. Also, to give additional credit to those providing the data, we have added them as co-authors on the paper.

(2) Table 2: List the proxy climate time series shown in Figs 8, 9, and 10, along with a corresponding Data Citations. Add the original publication citations for each record in Table 2 (like in Table 1).

Response: OK

(3) Submit the time series of the resulting hydroclimate composites (Figs 14, 16 and 19) for archival and include the data citation.

Response: They will be made available prior to publication

Possibly essential, depending on source of the data:

(4) If the data shown in Fig 6 are based on chronologies already archived and easily accessible in the ITRDB, then all is well. If instead the chronologies from the ITRDB were detrended or otherwise modified by the authors, then those new chronologies must be submitted for archival as part of this study. Either way, please clarify the data methods used to create Fig 6.

Response: All chronologies included in Fig. 6 are freely available from the ITRDB. In order to make them comparable, they were all standardised in a similar fashion. We will clarify the methods used to standardise the tree-ring data used in Fig.6 as follows:

"The biologically induced age trend was removed from the TRW data through standardization with a cubic-smoothing spline with a 50% frequency cut off at 35 years (Cook and Peters, 1981). This detrending preserves annual to decadal scale variations in the

detrended tree-ring data. The resulting dimensionless indices were arithmetically aver-aged into single site chronologies. Variance changes arising from changes in sample replication over time were corrected (Frank et al., 2007). Resulting chronologies were truncated where the sample size dropped below 5 trees".

Since the chronologies standardised using the above described method are presently being used in another study, the standardised chronologies will not be archived until that study is finished. However, since free software and code to standardise tree-ring data are freely available, given this description the data used here are easily re-created. It should also be noted that the method of standardisation used depends on the purpose of the study and the tree-ring data itself, so the tree-ring data set described in this study is just one of many potential versions of it.

Recommended:

(5) Contrary to what is shown in Table 2, many of the records appear to have been taken from Ljungqvist et al.'s global compilation. Essential metadata that are needed for intelligent reuse of the data in new synthesis are missing from the Ljungqvist et al. compilation, which undermines a primary goal of the PAGES data stewardship activity. We strongly encourage the authors to use the opportunity of this synthesis paper to start with the original datasets and to submit a more complete set of metadata for archival. We note that the Ljungqvist et al. dataset is truncated at 850 AD (the time frame considered in their study). For the current study, the full time series should be used and archived.

Response: The data from Ljungqvist et al.'s series were chosen for the compilation of a high-resolution hydroclimate composite for the Arctic. A more complete set of metadata of the series used will be added to the revised version of the manuscript and will include: location, archive type, proxy measurement, dating control, time cover but also original reference and data citation. Even if the dataset is truncated at 850 AD, it is sufficient to have an overview of the expression of the major climatic period that

occurs during the last two millennium (i.e. the MCA and the LIA), and for this particular review we find that to be sufficient. As more data becomes available, we envisage that the analysed period will be extended back in time, but this is not within the scope of this paper.

───────────────────────────

---

## Author Comment (AC3) · 8 Aug 2017

**Response File – Climate of the Past Manuscript cp-2017-34 Fredrik Charpentier Ljungqvist's comments**

We thank Fredrik Charpentier Ljunqvist for his insightful comments and suggestions which will help to improve the manuscript. Below we provide responses the comments and indicate our plans to revise the manuscript. Author responses are shown in ***bold italics***.

This is a very well written – and very timely and important – article that I hope will be published speedily after only minor revision. It serves both as an excellent review article of the state-of-the-art knowledge about the hydroclimate signal in various hydroclimate proxy records from the Arctic/sub-Arctic region and at the same time presents new important findings leading the research forward. Because the article to a considerable extent discusses a recent article of mine (Ljungqvist et al. 2016), I have read it very carefully with great interest and found some minor things that the authors may want to correct or improve prior to final publication.

The article, with new additional proxy records, represents a clear improvement of the understanding of centennial-scale Arctic hydroclimate variability compared to Ljungqvist et al. (2016). The new reconstruction shows more variability during the Little Ice Age. Although this likely partly is because of new additional proxy records it may also be related to slightly different filtering techniques to extract centennial-scale variations.

***Response: Thank you for the positive comments and encouragement of the new synthesis presented in our paper.***

I have listed my comments below after page number and line number:

Abstract, line 29: To mention the Arctic amplification phenomenon in the introduction to the Abstract seems a bit out of place here in this article devoted to the study of long-term Arctic hydroclimate variations.

***Response: Given the influence of Arctic amplification on regional hydrology, we do feel that mentioning this already in the beginning of the abstract is appropriate.***

Page 2, line 8: It would be clearer to write "anthropogenic greenhouse gas emissions" here instead of the more vague "human activities".

***Response: changed as suggested***

Page 2, line 11: Add the reference Hind et al. (2016) to the discussion of Arctic amplification.

***Response: reference added***

Page 2, lines 29–30: Add Shi et al. (2012) to the list of references here.

***Response: reference added***

Page 6, lines 8–11: References should be provided to the statement that lake cryosphere has not changed during the past two millennia. I am not so sure that this statement is fully correct, a least not at all locations in the Arctic.

*Response: Reference will be provided and statement made more clear.*

Page 6, lines 29–30: These processes are partly dependent on the depth of the active layer. In regions with a depth active layer (e.g. permafrost regions with warm summers) it is less the case.

*Response: Noted, will be added in the revised version.*

Page 10, line 9: Southern Scandinavia is south of 60N and not in the Arctic. Better to write Central Scandinavia – a region that still is "less harsh" from an Arctic point of view.

*Response: changed as suggested*

Page 13, line 17: Also cite Borgmark and Wastegård (2008) here.

*Response: reference added*

Page 14, lines 12–14: Please, double-check the time periods here.

*Response: Yes, these will be revised (thanks for noting)*

Page 15, lines 24–25: Add the also very relevant references Esper et al. (2002), Schneider et al. (2015), and Stoffel et al. (2016) here.

*Response: reference added*

Page 15, line 26: Add Shi et al. (2012) to the list of references here.

*Response: reference added*

Page 17, line 14: Medieval Warm Period/Medieval Climate Anomaly – medieval is too vague and a different meaning (as a time period) in history than as a climate period.

*Response: Quite right, a mistake and now changed as suggested*

Page 18, line 11: Maybe it is worth to mention that a tree-line as high as 73N only occurs in parts of central Siberia (e.g. the Taimyr Peninsula)?

*Response: changed as suggested*

Page 22, line 31: "GISP-2" should be written "GISP2".

*Response: Yes, changed as suggested*

Page 22, line 32: "O" in "Ymer O" should be with upper case "O".

*Response: Indeed! Changed as suggested*

Page 23: line 2: Geirsdóttir with "ó".

*Response: changed as suggested*

Page 25, line 2: Why this two time periods for the MCA and LIA, respectively? Some motivation for the choice of time periods would be good. Most data for Fennoscandia indicates pretty old cold conditions during parts of the 12th century whereas many regions

appear to have been rather, or very, warm during the 10th century (which also seems to have been the warmest century of the MCA in the Northern Hemisphere).

*Response: Quite right, this will be changed in the revised version.*

Page 26, line 29: Add a reference to the new article by Helama et al. (2017) about the DACP here. The reference to Ljungqvist (2009) is wrong here: Ljungqvist is misspelt ("k" instead of "q") and it should be Ljungqvist (2010) – that discusses the DACP – and NOT Ljungqvist (2009) that does not do so.

*Response: You are completely right. Helama et al. (2017) added and the right Ljunqqvist paper cited (and so sorry about the misspelling)*

Page 26, line 30: The word "disturbed" is ambiguous and vague here. It was cold but in what other ways "disturbed" compared to other periods. Larger variability in the climate?

*Response:*

Page 27, line 16: The word "variability" is misspelt here.

*Response: corrected*

Page 28, line 3: "PAGES" should be written with upper case letters (e.g. PAGES and not Pages).

*Response: Ooops, changed as suggested*

Page 29, line 28: Also cite Schmidt et al. (2012) here.

*Response: reference added*

Line 31, 26: There is an error here: Ljungqvist et al. (2016) does NOT present a calibrated reconstruction. It is an uncalibrated index, ranging from –2 to +2, and with exceeding values truncated to –2 and +2, respectively. All values are standard deviations with respect to the mean of 1000–1899 CE. In some aspects, the approach in Ljungqvist et al. (2016) has some similarities with PDSI and other hydroclimate indices. So, in this respect it is no real differences between Ljungqvist et al. (2016) and the new hydroclimate index in this article.

*Response: Thank you for spotting this mistake. We have now amended this in the text.*

Page 42, line 8. "ans" should be "and".

*Response: changed as suggested*

Page 61, line 20 (and in numerous citations throughout the article): Weissbach should be Weißbach.

*Response: Found the ß symbol, so changed as suggested*

---

## Author Comment (AC4) · 8 Aug 2017

**Response File – Climate of the Past Manuscript cp-2017-34 reviewer #2 comments**

We thank the two anonymous reviewers for their time reviewing our manuscript and their insightful comments and suggestions.

Below we provide responses to each of the reviewers' comments and indicate our plans to revise the manuscript. Author responses are shown in ***bold italics***.

As seen below, referee #2 advocates a division of this paper into two separate ones. However, we hope that we have argued successfully for keeping the present structure, given that this is a review paper.

**Referee #2**

This paper attempts to provide a synthesis of palaeoclimate records spanning the last 2000 years in the Arctic region. In general, the content and subject matter are important and certainly well suited to Climates of the Past. However, prior to publication, I strongly recommend the authors undergo major revisions and resubmit their manuscript for further review.

***Response: We thank the reviewer for her/his detailed examination of our manuscript and the very insightful comments. It seems that the reviewer has possibly misunderstood the purpose of this paper, which is being a review paper, aiming at highlighting the methods used to investigate hydroclimate variability in the Arctic region, as well as providing an overview of existing understanding of it. If this is unclear, we will make sure to highlight it in both the abstract and introduction of the revised version.***

**Major comments**

The content of this paper performs two broad functions. The first 23 pages provide a brief background to Arctic climate research, followed by a long review of the techniques used to infer past climate variability in the Arctic. The second part of the paper consists of a synthesis of published hydroclimate reconstructions and model hindcasts for the region, spanning the last 2000 years. The paper's title and abstract only refer to the second component (the synthesis), thus the extremely long introductory review comes as some surprise. As a first step, I suggest that the authors consider cutting the paper in half and creating (1) a review of palaeoclimate techniques applied to the Arctic, and (2) a synthesis of the palaeoclimate data. With respect to (1), the authors must carefully consider whether this would represent a valuable addition to the literature beyond the several books and review papers on palaeoclimate techniques. However, in order to meet the objective described in the abstract, this paper needs to be shorter and more focused on the data synthesis.

***Response: As stated above, this paper is not a research article, but a review article. The reviewer is right that the various methods used are described elsewhere in the literature, but, as far as we know, not collectively, with an Arctic focus. Our aim with this review paper is to provide a holistic***

*understanding of the complexity of attempting to infer past hydroclimate variability across the whole Arctic (and even at single sites). To achieve that aim, the archives as well as methods need to be described, and examples of hydroclimate inferences given. Consequently, this article could be seen as a "mini textbook" of Arctic paleohydroclimate. Thus, dividing this paper into two separate papers would be not meaningful. However, we will focus the "introductory review", i.e. the archive description part, better in the revised version.*

With respect to the palaeoclimate synthesis, this section warrants a more detailed and systematic approach than is provided in the current manuscript. This systematic approach should include reverting to a more traditional journal article format, with an introduction, methods, results and discussion. As a minimum, the methods section should provide a clear and detailed description of the process of identifying and screening the published records for the Arctic region, which is not satisfactorily clear. The results section should detail which records were considered, how many were included/excluded and for what reasons. The PAGES 2k network have provided very clear guidelines for this process, and the screening process is described briefly on pages 27-28, however a detailed description and summary is necessary in order for readers to appreciate how comprehensive the search has been. For example, are all records described here included in Ljungqvist et al. (2016)? If not, which additional records were included, and which were excluded?

Furthermore, the approach to deriving the new hydroclimate proxy synthesis, described perfunctorily on page 28, requires a much more detailed description and appraisal as is afforded here. In this respect, I have several questions which are not answered in the manuscript: (1) How was the age uncertainty in these records dealt with when de- riving averages for the multiple records?; (2) How were the timesteps aligned in order to derive an average of the multiple records? Was this by linear interpolation or another approach? Were the data smoothed in any way, or binned? (3) The synthesis contains records that have an average sample resolution of <50 years, yet the resulting timeseries suggests variability at much higher frequencies – how is this possible? Is the synthesis weighted more heavily towards the annually resolved records? (4) The spatial coverage of records used is uneven, with certain regions being more heavily sampled than others. Of note, for example, are the several Greenland ice core records are included in the synthesis. How does the regional synthesis deal with the bias towards those heavily replicated regions? (5) Finally – I would argue it is misleading to state that the results generated here are 'not a reconstruction'. True, the hydroclimate timeseries isn't calibrated against a particular climate signal, however it is a qualitative reconstruction of relative hydroclimate variability in the Arctic. Generally speaking, given the proliferation of numerical approaches to deriving regional and global syntheses of time-uncertain palaeoclimate records (see for example Anchukaitis and Tierney, 2012, Climate Dynamics), there is considerable un-realised potential in this research that could (and should) be investigated in more detail. If more involved numerical approaches are deemed unsuitable, then some justification as to why must be given.

*Response: The aim of this particular exercise was to show the potential to derive higher-resolution hydroclimate information than provided by Ljunqvist et al. (centennial). See also the comments by Ljungqvist on this manuscript. It*

*is clear that the synthesis is biased, and this is partly the point: showing the uneven spatial representation leading to biases if the aim is to represent the whole Arctic.*

*Moreover, since at this point we do not intend to make a stand alone paper of the synthesis, we will not go into too many details here. Still, we will follow the reviewers' recommendations regarding clarification of methods and data used in the revised version. We do however, hope that a more thorough attempt to reconstruct past Arctic hydroclimate variability will be made as new records emerge. Hopefully, our paper can serve as an inspiration to that.*

Related to the review of regional palaeoclimate records, I found the multiple plots of palaeoclimate timeseries (Figures 7-11) quite unhelpful, not least due to the variety of ways the data are plotted (including the use of various graphical styles and time axes being both vertical and horizontal). It would be much more helpful to view a smaller selection of these records in a single figure (maximum two if necessary) on a common timescale in order to assess the Arctic-wide synchronicity or otherwise. It would also be helpful to view the regional synthesis timeseries in comparison with the records from which it was derived, so the reader can get a feel for how certain records have influenced the synthesis.

*Response: The figures are intended to highlight the nature of the regional hydroclimate information gained from different proxies (Figs. 7-10), as well as a regional comparison of a variety of hydroclimate proxies with different resolution (Fig. 11) in a review context. It would be possible to compile these into one of two figures, but we feel that this would be less meaningful. As, the figures are intended to highlight the various Arctic hydroclimate archives, and consequently we feel that it is better to show these as they are usually depicted. Thus, unless the editor objects, we will keep the figures as they are.*

Parts of the manuscript read well, however I would advise the authors ask a native English speaker to proof-read the manuscript before resubmission.

*Response: We will do that given the opportunity to revise the manuscript*

**Minor comments**

Abstract: The abstract describes 'inadequate proxy data coverage' (Page 1, Line 37), yet then goes on to call for 'detailed regional studies, e.g. including field reconstructions'

*Response: Yes, given the large regional hydroclimate differences within the Arctic, it would be more useful to focus on those regions that are presently well replicated rather than attempting a whole Arctic study, which would be regionally biased. We will clarify that in the revised version.*

(P2, L3). How is the latter possible if there's inadequate data?

*Response: See above comment*

Section 2.2.1: I'm not entirely sure this section is necessary for this paper.

*Response: Since this is a review paper, we feel that also mentioning the potential future impacts of hydroclimate changes are important to acknowledge, i.e. connecting the past to the future.*

P4,L25: the Arctic's. Errors related to the articles (misuse or non-use of the and/or a) are frequent throughout the manuscript.

*Response: Thanks, this will be corrected throughout*

P5,L18: 'there are', not 'there is'; 'phenomenon, which also: : :'

*Response: corrected*

P6,L7: This sentence could be worded better – e.g.

*Response: ???*

P24,L11: 'extensive' -> 'extensively'

*Response: corrected*

P24, L11: 'Typically annual precipitation: : : have been the targets'. This is not a complete sentence.

*Response:*

P24, L15: 'Although potentially: : :' Also not a complete sentence, and what is meant b the records not being available – not published?

*Response: Yes, this is an awkward sentence, which will be revised. Also, networks of these are not yet available.*

*We revised the sentence as follows: Presently, there are few published hydroclimate reconstructions using other proxies, although these proxies have the potential to produce records with high temporal resolution.*

P24, L20: 'Towards the west'. The spatial context is very vague here – do you mean western Canada?

*Response: Revised as follows: In western Canada and Alaska, there was an increase in precipitation during the past 2000 years, whereas a long-term decrease was seen towards the east.*

P24, L20: 'there seems to be'. Use of present tense. In next line, past tense is used. Ensure there's a consistent approach to tense (ideally use past when discussing past events) throughout.

*Response: thanks, this will be corrected throughout*

P24, L23: 'All show: : :' What shows? Maybe better link up to previous sentence.

*Response: thanks, will change to "They all show…"*

P24, L26: 'Several'. Be more specific here when reviewing records. How many have been published?

*Response: In this context we do not feel that it is not necessary to give the exact number since many of these are from locations below 60N, i.e. outside the PAGES 2k Arctic limit. The idea is to highlight the tradition of paleohydrological studies in this region, but if this is unclear, we will re-formulate this sentence in the revised version.*

P24, L27: 'These..' merge with previous sentence.

*Response: corrected*

P25, L9: 'A visual inspection: : :' As described above, it would be preferable to summarise what records exist before identifying those relevant to this synthesis.

*Response: These are the presently available records containing hydroclimate information from the Arctic part of Fennoscandia (see above) and they are presented in Table 1. We will revise this sentence so that this is clear.*

P26, L25: By this point, it would be useful to refer to a figure with some data.

*Response: They are shown in Fig 11, but we will refer to that figure in this sentence.*

P27, L16: 'variability' typo

*Response: corrected*

P27, L18: 'method outlined below'. As described above, it would be better to outline this in a proper methods section.

*Response: see comments above*

P28, first paragraph. As above, put this in the methods.

*Response: see comments above*

P28, L6: What is meant by 'even more important'?

*Response: This sentence has been changed to "This drastic selection is necessary to allow for comparison of data at centennial scales and facilitates the time series analyses*

P28, L9: What is meant be 'e.g. tendencies'

*Response: This sentence has been changed to "… offer the possibility to interpret hydroclimate variability in the Arctic from low to high frequencies."*

P28, L17: 'This signal is not a signal of precipitations' This sentence needs some attention.

*Response: This sentence has been changed to "This is not a signal of precipitation alone, but most likely combination of all processes related to the hydrological cycle"*

P28, L22: The value of the Mann-Kendall test is not clear in this context.

*Response: We think that the reason for using the M-K test is clearly stated (if that is what is referred to by the reviewer?)*

P29, L3: Wavelet description. Unless you are using a non-standard wavelet package, I don't think it's necessary to provide such detail. That said, wavelet analyses are notoriously susceptible to errors related to unevenly spaced data – was this considered in your analysis?

*Response: Agreed, the revised description of the wavelet will be less detailed.*

P29, L16: 'To minimise the impact of the 1456-1485 CE event': : : Please provide more justification as to why it was necessary to filter out this event, and on the effects of that decision.

*Response: This will be added to the revised version.*

P29, L20: Comparing the North Atlantic and Alaskan records to the 'global' analysis, which constitutes both regions. This (as far as I can tell) is a flawed comparison, since surely the North Atlantic subset will be most similar to the global record, since 12 of the 17 constituent records are from the North Atlantic.

*Response: Yes, that is completely true and also the reason form this exercise, as described in the opening sentence of this section. However, given appearances of the time series for the two regions (Fig. 16) in comparison to the "global" one (Fig. 14), this is already quite evident. We will remove Fig. 17 and briefly mention this in the revised text.*

P30, paragraph 2. Comparing models with palaeoclimate data. This is a very brief and one-dimensional comparison given the importance of models for future projection. Much more detail should be provided on the similarities/differences and what that means for either the validity of the models or the palaeo- data.

*Response: We agree that the one-dimensional comparison is brief, but spatial patterns and their temporal evolution over the past time is the main and the most important information that a grid reconstruction can convey. We*

*therefore compared the similarities/differences of the temporal evaluation between MCA and LIA in both the grid reconstruction and the model simulations. We then discussed the possible reasons that could cause the discrepancy of the different expression on the temporal evolution between the reconstruction and the models (See P30 L19-26).*

P30, L30. At some point you need to justify why this new synthesis is an improvement on than L16, or indeed why it is necessary beyond L16.

*Response: It is difficult to say which reconstruction is better. However, the new synthesis shows a shorter period of wet anomalies during the MCA, and the variance is much larger after ca 1200 CE. Given high heterogeneity of the spatial patterns of precipitation, the new synthesis provides a new hypothesis of the temporal evolution of the arctic precipitation after ca 1200 CE.*

P31, L25: 'Quite flat' – a more scientific term could be used here.

*Response: Agreed, the sentence will be changed to "This is in agreement with L16, albeit the new Arctic mean displays more variability during the LIA than L16*

P31, L26-27: I fail to see why the absence of calibration for the new record would have any effect on the trend or variability within the record. The units and range would change, but the pattern would be identical before and after calibration.

*Response: Depending on the trends of the included records, this could have a distinct impact on the reconstruction when fitted to observations during the calibration period.*

P32, L1: 'not fully capturing the observed changes in the latter half of the 20th century'. Have you considered that other (non-climate) anthropogenic activities, such as recovery from acid rain, nutrient deposition or other atmospheric transport of pollutants may have influenced the recent signal in some proxies?

*Response: Good point, this will be added to the revised version*

P32, L12: 'this period did possibly undergo'. Mixed up nouns and verbs in this sentence – need to re-word.

*Response: corrected*

P32, L19: The paragraph on seasonal effects would be better merged into the proceeding text and not afforded a separate subheading.

*Response: OK, changed*

P33, L3: change 'unbalance' -> 'imbalance'

*Response: corrected*

P33, L9: Here you list future recommendations. Why not include these ideas in the bullet points listed below?

***Response: Good point, thanks.***

P33, L15: Bullet point 1 is two points. Also, by listing all records identified and screened in the results section, you would clearly make the point about data suitability and availability.

***Response: Good point, thanks.***

P33, L19: 'Proper Arctic2k hydro database. I thought this was the point of this paper?

***Response: No, no such dedicated data base does yet exist.***

P33, L23: 'Field reconstruction' – I got the feeling from reading this paper that a field reconstruction isn't really feasible due to a lack of spatial data coverage.

***Response: There are potentials for some regions with good data coverage, e.g. Fennoscandia and parts of N America and possibly Greenland. But it is not possible for the whole Arctic.***

P33, L25: Better collaboration between modellers and palaeo-data collectors is often called for. Can you be more specific as to what the two disciplines could do to improve collaboration?

***Response: We will elaborate in the revised version***
Tables 1 and 2: Why do we need two tables here? Why not merge? Also, are these all the published records from the Arctic, or just those you could access?

***Response: We want to keep them separate because table 1 represents the data available for Fennoscandia and Table 2 the data used in the synthesis. These data are those that are available.***

Table 3: this is unnecessary. Just indicate which records are used in table 2.

***Response: Agreed***

Figures 1-5. Five figures here is too many. Boil them down to one or 2 most important.

***Response: We have replaced Fig. 1 by Fig. 3, and put Fig. 2, 4 and 5 together as Fig. 2.***

Figures 7-11. See comment above.

***Response: See comments above***

Figures 12-13. Merge these figures to 1.

*Response: Agreed*

Figure 17. What are the red lines here? Best fit lines? If so, they don't appear to bisect the data as would be expected. Perhaps there's an issue?

*Response: Fig. 17 will be removed (see response above)*

Figure 18. It would be useful to map z scores, as is the case in the final synthesis. Also, I fear you may be over-interpreting the scale of the yellow-green change in Greenland in Fig. 18a – the range is just 0.2 hydroclimate index units (also explain what that unit actually is).

*Response: Good point, we will map the z-scores instead.*

Figure 20. I'm not sure this figure is necessary.

*Response: Seasonality is an important issue in paleo climate reconstruction, since the archives may contain hydroclimate information for different seasons. So we chose to keep this figure.*

---

## Author Response (AR1)

**Response File – Climate of the Past Manuscript cp-2017-34**

We thank the reviewers for their time reviewing our manuscript and their insightful comments and suggestions.

Below we provide the original responses to each of the reviewers' comments that indicated our plans to revise the manuscript and subsequently the specific changes that we have made to the revised manuscript. Reviewer comments are shown in *italic*. Our original responses are shown in **bold** and the updated descriptions of how we have revised the manuscript are shown in **bold blue text**. Quotations from the original manuscript are shown in ***bold italics*** and quotations from the revised manuscript are shown in ***bold blue italics***.

**Reviewer #1**

*The overall quality of the discussion paper "Arctic hydroclimate variability during the last 2000 years – current understanding and research challenges" intended for the Special Issue "Climate of the past 2000 years: global and regional syntheses", is good. The manuscript presents a substantial, thorough, and updated contribution on hydroclimate variability during the past 2 ka in the Arctic. The concepts, ideas, methods, and data from different climate archives are clearly presented. The results are discussed in an appropriate and balanced way, including appropriate references. The scientific results and conclusions are presented in a clear, concise, and well structured way. The number and quality of figures and tables are appropriate, as well as the English language.*

**We thank the reviewer for these encouraging comments.**

*Specific comment: Please note that Lake Nerfloen listed in Table 2 is located in western Norway, not Northern (N) Norway.*

**This has been corrected in the revised version. Done**

**Reviewer #2**

*This paper attempts to provide a synthesis of palaeoclimate records spanning the last 2000 years in the Arctic region. In general, the content and subject matter are important and certainly well suited to Climates of the Past. However, prior to publication, I strongly recommend the authors undergo major revisions and resubmit their manuscript for further review.*

**We thank the reviewer for her/his detailed examination of our manuscript and the very insightful comments. It seems that the reviewer has possibly misunderstood the purpose of this paper, which is being a review paper, aiming at highlighting the methods used to investigate hydroclimate variability in the Arctic region, as well as providing an overview of existing understanding of it. If this is unclear, we will make sure to highlight it in both the abstract and introduction of the revised version.**

**We have now emphasised that this is a review paper, both in the abstract and in the introduction (see below).**

*Major comments*

*The content of this paper performs two broad functions. The first 23 pages provide a brief background to Arctic climate research, followed by a long review of the techniques used to infer past climate variability in the Arctic. The second part of the paper consists of a synthesis of published hydroclimate reconstructions and model hindcasts for the region, spanning the last 2000 years. The paper's title and abstract only refer to the second component (the synthesis), thus the extremely long introductory review comes as some surprise. As a first step, I suggest that the authors consider cutting the paper in half and creating (1) a review of palaeoclimate techniques applied to the Arctic, and (2) a synthesis of the palaeoclimate data. With respect to (1), the authors must carefully consider whether this would represent a valuable addition to the literature beyond the several books and review papers on palaeoclimate techniques. However, in order to meet the objective described in the abstract, this paper needs to be shorter and more focused on the data synthesis.*

**As stated above, this paper is not a research article, but a review article. The reviewer is right that the various methods used are described elsewhere in the literature, but, as far as we know, not collectively, with an Arctic focus. Our aim with this review paper is to provide a holistic understanding of the complexity of attempting to infer past hydroclimate variability across the whole Arctic (and even at single sites). To achieve that aim, the archives as well as methods need to be described, and examples of hydroclimate inferences given. Consequently, this article could be seen as a "mini textbook" of Arctic palaeohydroclimate. Thus, in that respect, dividing this paper into two separate papers would be not meaningful. However, we will focus the "introductory review", i.e. the archive description part, better in the revised version.**

**We have emphasised that this is a review in both the abstract:**

**"*The aim of this review is to summarise* the current understanding of Arctic hydroclimate during the past 2000 years. *First, this paper reviews the main natural archives and proxies used to infer past hydroclimate variations in this remote region, and outlines the difficulty of disentangling the temperature from the moisture signal in these records. Second, a comparison of two sets of hydroclimate records covering the Common Era from two data-rich regions, North America and Fennoscandia, reveals inter- and intra-regional differences. Third, building on earlier work, this paper shows the potential for providing a high-resolution hydroclimate reconstruction for the Arctic and a comparison with last-millennium simulations from fully-coupled climate models.".***

***…" Finally, this review illustrates that the proxy data regional coverage is inadequate, with distinct data gaps in most of Eurasia and parts of North America, making robust assessments for the whole Arctic impossible at the present. Given the heterogeneity of Arctic hydroclimate variations, we recommend detailed regional studies, rather than the entire Arctic region."***

 **and in the introduction:**

**The aim of this *review* is to summarise the current understanding of Arctic hydroclimate, focusing on the last two millennia. *The paper uses the PAGES 2k* definition of the Arctic, i.e. the region north of 60°N. Section 2, briefly presents the current state and a future outlook of Arctic hydroclimate and impacts, from observations and climate model**

*simulations from the Coupled Model Intercomparison Project Phase 5 (CMIP5, Taylor et al., 2012). Section 3 reviews the various archives and proxies used to derive information on past hydroclimate variability. Section 4 presents multiproxy comparisons of hydroclimate variability in Arctic Canada and Fennoscandia, two regions with denser networks of sites. In section 5, a new compilation of Arctic hydroclimate data, which illustrates the potential to derive higher temporal resolution than that of Ljungqvist et al. (2016), is compared to model simulations from the third Paleoclimate Modelling Intercomparison Project Phase III (PMIP3, Braconnot et al., 2012). The current understanding of Arctic hydroclimate during the last 2k is summarized in section 6, and some recommendations for future work are given in section 7.*

**We have also tried to streamline (shorten and sharpen) the text to make it more readable, as well as adding some additional relevant references.**

*With respect to the palaeoclimate synthesis, this section warrants a more detailed and systematic approach than is provided in the current manuscript. This systematic approach should include reverting to a more traditional journal article format, with an introduction, methods, results and discussion. As a minimum, the methods section should provide a clear and detailed description of the process of identifying and screening the published records for the Arctic region, which is not satisfactorily clear. The results section should detail which records were considered, how many were included/excluded and for what reasons. The PAGES 2k network have provided very clear guidelines for this process, and the screening process is described briefly on pages 27-28, however a detailed description and summary is necessary in order for readers to appreciate how comprehensive the search has been. For example, are all records described here included in Ljungqvist et al. (2016)? If not, which additional records were included, and which were excluded?*

*Furthermore, the approach to deriving the new hydroclimate proxy synthesis, described perfunctorily on page 28, requires a much more detailed description and appraisal as is afforded here. In this respect, I have several questions which are not answered in the manuscript: (1) How was the age uncertainty in these records dealt with when de- riving averages for the multiple records?; (2) How were the timesteps aligned in order to derive an average of the multiple records? Was this by linear interpolation or another approach? Were the data smoothed in any way, or binned? (3) The synthesis contains records that have an average sample resolution of <50 years, yet the resulting timeseries suggests variability at much higher frequencies – how is this possible? Is the synthesis weighted more heavily towards the annually resolved records? (4) The spatial coverage of records used is uneven, with certain regions being more heavily sampled than others. Of note, for example, are the several Greenland ice core records are included in the synthesis. How does the regional synthesis deal with the bias towards those heavily replicated regions? (5) Finally – I would argue it is misleading to state that the results generated here are 'not a reconstruction'. True, the hydroclimate timeseries isn't calibrated against a particular climate signal, however it is a qualitative reconstruction of relative hydroclimate variability in the Arctic. Generally speaking, given the proliferation of numerical approaches to deriving regional and global syntheses of time-uncertain palaeoclimate records (see for example Anchukaitis and Tierney, 2012, Climate Dynamics), there is considerable un-realised potential in this research that could (and should) be investigated in more detail. If more involved numerical approaches are deemed unsuitable, then some justification as to why must be given.*

The aim of this particular exercise was to show the potential to derive higher-resolution hydroclimate information than provided by Ljunqvist et al. (centennial). See also the comments by Ljungqvist on this manuscript. It is clear that the synthesis is biased, and this is partly the point: showing the uneven spatial representation leading to biases if the aim is to represent the whole Arctic.

Moreover, since at this point we do not intend to make a stand alone paper of the synthesis, we will not go into too many details here. Still, we will follow the reviewers' recommendations regarding clarification of methods and data used in the revised version. We do however, hope that a more thorough attempt to reconstruct past Arctic hydroclimate variability will be made as new records emerge. Hopefully, our paper can serve as an inspiration to that.

The hydroclimatic record corresponds to an average of all the series used. It was calculated taking into account the different time step of the series used (2). The resulting mean record suggest high frequency variability because it corresponds to a year per year average. Since all series are not at the annual resolution, a curve of the number of series per year has been added. It will be able to determine the temporal representatively of the regional mean record (4). Because several records have an average sample resolution > 50 years, the frequency variability inferior of this time step were not interpreted (3).

In the revised version, the description of the compilation has been changed, highlighting our intent to make a qualitative compilation of some of the available and previously used data. There are two main messages: 1) it is possible to get more high-resolution variability than provided by L16, which will slightly alter the composite hydroclimate evolution through time (e.g. Fig. 13), and 2) This particular composite (and indeed that from Ljungqvist et al.) is biased towards the data-rich North Atlantic sector. It is quite clear that none of these two "compilations" actually represents the whole Arctic. However, during the process of putting together this review more data has become available (e.g. Fennoscandia) and if more data become available (N America and Russia) there will be potentials for reconstructions with better whole-Arctic representation. Nevertheless, this is not the scope of this review.

*Related to the review of regional palaeoclimate records, I found the multiple plots of palaeoclimate timeseries (Figures 7-11) quite unhelpful, not least due to the variety of ways the data are plotted (including the use of various graphical styles and time axes being both vertical and horizontal). It would be much more helpful to view a smaller selection of these records in a single figure (maximum two if necessary) on a common timescale in order to assess the Arctic-wide synchronicity or otherwise. It would also be helpful to view the regional synthesis timeseries in comparison with the records from which it was derived, so the reader can get a feel for how certain records have influenced the synthesis.*

The figures are intended to highlight the nature of the regional hydroclimate information gained from different proxies (Figs. 7-10), as well as a regional comparison of a variety of hydroclimate proxies with different resolution (Fig. 11) in a review context. It would be possible to compile these into one of two figures, but we feel that this would be less meaningful. As, the figures are intended to highlight the various Arctic hydroclimate archives, and consequently we feel that it is better to show these as

**they are usually depicted. Thus, unless the editor objects, we will keep the figures as they are.**

**As above.**

*Parts of the manuscript read well, however I would advise the authors ask a native English speaker to proof-read the manuscript before resubmission.*

**We will do that given the opportunity to revise the manuscript.**

**The native English speaking co-authors have now gone through the manuscript in detail**

*Minor comments*

*Abstract: The abstract describes 'inadequate proxy data coverage' (Page 1, Line 37), yet then goes on to call for 'detailed regional studies, e.g. including field reconstructions'*

**Yes, given the large regional hydroclimate differences within the Arctic, it would be more useful to focus on those regions that are presently well replicated rather than attempting a whole Arctic study, which would be regionally biased. We will clarify that in the revised version.**

**We have changed the last sentences to:** *Finally, this review illustrates that the proxy data regional coverage is inadequate, with distinct data gaps in most of Eurasia and parts of North America, making robust assessments for the whole Arctic impossible at the present. Given the heterogeneity of Arctic hydroclimate variations, we recommend detailed regional studies, rather than the entire Arctic region. Also,* **field reconstructions** *are useful to disentangle spatial patterns and potential forcing factors.* **However, at** *present, it is only possible to carry out regional syntheses for a few regions, e.g. Fennoscandia, Greenland and western North America. To fully assess pan-Arctic hydroclimate variability for the last two millennia additional proxy records are required.*

*(P2, L3). How is the latter possible if there's inadequate data?*

**See above comment.**

**We hope that this is clearer now.**

*Section 2.2.1: I'm not entirely sure this section is necessary for this paper.*

**Since this is a review paper, we feel that also mentioning the potential future impacts of hydroclimate changes are important to acknowledge, i.e. connecting the past to the future.**

**We kept this section**

*P4,L25: the Arctic's. Errors related to the articles (misuse or non-use of the and/or a)*

*are frequent throughout the manuscript.*

**Thanks, this will be corrected throughout. Done**

P5,L18: 'there are', not 'there is'; 'phenomenon, which also: : :'

**Corrected. Done**

P6,L7: This sentence could be worded better – e.g.

**???**

**Sentence changed to:** *Hence, their ages, and the potential lengths of the records they contain, are diverse across the Arctic, ranging from the entire Holocene in Beringia and Scandinavia, to a few hundred years in Greenland or Iceland.*

*P24,L11: 'extensive' -> 'extensively'*

**Corrected. Done**

*P24, L11: 'Typically annual precipitation: : : have been the targets'. This is not a complete sentence.*

**This has been changed to:** *Typically annual precipitation, along with temperature, or lake levels have been the targets for reconstructions*

*P24, L15: 'Although potentially: : :' Also not a complete sentence, and what is meant b the records not being available – not published?*

**Yes, this is an awkward sentence, which will be revised. Also, networks of these are not yet available. We revised the sentence as follows:** *Presently, there are few published hydroclimate reconstructions using other proxies, although these proxies have the potential to produce records with high temporal resolution. However, networks of these are not yet available.*

*P24, L20: 'Towards the west'. The spatial context is very vague here – do you mean western Canada?*

**Revised as follows:** *In western Canada and Alaska, there was an increase in precipitation during the past 2000 years, whereas a long-term decrease was seen towards the east.*

*P24, L20: 'there seems to be'. Use of present tense. In next line, past tense is used. Ensure there's a consistent approach to tense (ideally use past when discussing past events) throughout.*

**Thanks, this will be corrected throughout: Done**

*P24, L23: 'All show: : :' What shows? Maybe better link up to previous sentence.*

**Thanks, we will change to "They all show…" Done**

*P24, L26: 'Several'. Be more specific here when reviewing records. How many have been published?*

**In this context we do not feel that it is not necessary to give the exact number since many of these are from locations below 60N, i.e. outside the PAGES 2k Arctic limit. The idea is to highlight the tradition of paleohydrological studies in this region, but if this is unclear, we will re-formulate this sentence in the revised version.**

**Changed text in the revised version:** *In Fennoscandia,  palaeolimnological studies have recently produced records indicative of past regional hydroclimatic variability.  Such records are based on micro-, macro- and megafossil assemblages, in addition to lithological data. Here we use sixteen palaeolimnological records from the Arctic region (Table 2, Fig. 11) to illustrate hydroclimatic shifts and variations in Fennoscandia over the Common Era*

*P24, L27: 'These..' merge with previous sentence.*

**Corrected. Done**

*P25, L9: 'A visual inspection: : :' As described above, it would be preferable to summarise what records exist before identifying those relevant to this synthesis.*

**These are the presently available records containing hydroclimate information from the Arctic part of Fennoscandia (see above) and they are presented in Table 2. We will revise this sentence so that this is clear.**

**A visual inspection** *of the records shown in Fig. 11…*

*P26, L25: By this point, it would be useful to refer to a figure with some data.*

**They are shown in Fig 11, but we will refer to that figure in this sentence. See above**

*P27, L16: 'variability' typo*

**Corrected Done**

*P27, L18: 'method outlined below'. As described above, it would be better to outline this in a proper methods section.*

**See comments above**

**We have now stressed that we are making a synthesis (qualitative rather than quantitative): "***Here a new synthesis of Arctic hydroclimate variability extending back to 800 CE is performed, utilizing both high-and low-resolution records. Note that this is not a quantitative reconstruction, but only provides a qualitative view of relative hydroclimate variability in the Arctic. The aim is to assess the potential to derive an Arctic hydroclimate record with more high-frequency information than that derived for the same region from the results of Ljungqvist et al. (2016)***"**

*P28, first paragraph. As above, put this in the methods.*

**See comments above**

**We have rearranged the text in the paragraphs describing the methods used so that it makes more sense to the reader.**

*P28, L6: What is meant by 'even more important'?*

**This sentence has been changed to "*This drastic selection is necessary to allow for comparison of data at centennial scales and facilitates the time series analyses*"**

*These strict selection criteria are necessary to allow for comparison of data at centennial scales and facilitate time series analysis*

*P28, L9: What is meant be 'e.g. tendencies'*

**This sentence has been changed to** "… *offer the possibility to interpret hydroclimate variability in the Arctic from low to high frequencies.*"

*P28, L17: 'This signal is not a signal of precipitations' This sentence needs some attention.*

**This sentence has been changed to "*This is not a signal of precipitation alone, but most likely combination of all processes related to the hydrological cycle*" Done**

*P28, L22: The value of the Mann-Kendall test is not clear in this context.*

**We think that the reason for using the M-K test is clearly stated (if that is what is referred to by the reviewer?)**

**The M-K test was used to analyse the long-term trends in the records**

*P29, L3: Wavelet description. Unless you are using a non-standard wavelet package, I don't think it's necessary to provide such detail. That said, wavelet analyses are notoriously susceptible to errors related to unevenly spaced data – was this considered in your analysis?*

**Agreed, the revised description of the wavelet will be less detailed. Done**

*P29, L16: 'To minimise the impact of the 1456-1485 CE event': : : Please provide more justification as to why it was necessary to filter out this event, and on the effects of that decision.*

**This will be added to the revised version.**

**In the revision we state: "*Because wavelet analysis is sensitive to large events that may hide the lowest frequencies recorded, the 1456-1485 CE event was extracted by wavelet filtering and the signal reconstructed by inverse Fourier transform before using CWT.*"**

*P29, L20: Comparing the North Atlantic and Alaskan records to the 'global' analysis, which constitutes both regions. This (as far as I can tell) is a flawed comparison, since surely the North Atlantic subset will be most similar to the global record, since 12 of the 17 constituent records are from the North Atlantic.*

**Yes, that is completely true and also the reason form this exercise, as described in the opening sentence of this section. However, given appearances of the time series for the two regions (Fig. 16) in comparison to the "global" one (Fig. 14), this is already quite evident. We will remove Fig. 17 and briefly mention this in the revised text.**

**Figure 17 was removed**

*P30, paragraph 2. Comparing models with palaeoclimate data. This is a very brief and one-dimensional comparison given the importance of models for future projection. Much more detail should be provided on the similarities/differences and what that means for either the validity of the models or the palaeo- data.*

**We agree that the one-dimensional comparison is brief, but spatial patterns and their temporal evolution over the past time is the main and the most important information that a grid reconstruction can convey. We therefore compared the similarities/differences of the temporal evaluation between MCA and LIA in both the grid reconstruction and the model simulations. We then discussed the possible reasons that could cause the discrepancy of the different expression on the temporal evolution between the reconstruction and the models.**

**We have made some changes on this section, partly based on the new figure 12, including a statement regarding the comparison of differences in proxies and models across the MCA-LIA:** *Caution needs to be advised, since the magnitudes of the differences in proxy-derived hydroclimate between the MCA and LIA is consistently larger than in the model ensemble mean or in the individual model simulations. However, the discrepancy between models and proxies may imply that the changes in spatial hydroclimate patterns from the MCA to the LIA over Fennoscandia and Greenland are not related to changes in external forcings, but possibly by internal variability.*

**However, given the scope of the paper, as well as the scarceness of the proxy data, we don't want to elaborate too much on this topic, which is clearly worth a study of its own.**

*P30, L30. At some point you need to justify why this new synthesis is an improvement on than L16, or indeed why it is necessary beyond L16.*

**It is difficult to say which synthesis is better. However, the new synthesis shows a shorter period of wet anomalies during the MCA, and the variance is much larger after ca 1200 CE. Given high heterogeneity of the spatial patterns of precipitation, the new synthesis provides a new hypothesis of the temporal evolution of the arctic precipitation after ca 1200 CE.**

**Moreover, it is clear that the new synthesis much more variability at centennial timescales after ca 1500 CE than L16, which is more in lie with the model ensemble mean. A brief comment on this has been added to section 6.2:** *At least from the LIA and*

*onwards, there is a better agreement between the model ensemble mean and the new synthesis than with L16.* **Both Arctic hydroclimate records derived from L16 and the composite presented here are,** *however,* **insufficient for drawing any firm conclusions for the whole region.**

*P31, L25: 'Quite flat' – a more scientific term could be used here.*

**Agreed, the sentence will be changed to "This is in agreement with L16, albeit the new Arctic mean displays more variability during the LIA than L16"**

**Changed to:** *The new Arctic hydroclimate mean synthesis (Fig 13) suggests drying during the MCA, but wet conditions in the early part of the LIA and drier conditions in the latter part. This is largely in agreement with L16, although the latter shows less variability during the mainly dry LIA*

*P31, L26-27: I fail to see why the absence of calibration for the new record would have any effect on the trend or variability within the record. The units and range would change, but the pattern would be identical before and after calibration.*

**Depending on the trends of the included records, this could have a distinct impact on the reconstruction when fitted to observations during the calibration period.** **Yes**

*P32, L1: 'not fully capturing the observed changes in the latter half of the 20th century'. Have you considered that other (non-climate) anthropogenic activities, such as recovery from acid rain, nutrient deposition or other atmospheric transport of pollutants may have influenced the recent signal in some proxies?*

**Good point, this will be added to the revised version**

**We removed this part of the text during the revision**

**P32, L12: 'this period did possibly undergo'. Mixed up nouns and verbs in this sentence – need to re-word.**

**Corrected: "…** *this*  *was* *possibly*  *a period of* *noticeable* *climatic fluctuations*."

*P32, L19: The paragraph on seasonal effects would be better merged into the proceeding text and not afforded a separate subheading.*

**OK, changed** **Done**

*P33, L3: change 'unbalance' -> 'imbalance'*

**Corrected** **Done**

*P33, L9: Here you list future recommendations. Why not include these ideas in the bullet points listed below?*

**Good point, thanks.**

*Actually, this is recommended in bullet point 3 (now 4): Consolidate data, and even attempt to make a field reconstruction, for regions with sufficient number of hydroclimate proxy records in time and space. Presently there seems to be opportunities for a cross-Atlantic study, which may shed light onto observed regional hydroclimate patterns and the mechanisms behind those*

*P33, L15: Bullet point 1 is two points. Also, by listing all records identified and screened in the results section, you would clearly make the point about data suitability and availability.*

**Good point, thanks.**

- *Increase the spatial coverage of hydroclimate proxies. This is particularly important for Eurasia (except Fennoscandia), but especially North America.*
- *There are several hydroclimate records that would add valuable information which are not publicly available, so it is important to encourage palaeoclimate researchers to share their data.*

**Still, we don't feel that there is a need to again list the suitable/available records in this section. We do note, however, that in connection with this review, several new records have been made available:** *It is encouraging that several new hydroclimate records have been made available during the process of preparing this review (see Table 2)*

*P33, L19: 'Proper Arctic2k hydro database. I thought this was the point of this paper?*

**No, no such dedicated data base does yet exist**. **We do hope that this review paper can lead to the creation of such a dedicated data base.**

*P33, L23: 'Field reconstruction' – I got the feeling from reading this paper that a field reconstruction isn't really feasible due to a lack of spatial data coverage.*

**There are potentials for some regions with good data coverage, e.g. Fennoscandia and parts of N America and possibly Greenland. But it is not possible for the whole Arctic.**

**This is stated in section 6.2:** *However, due to the low number of available hydroclimate proxy records from the Arctic, and the* imbalance *in spatial coverage* (Table 3, Fig. 12), it is currently impossible to prepare *a field reconstruction* for the whole region*.*

*P33, L25: Better collaboration between modellers and palaeo-data collectors is often called for. Can you be more specific as to what the two disciplines could do to improve collaboration?*

**We will elaborate in the revised version**

**Since there is a paper in this CP special issue dedicated to comparison of proxy and model estimates of hydroclimate variability, we refer to that paper rather than adding**

**too much text to this particular bullet point:** *Closer collaboration with the palaeoclimate modelling community. From the comparison between the existing "observational" data (reanalysis and proxies) and climate model simulations, discrepancies* as well as similarities *in both rate and spatial distribution were evident, and this needs to be addressed (Smerdon et al. 2017).*

*Tables 1 and 2: Why do we need two tables here? Why not merge? Also, are these all the published records from the Arctic, or just those you could access?*

**We want to keep them separate because table 1 represents the data available for Fennoscandia and Table 2 the data used in the synthesis. These data are those that are available.**

**We have now added a table presenting the N American data (Table 1). Table 2 describes the Fennoscandian data, and 3 the data for the synthesis.**

*Table 3: this is unnecessary. Just indicate which records are used in table 2.*

**Agreed. Done (used record are given in bold in new table 3)**

*Figures 1-5. Five figures here is too many. Boil them down to one or 2 most important.*

**We have replaced Fig. 1 by Fig. 3, and put Fig. 2, 4 and 5 together as Fig. 2. Done**

*Figures 7-11. See comment above.*

**See comments above OK**

*Figures 12-13. Merge these figures to 1.*

**Agreed. Done (new figure 8)**

**Figure 17. What are the red lines here? Best fit lines? If so, they don't appear to bisect the data as would be expected. Perhaps there's an issue?**

**Fig. 17 will be removed (see response above). Done**

*Figure 18. It would be useful to map z scores, as is the case in the final synthesis. Also, I fear you may be over-interpreting the scale of the yellow-green change in Greenland in Fig. 18a – the range is just 0.2 hydroclimate index units (also explain what that unit actually is).*

**Good point, we will map the z-scores instead.**

**We mapped the z-scores and also revised the interpretation of the proxy-based Greenland precipitation**

*Figure 20. I'm not sure this figure is necessary.*

**Seasonality is an important issue in paleo climate reconstruction, since the archives may contain hydroclimate information for different seasons. So we chose to keep this figure.**

**Figure 20 (now Fig. 14) is kept.**

*Data review team*
*Essential additions for this paper:*
*(1) Table 1: Add Data Citations for all of the proxy datasets listed in this table and shown in Fig 11. For those data not already in a public repository, submit essential metadata and data, and add the corresponding Data Citations.*

**In the updated version of the manuscript, "Data citations" will be added with the Data URL for the series used in this study.**

**This has now been done in new tables 1 (North America) and 2 (former table 1: Fennoscandia). Note that one dataset is not yet accessible due to the PI being out on an oceanographic expedition. The record will, however, be uploaded as soon as she returns ashore.**

*(2) Table 2: List the proxy climate time series shown in Figs 8, 9, and 10, along with a corresponding Data Citations. Add the original publication citations for each record in Table 2 (like in Table 1).*

**In the updated version of the manuscript, "Data citations" will be added with the Data URL for the series used in this study.**

**The data in Fig 4 (old Fig. 8) is already in Table 2 (Gunnarson 2008 + URL), but we have added the link to the figure caption. Data citations for N American data in Figs. 5-6 (old figs. 9-10 is) are in the new table 1.**

*(3) Submit the time series of the resulting hydroclimate composites (Figs 14, 16 and 19) for archival and include the data citation.*

**The regional curves obtained will be published online after the publication of this article.**

**The curves are now available online: https://figshare.com/articles/Global_North_Atlantic_Alaska_synthesis_record_txt/5502199**

*Possibly essential, depending on source of the data:*
*(4) If the data shown in Fig 6 are based on chronologies already archived and easily accessible in the ITRDB, then all is well. If instead the chronologies from the ITRDB were detrended or otherwise modified by the authors, then those new chronologies must be submitted for archival as part of this study. Either way, please clarify the data methods used to create Fig 6.*

**We will clarify the methods used to standardise the tree-ring data used in Fig.6 as follows: "The biologically induced age trend was removed from the TRW data through standardization with a cubic-smoothing spline with a 50% frequency cutoff at 35 years (Cook and Peters, 1981). This detrending preserves annual to decadal scale variations in the detrended tree-ring data. The resulting dimensionless indices were arithmetically**

averaged into single site chronologies.  Variance changes arising from changes in sample replication over time were corrected (Frank et al., 2007). Resulting chronologies were truncated where the sample size dropped below 5 trees".

Since the chronologies standardised using the above described method are presently being used in another study, the standardised chronologies will not be archived until that study is finished. However, since free software and code to standardise tree-ring data are freely available, given this description the data used here are easily recreated. It should also be noted that the method of standardisation used depends on the purpose of the study and the tree-ring data itself, so the tree-ring data set described in this study is just one of many potential versions of it.

Given the brief discussion related to Fig. 6, and that this information has previously been published, we have now omitted figure 6 and instead refer to St George and Ault 2014 (see their Fig. 1) which basically provides the same information.

*Recommended:*
*(5) Contrary to what is shown in Table 2, many of the records appear to have been taken from Ljungqvist et al.'s global compilation.  Essential metadata that are needed for intelligent reuse of the data in new synthesis are missing from the Ljungqvist et al. compilation, which undermines a primary goal of the PAGES data stewardship activity. We strongly encourage the authors to use the opportunity of this synthesis paper to start with the original datasets and to submit a more complete set of metadata for archival. We note that the Ljungqvist et al. dataset is truncated at 850 AD (the time frame considered in their study). For the current study, the full time series should be used and archived.*

The used of Ljungqvist et al.'s series was chosen to access to a high resolution hydroclimate composite for the Arctic. A more complete set of metadata of the serie used will be added in the new version of the manuscript and will include: location, archive type, proxy measurement, dating control, time cover but also original reference and data citation. Even if the dataset is truncated at 850 AD, it is sufficient to have an overview of the expression of the major climatic period that occurs during the last two millennium (i.e. the Medieval Climate Anomaly (950-1250 AD) and the Little Ice Age (1450-1850 AD)).

We keep the time frame given that this exercise is to show the potential of deriving a reconstruction with higher resolution than L16. Still, it is clear that such a reconstruction will not represent the full Arctic: when more data becomes available this will more likely be feasible. The original data sources are now given, as well as data availability. Moreover, the new composites are also available (see above).

*Fredrik Charpentier Ljungqvist's comments*
*This is a very well written – and very timely and important – article that I hope will be published speedily after only minor revision. It serves both as an excellent review article of the state-of-the-art knowledge about the hydroclimate signal in various hydroclimate proxy records from the Arctic/sub-Arctic region and at the same time presents new important findings leading the research forward. Because the article to a considerable extent discusses a recent article of mine (Ljungqvist et al. 2016), I have read it very carefully with great interest and found some minor things that the authors may want to correct or improve prior to final publication.*

*The article, with new additional proxy records, represents a clear improvement of the understanding of centennial-scale Arctic hydroclimate variability compared to Ljungqvist et al. (2016). The new reconstruction shows more variability during the Little Ice Age. Although this likely partly is because of new additional proxy records it may also be related to slightly different filtering techniques to extract centennial-scale variations.*

**Thank you for the positive comments and encouragement of the new synthesis presented in our paper.**

*I have listed my comments below after page number and line number:*

*Abstract, line 29: To mention the Arctic amplification phenomenon in the introduction to the Abstract seems a bit out of place here in this article devoted to the study of long-term Arctic hydroclimate variations*

**Given the influence of Arctic amplification on regional hydrology, we do feel that mentioning this already in the beginning of the abstract is appropriate.**

*Page 2, line 8: It would be clearer to write "anthropogenic greenhouse gas emissions" here instead of the more vague "human activities".*

**Changed as suggested. Done**

*Page 2, line 11: Add the reference Hind et al. (2016) to the discussion of Arctic amplification.*

**Reference added Done**

*Page 2, lines 29–30: Add Shi et al. (2012) to the list of references here.*

**Reference added Done**

*Page 6, lines 8–11: References should be provided to the statement that lake cryosphere has not changed during the past two millennia. I am not so sure that this statement is fully correct, a least not at all locations in the Arctic.*

**Reference will be provided and statement made more clear.**

**We changed this sentence to:** ***The last 2000 years were in general characterized by small glaciers advances, but with regional differences, prior to the general melt of the recent decades (Solomina et al., 2016). This relative stability in surface area over the last two millenia makes lakes excellent recorders of hydroclimate variability for this period.***

Page 6, lines 29–30: These processes are partly dependent on the depth of the active layer. In regions with a depth active layer (e.g. permafrost regions with warm summers) it is less the case.

**We have decided not to go into further elaboration on this particular issue, but revised the sentence as follows:** *During summer, the permafrost* *thaws* *in its upper part (active layer), leaving sediment easily mobilised by* *small* *amount**s** of rainfall.*

*Page 10, line 9: Southern Scandinavia is south of 60N and not in the Arctic. Better to write Central Scandinavia – a region that still is "less harsh" from an Arctic point of view.*

**Changed as suggested** **Done**

*Page 13, line 17: Also cite Borgmark and Wastegård (2008) here.*

**Reference added** **Done**

*Page 14, lines 12–14: Please, double-check the time periods here.*

**Yes, these will be revised (thanks for noting)**

**The sentence have now been re-written as follows:** *During the last two millennia, they found* *generally dry conditions until ca 1600 cal BP (ca.400 CE), varying water tables during the following four centuries, and dry conditions from ca 1200 to 700 cal BP (ca. 800-1300 CE, covering the MCA). The subsequent centuries were again variable, while the period 500-200 cal BP (1500-1800 CE, covering the LIA) was wet and the last two centuries dry except for the very recent years.*

*Page 15, lines 24–25: Add the also very relevant references Esper et al. (2002), Schneider et al. (2015), and Stoffel et al. (2016) here.*

**Reference added** **Done**

*Page 15, line 26: Add Shi et al. (2012) to the list of references here.*

**Reference added** **Done**

*Page 17, line 14: Medieval Warm Period/Medieval Climate Anomaly – medieval is too vague and a different meaning (as a time period) in history than as a climate period.*

**Quite right, a mistake and now changed as suggested** **Done**

*Northern Fennoscandian region during the Little Ice Age and Mediaeval period* **changed to** *Northern Fennoscandian region during the LIA and MCA*

*Page 18, line 11: Maybe it is worth to mention that a tree-line as high as 73N only occurs in parts of central Siberia (e.g. the Taimyr Peninsula)?*

**Changed as suggested**

**Within the Arctic 2k region, trees are naturally constrained to exist below the latitudinal tree line,** *extending as far as ca. 73°N in parts of central Siberia,*

*Page 22, line 31: "GISP-2" should be written "GISP2".*

**Yes, changed as suggested Done**

*Page 22, line 32: "O" in "Ymer O" should be with upper case "O".*

**Indeed! Changed as suggested Done**

Page 23: line 2: Geirsdóttir with "ó".

**Changed as suggested Done**

*Page 25, line 2: Why this two time periods for the MCA and LIA, respectively? Some motivation for the choice of time periods would be good. Most data for Fennoscandia indicates pretty old cold conditions during parts of the 12th century whereas many regions appear to have been rather, or very, warm during the 10th century (which also seems to have been the warmest century of the MCA in the Northern Hemisphere).*

**Quite right, this will be changed in the revised version.**

**In the text, we are now not including these time frames but we simply refer to LIA and MCA without any temporal indications.**

*Page 26, line 29: Add a reference to the new article by Helama et al. (2017) about the DACP here. The reference to Ljungqvist (2009) is wrong here: Ljungqvist is misspelt ("k" instead of "q") and it should be Ljungqvist (2010) – that discusses the DACP – and NOT Ljungqvist (2009) that does not do so.*

**You are completely right. Helama et al. (2017) added and the right Ljunqqvist paper cited (and so sorry about the misspelling). Done**

*Page 26, line 30: The word "disturbed" is ambiguous and vague here. It was cold but in what other ways "disturbed" compared to other periods. Larger variability in the climate?*

**We have rephrased as this:** *Apart from climatic changes related to temperature fluctuations, the DACP was likely a period of marked variable climate conditions.*

*Page 27, line 16: The word "variability" is misspelt here.*

**Corrected Done**

*Page 28, line 3: "PAGES" should be written with upper case letters (e.g. PAGES and not Pages).*

**Ooops, changed as suggested Done**

*Page 29, line 28: Also cite Schmidt et al. (2012) here.*

**Reference added Done**

*Line 31, 26: There is an error here: Ljungqvist et al. (2016) does NOT present a calibrated reconstruction. It is an uncalibrated index, ranging from –2 to +2, and with exceeding values truncated to –2 and +2, respectively. All values are standard deviations with respect to the mean of 1000–1899 CE. In some aspects, the approach in Ljungqvist et al. (2016) has some similarities with PDSI and other hydroclimate indices. So, in this respect it is no real differences between Ljungqvist et al. (2016) and the new hydroclimate index in this article.*

**Response: Thank you for spotting this mistake. We have now amended this in the text.**

**The sentence "***However, it should be remembered that L16 is a calibrated reconstruction, while the new Arctic hydroclimate record presented here is the average of a compilation of selected series***" was removed from the revised version.**

*Page 42, line 8. "ans" should be "and".*

**Changed as suggested Done**

*Page 61, line 20 (and in numerous citations throughout the article): Weissbach should be Weißbach.*

**Found the ß symbol, so changed as suggested Done**

[revised manuscript text omitted]

---

## Author Response (AR2)

**Dear editor**

**Many thanks for the thoughtful comments from you and the data review team. We have followed the suggestions closely when revising the manuscript. We hope that you find our efforts sufficient for accepting the manuscript. In the manuscript, we only highlight the major changes in the text (given in red). In addition, we have cleaned the text from typos etc., but to facilitate reading we did not highlight the small omissions or additions.**

Sincerely
The Arctic 2k writing team

Comment: Selection criteria: Selection criteria are given on page 27, but it isn't clear how that relates to records from table 3 that were not used. For example, it is unclear to me why the Greenland ice core accumulation records aren't used, but the Greenland lamina records are. Also, the accumulation records and lamina records are – I believe – essentially the same thing and a consistent terminology should be used. Both are derived by measuring the annual layers in ice cores.

**Response: As noticed by the reviewer, the selection criteria are given on page 26: "… all records i) are from north of 60°N; ii) extend back to at least 800 CE; iii) extend into the 1900s CE in order to include the warming period of the 20th century (PAGES 2k Consortium, 2013); iv) have an average sample resolution of less than 50 years; and v) have at least two age control points during the defined study period" [p26, L24-27].**

*Not all the records from table 3 were used because not all the records followed all of these criteria. To help the readers understand why several records were not selected, we included the following table (new table S2), which highlights which selection criteria were met or not..*

| ID Records | Be from north of 60°N | Extend back at least 800 CE | Extend into the 1900s CE in order to include the recent warming period | Have an average sample resolution < 50yrs | Have at least two age control points during the 800-2000 CE period |
|---|---|---|---|---|---|
| Hydro2k_01 | 🟩 | 🟩 | 🟥 | 🟩 | 🟩 |
| Hydro2k_02 | 🟩 | 🟩 | 🟥 | 🟩 | 🟥 |
| Hydro2k_03 | 🟩 | 🟥 | 🟥 | 🟥 | 🟥 |
| **Hydro2k_04** | 🟩 | 🟩 | 🟩 | 🟩 | 🟩 |
| Hydro2k_05 | 🟩 | 🟩 | 🟥 | 🟥 | 🟩 |
| Hydro2k_06 | 🟩 | 🟥 | 🟩 | 🟩 | 🟩 |
| **Hydro2k_07** | 🟩 | 🟩 | 🟩 | 🟩 | 🟩 |
| Hydro2k_08 | 🟩 | 🟩 | 🟥 | 🟩 | 🟩 |
| Hydro2k_09 | 🟩 | 🟩 | 🟩 | 🟥 | 🟩 |
| Hydro2k_10 | 🟩 | 🟥 | 🟩 | 🟩 | 🟩 |
| Hydro2k_11 | 🟩 | 🟥 | 🟥 | 🟩 | 🟩 |
| Hydro2k_12 | 🟩 | 🟩 | 🟩 | 🟥 | 🟩 |
| **Hydro2k_14** | 🟩 | 🟩 | 🟩 | 🟩 | 🟩 |
| **Hydro2k_15** | 🟩 | 🟩 | 🟩 | 🟩 | 🟩 |
| **Hydro2k_16** | 🟩 | 🟩 | 🟩 | 🟩 | 🟩 |
| **Hydro2k_17** | 🟩 | 🟩 | 🟩 | 🟩 | 🟩 |
| **Hydro2k_18** | 🟩 | 🟩 | 🟩 | 🟩 | 🟩 |
| **Hydro2k_19** | 🟩 | 🟩 | 🟩 | 🟩 | 🟩 |
| **Hydro2k_20** | 🟩 | 🟩 | 🟩 | 🟩 | 🟩 |

| | Col 1 | Col 2 | Col 3 | Col 4 | Col 5 |
|---|---|---|---|---|---|
| **Hydro2k_21** | 🟩 | 🟩 | 🟩 | 🟩 | 🟩 |
| **Hydro2k_22** | 🟩 | 🟩 | 🟩 | 🟩 | 🟩 |
| **Hydro2k_23** | 🟩 | 🟩 | 🟩 | 🟩 | 🟩 |
| Hydro2k_24 | 🟩 | 🟥 | 🟩 | 🟩 | 🟩 |
| Hydro2k_27 | 🟩 | 🟩 | 🟩 | 🟥 | 🟥 |
| **Hydro2k_31** | 🟩 | 🟩 | 🟩 | 🟩 | 🟩 |
| **Hydro2k_32** | 🟩 | 🟩 | 🟩 | 🟩 | 🟩 |
| **Hydro2k_34** | 🟩 | 🟩 | 🟩 | 🟩 | 🟩 |
| **Hydro2k_35** | 🟩 | 🟩 | 🟩 | 🟩 | 🟩 |
| **Hydro2k_37** | 🟩 | 🟩 | 🟩 | 🟩 | 🟩 |
| Hydro2k_38 | 🟩 | 🟥 | 🟩 | 🟩 | 🟩 |
| Hydro2k_39 | 🟩 | 🟥 | 🟩 | 🟩 | 🟩 |
| Hydro2k_40 | 🟩 | 🟥 | 🟩 | 🟩 | 🟩 |
| Hydro2k_41 | 🟩 | 🟩 | 🟩 | 🟥* | 🟩 |
| Hydro2k_42 | 🟩 | 🟩 | 🟩 | 🟥* | 🟩 |
| Hydro2k_43 | 🟩 | 🟥 | 🟩 | 🟩 | 🟥* |
| Hydro2k_44 | 🟩 | 🟥 | 🟩 | 🟩 | 🟥* |
| Hydro2k_45 | 🟩 | 🟥 | 🟩 | 🟩 | 🟥* |
| Hydro2k_46 | 🟩 | 🟥 | 🟩 | 🟩 | 🟥* |
| Hydro2k_48 | 🟩 | 🟥 | 🟩 | 🟩 | 🟥* |
| Hydro2k_49 | 🟩 | 🟥 | 🟩 | 🟩 | 🟥* |

*The published data are at annual resolution but result of linear interpolation. The original resolution is not sufficient to fit with the selection criteria.*

Comment: Model results: the paper uses results from just 3 models, but there isn't any explanation as to why only these three models were used? There are 9 different models available that ran last millennium + historical simulations for CMIP3.

Response: *Thanks for the comments. Only three models were used in the previous analysis due to their accessibility. At present, data from 3 more models are accessible, so we updated our analysis results using data from 6 models. We did not use the data from MIROC-ESM due to its millennium-long temperature trend bias.*

*Consequently, figs. 12, 13, S1 and S2 were updated. Also, table 4 was updated (now Table S3), as well as the text and figure captions corresponding to the figures.*

Comment: Please review text, figures and tables to ensure consistency in age representations. For example just across the figures "years CE", "BC/AD", CE/BCE", and "Time" are all used.

Response: *Done.*

Comment: Some of the figures need to be tidied up in their labelling. Please check all figures for completeness of presentation, and in particular ensure that figures 4, 5 and 11 are revised to improve labelling and

Response: *The figures have been tidied up and the captions more fitting.*

Comment: Ensure that δ18O (and other similar terms) use the "delta" symbol rather than just "d"

**Response:** *This has now been corrected (Table 3)*

Comment: Amongst the author team please carry out a thorough proof read of the manuscript. There are many typos that should be corrected in the revised manuscript.

**Response:** *Done.*

**The data stewardship team have also examined the revised manuscript and have the following comments**

Comment: The authors are making progress on their data management, but more work is needed before the paper is publishable. There remain many errors and issues with the data, which makes it difficult to review the data stewardship currently. The figure numbers are mixed up (mismatches in several places), citations to several datasets in Table 3 are incorrect, some datasets lack basic metadata, and a plan for long-term archival of some of the newly available data has not been identified (the excel file posted on the Wix.com site is not sufficient).

**Response:** *We have checked the figure numbers. See below for comments regarding citations in table three. The analysis results derived from climate model simulations have been permanently published on the website: https://figshare.com/articles/Data_Linderholm_et_al_2017/5729214*

Comment: In response to the data-review team's comment #5 regarding the use of the 850 AD truncated datasets (and as we discussed below), the authors say that "the original data sources are now given", but this does not appear to be the case? Before digging into a review, it would be helpful if the authors would correct obvious errors/ incorrect figure numbers/ incorrect citations and also prepare a more permanent open-access archive (e.g. ncdc palaeoclimate archive) for the newly available data.

**Response:** *Citations for the datasets in Table 3 were checked and the good reference for Sundqvist et al. (2014) is modified (see the new version of the table). A new data file was archived and available at https://doi.org/10.6084/m9.figshare.5683666.v1. It contains the metadata and the individual proxy series.*

*As reviewer #2 mentioned, some of the time series used were truncated from longer original record. We would like to precise here that a database is built to respond to a specific problematic, here the Arctic hydroclimate variability during the last millennium. In order to alert readers, a "Read Me" file is also added to explain that the archived data is parts of a longer original time series.*

[revised manuscript text omitted]

earlier work, this paper shows the potential for providing a high-resolution hydroclimate reconstruction for the Arctic and a comparison with last-millennium simulations from fully-coupled climate models. In general, hydroclimate proxies and simulations indicate that the Medieval Climate Anomaly tends to have been wetter than the Little Ice Age (LIA), but there are large regional differences. However, the regional coverage of the proxy data is inadequate, with distinct data gaps in most of Eurasia and parts of North America, making robust assessments for the whole Arctic impossible at the present. To fully assess pan-Arctic hydroclimate variability for the last two millennia additional proxy records are required.

**1. Introduction**

Global climate is changing rapidly, largely due to increased anthropogenic greenhouse gas emissions (IPCC, 2013). However, distinct regional differences in the magnitude of observed warming in recent decades are apparent; for example, the Arctic has warmed at more than twice the rate of the global average (Cohen et al., 2014). This *Arctic amplification* (Serreze et al., 2009) is due to complex feedback processes within the atmosphere-cryosphere-ocean system, including surface albedo and heat exchange with the ocean (Johannessen et al., 2003; Hind et al. 2016), and most importantly substantial losses of sea-ice extent and late-spring snow cover (Overland, 2014).

[revised manuscript text omitted]
. Long ice-cover season substantially reduces the input of particles from the watershed to the lake, frequently to an unmeasurable quantity. Therefore, what is recorded in Arctic lacustrine sediments is strongly biased towards the ice-free periods, i.e. spring snowmelt, short summer, and early autumn. Another charateristic of the Arctic is physical weathering related to gelifaction and sparse vegetation cover, making large quantities of easily-eroded minerogenic matter available to be transported into lakes (Zolitschka et al., 2015).

Lake systems in the Arctic differ depending on the presence or absence of glaciers in their watershed. Snowfed watersheds experience maximum discharge during snow melt in spring. They become depleted in water once the snow cover has melted, reducing sediment transport in the latter part of the ice-free season. On the other hand, glaciarised watersheds are not water-limited, i.e. the water supply to the lake tributary can last the entire summer and autumn until temperatures decrease below zero. Discharge, and therefore sediment transport, is usually driven by temperature at the elevation of the glacier and is usually at a maximum during summer. In addition, lake systems in glaciarised watersheds may on rare occasions be subjected to catastrophic floods (called Jökulhlaups) which are due to collapsing ice dams retaining vast amounts of water in intra- or supra-glacial lakes, and resulting in high sediment fluxes to the lakes.

Many watersheds in the Artic are also affected by the presence of permafrost. During summer, the permafrost thaws in its upper part (active layer), leaving sediment easily mobilised by small amounts of rainfall. This increases the risk of slope detachments, and can result in debris flows or very high sediment yields in lake tributaries (Lewis et al., 2005). The presence of permafrost also makes the dating of lacustrine sediments difficult because organic matter can be stored in the soils for a long period prior to being transported to the lakes (Abbott and Stafford, 1996).

In the High Arctic, sources of organic matter in lake sediments are both allochthonous and autochthonous, i.e. produced in the watershed or the lake. The relative contribution of these sources may, in part, be controlled by climate (Outridge et al., 2017), although the allochtonous organic mattter remains dominant, and the total amount preserved remains low (Abbott and Stafford, 1996; Galman et al., 2008). Conversely, lakes located in the southernmost part of the Arctic, such as in the Boreal Forest of

Scandinavia or North America, experience a season with higher primary productivity. Their total organic carbon content can be relatively high (Galman et al., 2008) when anoxic conditions at the bottom of the water column prevail, slowing down its degradation.

**3.1.2 Extracting hydroclimatic information from Arctic lakes**

5   Most of the proxies used elsewhere in the world in the purpose of reconstructing past hydroclimate can be also analysed in Arctic lakes. Extensive experience has enabled their use in the Arctic in spite of the harsh nature of the environment.

*Pollen* can successfully be used to reconstruct precipitation because the response of plants to moisture changes is direct and well-studied. Although a substantial proportion of the pollen in the High Arctic arrives from forested regions to the south,

10   pollen assemblages can still be used to reconstruct the local conditions (e.g. Gajewski, 2002; 2006; 2015b). The sediments may be contaminated by older pollen stored in the soils or, in some cases, from Tertiary deposits in the watershed (Gajewski et al., 1995). Nevertheless, annual precipitation has been reconstructed, along with temperature (Gajewski, 2015a) using pollen assemblages, and are presented in section 4.2.

15   *Chironomids* are primarily affected by lake depths, as well as is temperature and water chemistry, in the Arctic (Gajewski et al., 2005). Provided that changes in precipitation regimes 
[revised manuscript text omitted]
 900 to 1900 CE, the model ensemble mean shows slight negative trends in precipitation during spring, summer and winter, but an obvious negative trend in (Fig. S2a). Moreover, regional differences in long-term trends are indicated both within regions and between seasons in the six studied models (Fig. S2b). The implication of this is that in order to provide an average view of hydroclimate variability for the Arctic, there must be an even distribution of high-quality, numerically calibrated, verified and replicated climate-sensitive records.. More attention should also be paid to the target season of the climate signal when developing large-scale composites to avoid mixing of hydroclimate information across the seasons.

**6.2. Towards a better understanding of spatiotemporal hydroclimate variability in the Arctic**

Spatially explicit hydroclimate reconstructions provide excellent opportunities to study spatiotemporal variations, influences of forcings (e.g. Seager et al., 2007) and for proxy-model comparisons. However, due to the low number of available hydroclimate proxy records from the Arctic, and the imbalance in spatial coverage (Table 3, Fig. 8), it is currently impossible to prepare a field reconstruction for the whole region. As noted in section 3.3, there exist two tree-ring drought atlases covering parts of the Arctic (Fig. 3); however, the data representation is limited and the usage of temperature sensitive tree-ring proxies as hydroclimate indicators need to be properly addressed. Given the precipitation sensitivity of some high-latitude trees (St George and Ault, 2014), as well as more efforts in utilizing isotope records from trees, it may be possible to extend any analyses of hydroclimate variability into Eurasia. Targeted regional spatial reconstructions could be achieved for well-replicated regions, such as Fennoscandia, the Nordic Sea region, or western North America. To facilitate a compilation of Arctic hydrocliamte variability, a dedicated hydroclimate proxy database needs to be developed with firm criteria for which records to include. It is encouraging that several new hydroclimate records have been made available during the process of preparing

this review (see Table 2). Moreover, the new synthesis presented in section 5.1 shows the potential to provide regional hydroclimate records with high temporal resolution, providing useful information on multi-decadal timescales (e.g. Fig. 13).

**7. Recommendations for future work**

- Expanding the spatial coverage of hydroclimate proxy records is important, particularly for Eurasia, outside Fennoscandia, and North America. Several hydroclimate records that would add valuable information are not publicly available, so it is important to encourage palaeoclimate researchers to share and to publicly archive their data.
- A consistent and coherent Arctic2k hydroclimate database should be assembled by including all necessary meta data and information on seasonality of proxies included, following PAGES 2k data standard, to facilitate the development of site selection criteria for a more robust and defendable synthesis of Arctic hydroclimate history.
- A field reconstruction for regions with sufficient number of hydroclimate proxy records in time and space is critical to advance our understanding of dymanical controls of Arctic hydroclimate. Presently there seems to be opportunities for a trans-Atlantic comparison, which may shed light onto observed regional hydroclimate patterns and the forcing mechanisms.
- Closer collaborations between the palaeoclimate data and modelling communityies are needed to address and resolve discrepancies evident in comparisons 
[revised manuscript text omitted]

[Figure]

**Figure 3.** Drought Atlas reconstructions over the Arctic for North America (NADA, Cook et al., 2004, https://www.ncdc.noaa.gov/paleo/study/6319) and Europe (OWDA, Cook et al., 2015, https://www.ncdc.noaa.gov/paleo/study/19419). (a) The full spatial domains of the two atlases; and regional average over latitudes > 60°N in (b) Europe and (c) North America, transformed into z-scores and filtered with a 100-year loess (red lines).

[Figure]

**Figure 4.** Changes in subfossil Scots pine (*Pinus sylvestris L.*) sample numbers over time (black) from lakes in the central Scandinavian Mountains (Gunnarson 2008). The grey shaded area at the end represents living trees. Interpreted wet and dry periods are shown in grey breaks as well as the presence of Scots pines growing on a nearby peat bog indicating drier conditions (figure adapted from Gunnarson 2008). Data available at http://bolin.su.se/data/Gunnarson-2017.

[Figure]

Figure 5. Regional reconstructions of annual precipitation from the boreal zone of North America. The average of all pollen records from the different regions are shown. Grey silouettes are precipitation anomalies in mm; red lines are a LOESS fit (with a span of 0.2) to the data. The Beringa record is described in Viau et al. (2008) and the others in Viau et al. (2009). Some large anomalies are truncated; see Table 1 for data availability.

[Figure]

**Figure 6.** Annual precipitation reconstructions, based on pollen assemblages, from four lake sites in the Canadian Arctic: BC01, Melville Island (Peros et al., 2010); MB01, western Victoria Island (Peros et al., 2009); KR02, western Victoria Island (Peros et al., 2008); SL06, Boothia Peninsula (Peros et al., 2009). Dotted lines are LOESS lines fit to the data. For Lake KR02: red=Modern analogue technique, blue=WAPLS, black=PLS. See Table 1 for information on data availability.

[Figure]

**Figure 7.** Hydroclimatic variations in Sweden (SWE) and Finland (FIN) over the Common Era (see Table 2 for details including data availability). The mean levels (violet line) during the Medieval Climate Anomaly (MCA) and Little Ice Age (LIA) were calculated from the published records (black line), those being additionally smoothed using 200-year spline function (green line). Proxy data indicating change from MCA towards wetter (drier) LIA conditions are noted by plus (minus) sign. The graphs have been arranged so that wet conditions are indicated upward and dry conditions downward.

[Figure]

**Figure 8.** Spatial distribution of the hydroclimate proxy records available in the Arctic region. Records used for the new synthesis are highlighted by larger symbols and black borders. See Table 3 for information on the records.

[Figure]

**Figure 9.** Upper panel: Mean Pan-Arctic hydroclimate index (grey line) based on 17 selected series (Table 3). The thick black line is a 30-year LOESS filter, and the dashed lines are linear trends determined by a Mann-Kendall test. Data are presented
5    as z-scores. Lower panel: Number of time series over time included in the synthesis.

[Figure]

**Figure 10.** Wavelet analysis of the Pan-Arctic hydroclimate record as shown in Fig. 9. Colors represent the amplitude of the signal at given time and spectral period; red equals the highest power, blue the lowest. White line corresponds to cone of influence on the wavelet coherence spectrum and global wavelet spectrum. Confidence levels of 95% (α=0.05) are indicated on the wavelet spectrum with the black lines.

[Figure]

**Figure 11.** Regional hydroclimate mean series (grey) with 30-year LOESS filters (black) for the North Atlantic region (upper left) and Alaska (upper right). Data are presented as z-scores. Lower panels show the corresponding numbers of records through time included in the synthesis (see Table 3 for information on the records)

[Figure]

5 **Figure 12.** Spatial pattern of differences in annual hydroclimate between the MCA (950-1250 CE) and the LIA (1450-1850 CE) based on (a) hydroclimate reconstruction (Ljungqvist et al., 2016) and (b) ensemble mean of 6 last-millennium simulations. The values in (a) are z-scores of the hydroclimate index, while those in (b) are z-scores of the annual total precipitation. The z-scores are based on the period 850-1850 CE.

[Figure]

**Figure 13.** Comparison of centennial-scale annual hydroclimate variability (after application of a Gaussian filter) in the Arctic (≥ 60°N) North Atlantic region from a reconstruction by Ljungqvist et al. (2016, grey), the new Pan-Arctic hydroclimate proxy synthesis (red) and an ensemble mean of 6 last-millennium precipitation simulations (blue). See text for more information.

[Figure]

**Figure 14.** a) Variability and linear trends of Arctic spring, summer, autumn and winter total precipitation anomalies over the period 1900-2010 CE from the ERA-20C reanalysis dataset (Poli et al. 2013). b): Spatial patterns of linear trends of the Arctic spring, summer, autumn and winter total precipitation anomalies over the period 1900-2010 CE. Shading marks those
5  grid cells where the trend is significant (p<0.01).

**Supplementary material**

**Table S1** Twelve CMIP5 climate models used in this study. The spatial resolution of atmosphere is expressed by the number of longitudinal grid cells × the number of latitudinal grid cells.

| Model | Institute/Country | Spatial resolution of atmosphere |
|---|---|---|
| GFDL-CM3 | NOAA GFDL/USA | 144×90 |
| GFDL-ESM2G | NOAA GFDL/USA | 144×90 |
| GFDL-ESM2M | NOAA GFDL/USA | 144×90 |
| GISS-E2-H | NASA GISS/USA | 144×90 |
| GISS-E2-R | NASA GISS/USA | 144×90 |
| HadGEM2-AO | MOHC/UK | 192×145 |
| HadGEM2-ES | MOHC/UK | 192×145 |
| MIROC5 | MIROC/Japan | 256×128 |
| MIROC-ESM | MIROC/Japan | 128×64 |
| MIROC-ESM-CHEM | MIROC/Japan | 128×64 |
| MRI-CGCM3 | MRI/Japan | 320×160 |
| NorESM1-ME | NCC/Norway | 144×96 |

**Table S2** Selection criteria for including existing hydroclimate records in the synthesis. Green/red means that the record pass/fail the individual criterion. An asterix (*) indicates that the published data are based on linear interpolation of the original data to obtain annual resolution.

| ID Records | Be from north of 60°N | Extend back at least 800 CE | Extend into the 1900s CE in order to include the recent warming period | Have an average sample resolution < 50yrs | Have at least two age control points during the period 800-2000 CE |
|---|---|---|---|---|---|
| Hydro2k_01 | Pass | Pass | Fail | Pass | Pass |
| Hydro2k_02 | Pass | Pass | Fail | Pass | Fail |
| Hydro2k_03 | Pass | Fail | Fail | Fail | Fail |
| **Hydro2k_04** | Pass | Pass | Pass | Pass | Pass |
| Hydro2k_05 | Pass | Pass | Fail | Fail | Pass |
| Hydro2k_06 | Pass | Fail | Pass | Pass | Pass |
| **Hydro2k_07** | Pass | Pass | Pass | Pass | Pass |
| Hydro2k_08 | Pass | Pass | Fail | Pass | Pass |
| Hydro2k_09 | Pass | Pass | Pass | Fail | Pass |
| Hydro2k_10 | Pass | Fail | Pass | Pass | Pass |
| Hydro2k_11 | Pass | Fail | Fail | Pass | Pass |
| Hydro2k_12 | Pass | Pass | Pass | Fail | Pass |
| **Hydro2k_14** | Pass | Pass | Pass | Pass | Pass |
| **Hydro2k_15** | Pass | Pass | Pass | Pass | Pass |
| **Hydro2k_16** | Pass | Pass | Pass | Pass | Pass |
| **Hydro2k_17** | Pass | Pass | Pass | Pass | Pass |
| **Hydro2k_18** | Pass | Pass | Pass | Pass | Pass |
| **Hydro2k_19** | Pass | Pass | Pass | Pass | Pass |
| **Hydro2k_20** | Pass | Pass | Pass | Pass | Pass |
| **Hydro2k_21** | Pass | Pass | Pass | Pass | Pass |
| **Hydro2k_22** | Pass | Pass | Pass | Pass | Pass |
| **Hydro2k_23** | Pass | Pass | Pass | Pass | Pass |
| Hydro2k_24 | Pass | Fail | Pass | Pass | Pass |
| Hydro2k_27 | Pass | Pass | Pass | Fail | Fail |
| **Hydro2k_31** | Pass | Pass | Pass | Pass | Pass |
| **Hydro2k_32** | Pass | Pass | Pass | Pass | Pass |
| **Hydro2k_34** | Pass | Pass | Pass | Pass | Pass |
| **Hydro2k_35** | Pass | Pass | Pass | Pass | Pass |
| **Hydro2k_37** | Pass | Pass | Pass | Pass | Pass |
| Hydro2k_38 | Pass | Fail | Pass | Pass | Pass |
| Hydro2k_39 | Pass | Fail | Pass | Pass | Pass |
| Hydro2k_40 | Pass | Fail | Pass | Pass | Pass |
| Hydro2k_41 | Pass | Pass | Pass | Fail* | Pass |

| | | | | | |
|---|---|---|---|---|---|
| Hydro2k_42 | ■ | ■ | ■ | ■* | ■ |
| Hydro2k_43 | ■ | ■ | ■ | ■ | ■* |
| Hydro2k_44 | ■ | ■ | ■ | ■ | ■* |
| Hydro2k_45 | ■ | ■ | ■ | ■ | ■* |
| Hydro2k_46 | ■ | ■ | ■ | ■ | ■* |
| Hydro2k_48 | ■ | ■ | ■ | ■ | ■* |
| Hydro2k_49 | ■ | ■ | ■ | ■ | ■* |

**Table S3** Six PMIP3 climate models used in this study. The spatial resolution of the model is expressed by the number of longitudinal grid cells × the number of latitudinal grid cells.

| Model | Institute/Country | Spatial resolution of model |
|---|---|---|
| HadCM3 | MOHC/UK | 96×73 |
| IPSL-CM5A-LR | IPSL/France | 96×95 |
| MPI-ESM-P | MPI-M/Germany | 196×98 |
| CCSM4 | NCAR/USA | 288×192 |
| CSIRO-Mk3L-1-2 | UNSW/Australia | 64×56 |
| MRI-CGCM3 | MRI/Japan | 320×160 |

[Figure]

**Figure S1.** Spatial pattern of differences in annual hydroclimate between the MCA (950-1250 CE) and the LIA (1450-1850 CE) based on (a) HadCM3, (b) IPSL-CM5A-LR, (c) MPI-ESM-P, (d) CCSM4, (e) CSIRO-Mk3L-1-2 and (f) MRI-CGCM3 last-millennium simulations. The values are z-score (based on the period 850-1850 CE) of the annual total precipitation.

[Figure]

**Figure S2.** a) Variability and linear trends of Arctic spring, summer, autumn and winter total precipitation anomalies (900-1900 CE) from a six-model (HadCM3, IPSL-CM5A-LR, MPI-ESM-P, CCSM4, CSIRO-Mk3L-1-2 and MRI-CGCM3) ensemble mean. b) Spatial patterns of linear trends in seasonal total precipitation anomalies over (900-1850 CE) from 6 climate models. Shading marks those grid cells where the trend is significant (p<0.01).

180